acoustics/oceanography/biogeography

passive acoustic monitoring, oceanography, bioacoustics, weather, soundscape

**Author for correspondence:**
Victoria E. Warren
e-mail: vwar775@aucklanduni.ac.nz

# Marine soundscape variation reveals insights into baleen whales and their environment: a case study in central New Zealand

Victoria E. Warren[1,2], Craig McPherson[3], Giacomo Giorli[2], Kimberly T. Goetz[2,4] and Craig A. Radford[1]

[1]Institute of Marine Science, Leigh Marine Laboratory, University of Auckland, 160 Goat Island Road, Leigh 0985, New Zealand
[2]National Institute of Water and Atmospheric Research, 301 Evans Bay Parade, Hataitai, Wellington 6021, New Zealand
[3]JASCO Applied Sciences (Australia) Pty Ltd, 14 Hook Street, Unit 1, Capalaba QLD 4157, Australia
[4]National Oceanic and Atmospheric Administration, National Marine Fisheries Service, Alaska Fisheries Science Center, National Marine Mammal Laboratory, 7600 Sand Point Way NE, Seattle, WA 98115, USA

VEW, 0000-0001-9040-4831; CM, 0000-0002-9406-9994; GG, 0000-0003-1230-4901; KTG, 0000-0002-1356-0512; CAR, 0000-0001-7949-9497

Baleen whales reliably produce stereotyped vocalizations, enabling their spatio-temporal distributions to be inferred from acoustic detections. Soundscape analysis provides an integrated approach whereby vocal species, such as baleen whales, are sampled holistically with other acoustic contributors to their environment. Acoustic elements that occur concurrently in space, time and/or frequency can indicate overlaps between free-ranging species and potential stressors. Such information can inform risk assessment framework models. Here, we demonstrate the utility of soundscape monitoring in central New Zealand, an area of high cetacean diversity where potential threats are poorly understood. Pygmy blue whale calls were abundant in the South Taranaki Bight (STB) throughout recording periods and were also detected near Kaikōura during autumn. Humpback, Antarctic blue and Antarctic minke whales were detected in winter and spring, during migration. Wind, rain, tidal and wave activity increased ambient sound levels in both deep- and shallow-water environments across a broad range of frequencies, including those used by baleen whales, and sound from shipping, seismic surveys and earthquakes overlapped in time, space and frequency with whale calls. The

results highlight the feasibility of soundscape analysis to quantify and understand potential stressors to free-ranging species, which is essential for conservation and management decisions.

# 1. Introduction

Ambient acoustic environments, or soundscapes, consist of cumulative contributions from biological (biophonic), abiotic (geophonic) and man-made (anthrophonic) sound sources [1]. Globally, baleen whales are major biophonic contributors to marine soundscapes due to their loud, stereotyped communication calls, which are often emitted repeatedly as song [2–4]. The detection of baleen whales in acoustic data can provide insights into their distributions and movements, and locations of critical habitat [5–10]. Passive acoustic monitoring (PAM) using moored autonomous instruments is a useful method to study baleen whales, and other vocal species, consistently over time and space in areas that are challenging to access, such as polar regions and open ocean [6,10–13]. An acoustic approach can collect multi-species data and rapidly increase understanding of free-ranging vocal species (e.g. [10]).

Effective management relies on a foundation of knowledge about the prevalence and distributions of wild species and anthropogenic activities, as well as the potential interactions and relationships that can occur between them. Baleen whales are vulnerable to natural threats, such as predation and disease, as well as an array of anthropogenic threats, including entanglement in fishing gear, habitat degradation and ship strike [14]. Underwater sound can also be a stressor for marine mammals; acoustic signals that are perceived as threatening can generate acute physical and behavioural responses, as well as chronic effects, including stress and displacement [15–17]. Furthermore, the range over which baleen whale calls are detectable has been shown to decrease when other sound sources occur in the same frequency bands [17–24]. Reductions in communication and listening ranges can result in missed opportunities to reproduce or maintain fitness, the lack of appropriate behavioural responses, and if danger is unable to be perceived, increased risk [17,20]. Soundscape analysis permits an integrated approach to study free-ranging vocal species while concurrently collecting information regarding some potential stressors, which can be used to inform risk assessment frameworks [15,25,26]. Stressors can occur in isolation; as an aggregate exposure resulting from several similar sources (such as multiple sound sources); or cumulatively with other stressors experienced by free-ranging animals [26]. If vital rates of individual animals are affected by stressors, population-level consequences can occur [17,26–28].

Temporal and spatial overlaps between biophonic and anthrophonic soundscape contributors pinpoint where wildlife and human activities occur concurrently, and spectral overlap between contributors can be indicative of acoustic interference [29]. In areas where knowledge gaps exist, soundscape analysis provides a suitable research tool to holistically increase understanding and inform management decisions. Specific examples where soundscape data are valuable include Arctic areas, occupied by many threatened species, that are becoming accessible as summer ice cover reduces [25,30], and areas proposed for development [31]. Here, we present a case study in New Zealand to demonstrate the utility of soundscape analysis to increase understanding in a data-deficient area where anthropogenic activity occurs within a diverse marine environment. We aim to tease apart specific elements of the marine soundscape to investigate the distributions of individual natural and anthropogenic elements and provide insight into how these contributors could interact. Identification of interactions is the first step towards risk assessment and potential mitigation.

Relatively little is known about baleen whale distributions and movements around New Zealand due to constrained research opportunities arising from the logistical challenges of reaching animals in offshore areas that are often characterized by rough sea conditions [32–35]. Indeed, of the 10 baleen whale species assigned a national conservation status in New Zealand, seven are listed as 'data deficient' [36], including the Antarctic blue whale (*Balaenoptera musculus intermedia*), which is listed as 'critically endangered' at the global level [37]. Central New Zealand provides an ideal case study because the region is likely to offer a diverse array of sound types due to high biodiversity, complex oceanography and numerous anthropogenic activities. For example, Cook Strait, the main central waterway in New Zealand, hosts a commercial shipping route, an inter-island ferry crossing, a migratory corridor for baleen whales, and acts as a bottleneck for weather and sea conditions [35,38–43]. In this case study, autonomous acoustic recorders were deployed around central New Zealand to (i) learn more about the spatio-temporal distributions of baleen whales in an area with little existing information and (ii) to quantify how and why ambient sound levels vary throughout the region. The second objective enabled a holistic

assessment of potential stressors experienced by baleen whales, while providing a foundation of information about the marine soundscape of the region, which will be of value for future management.

# 2. Materials and methods

## 2.1. Data collection

Four autonomous multi-channel acoustic recorders (AMARs, JASCO Applied Sciences) were deployed from early June to mid-December 2016 around central New Zealand: in the South Taranaki Bight; in the narrows of Cook Strait; northeast of Kaikōura; and off the east coast of Wairarapa (henceforth referred to as STB, Cook Strait, Kaikōura and Wairarapa, respectively) (figure 1 and table 1). Three AMARs were redeployed between late February and early September 2017: two in the same relative Kaikōura and Wairarapa locations and one in STB, 25.2 km southeast of the first STB deployment location (figure 1 and table 1). A recorder was not redeployed in Cook Strait in 2017. The AMARs at Cook Strait and STB were bottom-mounted on metal baseplates, with the hydrophone 75 cm off the seafloor, in water depths less than 300 m. At Kaikōura and Wairarapa, ultra-deep AMARs were deployed in water depths exceeding 1000 m and were moored on vertical line moorings, approximately 10 m above the seabed. All AMARs were retrieved via the use of acoustic releases and buoyancy aids.

Acoustic sampling was duty cycled over 900 s: 630 s at a sampling rate of 16 kHz, 125 s at a sampling rate of 250 kHz and 145 s of sleep. Only files with 16 kHz sampling rate were considered in this study due to the focus on low-frequency baleen whale sounds. Recorder sensitivity was calibrated via pistonphone, and all data were corrected by frequency- and recorder-specific sensitivity curves. The nominal pressure sensitivity level of the hydrophones was $-165$ dB re 1 V $\mu Pa^{-1}$ ($\pm 1$ dB) from 10 Hz to 100 kHz. The pressure spectral density noise floor was limited by the hydrophone preamplifier at 32 dB re 1 $\mu Pa^2 \, Hz^{-1}$. Analyses focused on frequency values between 10 and 7000 Hz to omit high-intensity infrasound and the effect of Nyquist roll-off close to 8000 Hz.

## 2.2. Acoustic processing

Acoustic analyses were conducted in Matlab [44]. Power spectral density (PSD) was calculated with 1 Hz resolution for each 630 s, 16 kHz file using a fast Fourier transform with window length of 16 000 samples (Hanning window) and 80% overlap (pwelch function, Matlab). The PSD outputs were used to compute spectral probability density (SPD) [45] with PSD bin widths of 1 dB re 1 $\mu Pa^2 \, Hz^{-1}$. The SPDs were calculated per deployment and per month at each station. In order to investigate temporal trends and broadband sounds, long-term spectral averages (LTSAs) were generated by concatenating the 1 Hz resolution PSD data for all files and displaying as scaled colours. Daily decidecade band medians centred at 12.5, 25, 63, 125, 250, 1000 and 4000 Hz were calculated from the per-file PSD outputs per station and per deployment by integrating PSD values within linear space [46].

## 2.3. Analysis of soundscape contributors

### 2.3.1. Biophonic

When PSD levels were noticeably higher than average in a particular frequency band, the data files corresponding to these periods were individually examined to determine the origins of the increased acoustic levels. Specific features within the data were visualized as spectrograms in PAMlab-Lite [47] using spectrogram parameters that best displayed the features of interest. Cetacean species were identified by comparing spectrograms of detected sounds with spectrograms of known calls produced by the same or similar species elsewhere [3,4,48–52]. For other suspected biophonic sounds that were not thought to be produced by marine mammals, sound levels were examined against day and night-time indicators (obtained for the location of Wellington and applied across all sites (https://www.gaisma.com/en/location/wellington.html)) as biological sounds often feature diel cycles as well as seasonal trends (e.g. [6,53,54]).

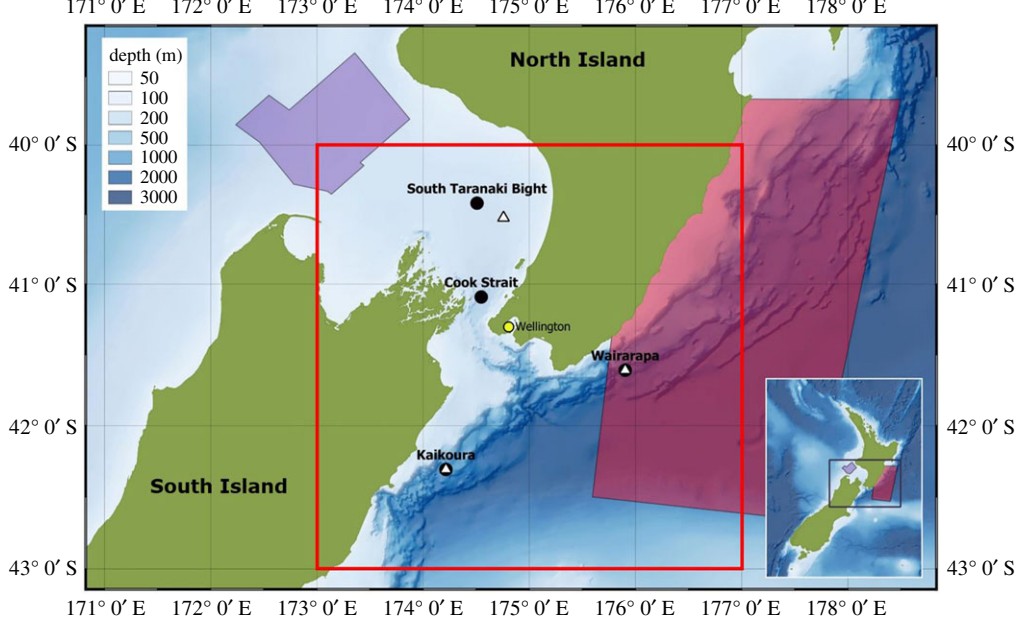

**Figure 1.** Map of central New Zealand, in context with its location in mainland New Zealand (inset). AMAR deployment locations illustrated by black circles (first deployment, June to December 2016) and white triangles (second deployment, February to September 2017). Location of the weather station in Wellington (yellow circle), boundaries of earthquake data (open red box), operational borders of the PGS Taranaki South three-dimensional seismic survey (purple filled area) and Schlumberger Pegasus Basin three-dimensional seismic survey (red filled area) are also illustrated.

**Table 1.** Acoustic deployment information per recording location (time standard: UTC).

|  | South Taranaki Bight | Cook Strait | Wairarapa | Kaikōura |
|---|---|---|---|---|
| deployment 1 |  |  |  |  |
| position | 40.42° S, 174.50° E | 41.09° S, 174.55° E | 41.61° S, 175.90° E | 42.31° S, 174.21° E |
| water depth (m) | 110 | 252 | 1481 | 1251 |
| deployment | 4 June 2016 | 4 June 2016 | 5 June 2016 | 6 June 2016 |
| retrieval | 20 Dec 2016 | 19 Dec 2016 | 21 Dec 2016 | 21 Dec 2016 |
| deployment 2 |  |  |  |  |
| position | 40.53° S, 174.76° E | — | 41.61° S, 175.90° E | 42.31° S, 174.21° E |
| water depth (m) | 112 | — | 1486 | 1254 |
| deployment | 23 Feb 2017 | — | 22 Feb 2017 | 21 Feb 2017 |
| retrieval | 29 Aug 2017 | — | 30 Aug 2017 | 8 Sept 2017 |

### 2.3.2. Anthrophonic

Worldwide, shipping noise concentrates in decidecade bands centred at 63 and 125 Hz [55]. Here, the contribution of vessel noise to the soundscape was examined via decidecade bands centred at these frequencies. Vessel contributions were also illustrated by LTSAs and SPD analyses.

Information regarding seismic survey activity conducted around central New Zealand during the recording periods was derived from Marine Mammal Impact Assessments (MMIAs) available from the New Zealand Department of Conservation website (https://www.doc.govt.nz/our-work/seismic-surveys-code-of-conduct/marine-mammal-impact-assessments/). Operational area coordinates were obtained from the MMIAs, or from Land Information New Zealand 'Notices to Mariners' (https://www.linz.govt.nz/sea/maritime-safety/notices-mariners). Seismic survey pulse sequences were detected in the data using an automated detector, for which full details are provided by Martin [56].

**Table 2.** Category values for rainfall and wind speed scenarios, based on data from Greta Point, Wellington. Also, category values for modelled significant wave height scenarios, based on modelled wave height data across all four recording locations.

|  | no | low | mid | high |
|---|---|---|---|---|
| rainfall (mm) | 0 | 0–1.3 | 1.3–2.6 | 2.6–3.9 |
| wind speed (m/s) | n/a | 2.1–10.9 | 10.9–19.8 | 19.8–28.6 |
| modelled significant wave height (m) | n/a | 0.1–3.4 | 3.5–6.7 | 6.8–10.2 |

Detections were made on a daily scale over the 16 kHz data and compared against the known periods of seismic survey activity from the above sources.

### 2.3.3. Geophonic

Weather data were obtained from the CliFlo database (https://cliflo.niwa.co.nz/) for the weather station located at Greta Point, Wellington (figure 1). Hourly rainfall and average hourly wind speed were obtained for both acoustic deployment periods. Twenty-four-point (one day) moving averages were placed over the hourly rain and wind data to clarify periods of persistent weather. Within the averaged data, the total ranges of wind and rain were split into three and classified as 'low', 'medium' or 'high' (table 2). Rain was also classified as 'no rain' if the average value was 0 mm (table 2). For each of the four recording locations, the 630 s acoustic files corresponding to the timestamps of wind and rain scenarios were extracted, and the median PSD was calculated for each weather scenario to examine the impact of varying wind speed and rainfall on the marine soundscape.

Wave statistics were obtained from the NZWAVE-NZCSM wave forecast, an implementation of the third-generation wave model Wavewatch III® Version 4.18, part of NIWA's operational environmental forecasting system. Significant wave heights (SWH, mean height of the highest 33% of waves) were extracted for the latitude and longitude of each AMAR location across both deployment periods. The total range of SWHs was split into three to indicate 'low', 'mid' and 'high' SWH scenarios (table 2). The median PSDs of acoustic files recorded with the same timestamps as the 'low', 'mid' and 'high' wave scenarios were calculated and compared to examine the influence of wave activity on the soundscape.

Modelled sea-level (tidal) data from the two deployment periods were obtained for acoustic recording locations from NIWA's online tide forecaster (https://www.niwa.co.nz/services/online-services/tide-forecaster). Timestamps of high tide, low tide, mid-flood tide (low tide +3 h), and mid-ebb tide (low tide −3 h) were extracted, and the median PSDs of the corresponding acoustic data at each station were computed and visualized per tidal state.

The timings and magnitudes of all earthquakes (exceeding magnitude 0.5) that occurred during the two deployment periods, between 40–43° S, 173–177° E (figure 1) were extracted from the Geonet database (https://www.geonet.org.nz/) and used to examine the effect of natural seismic activity on the soundscape of the region.

## 3. Results

At all recording locations, median PSDs were highest below 100 Hz, ranging from 75 to 97 dB re 1 µPa$^2$ Hz$^{-1}$ (figure 2). Above 100 Hz, sound levels decreased to values of 49–55 dB re 1 µPa$^2$ Hz$^{-1}$ at 7000 Hz (figure 2). At Cook Strait, PSD at frequencies above 70 Hz exceeded other stations by approximately 5 dB re 1 µPa$^2$ Hz$^{-1}$ while following the same overall trend (figure 2). Dips in PSD at 1500 and 2000 Hz occurred consistently during both deployments at Wairarapa and Kaikōura (figure 2). This was an artefact resulting from destructive interference due to the separation distance between the AMAR hydrophone and the spherical glass floats deployed at these deep-water locations and was not a true feature of the soundscape.

Between approximately 300 and 1000 Hz, the probability of a given PSD value (its SPD) showed central tendencies for each 1 Hz value, at all recording locations (figure 3). SPD showed greater variation at lower and higher frequencies. At STB and Cook Strait, variation below 100 Hz occurred across PSDs of 60 to 130 dB re 1 µPa$^2$ Hz$^{-1}$ (figure 3a and b). At Wairarapa and the first deployment at Kaikōura, a visible separation occurred in the SPDs at frequencies below 250 and 100 Hz, respectively (figure 3c and d). At these locations, the two low-frequency SPD modes were centred

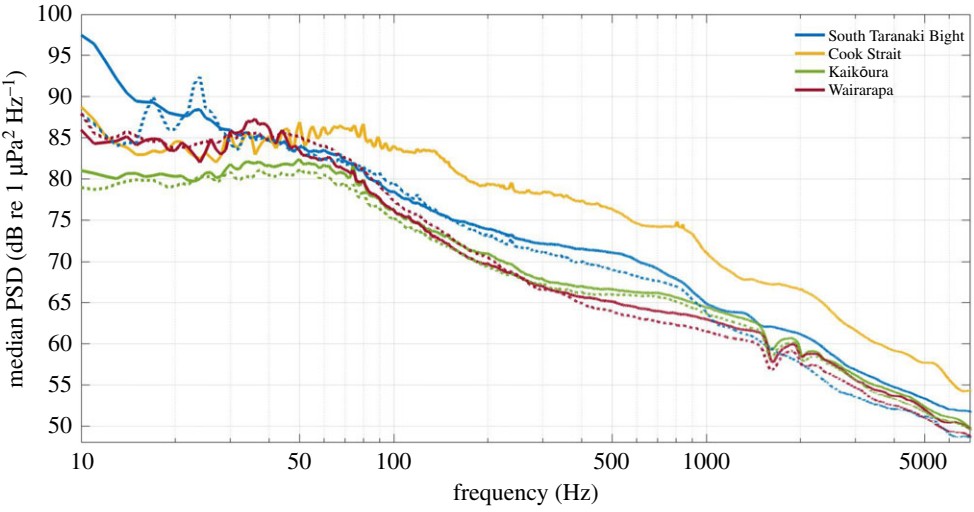

**Figure 2.** Median PSD per station, per deployment (solid line = June to December 2016, dotted line = February to September 2017), 1 Hz frequency resolution. An AMAR was not redeployed in Cook Strait in 2017. PSD dips at 1500 and 2000 Hz at Wairarapa and Kaikōura were due to destructive interference and were not true features of the soundscape.

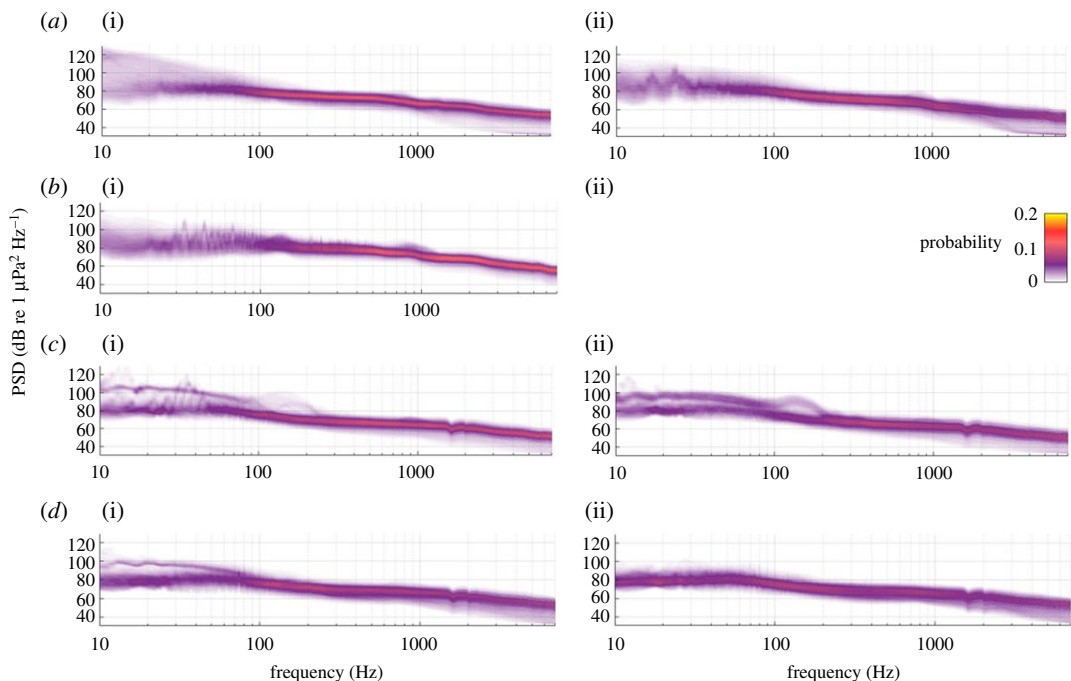

**Figure 3.** Spectral probability density (SPD) per Hz of ambient sound recorded at (*a*) STB, (*b*) Cook Strait, (*c*) Wairarapa and (*d*) Kaikōura for the 2016 (i) and 2017 (ii) deployments. The probability of a given power spectral density (PSD) at a given frequency is represented as colour (colour bar is applicable to all axes). An AMAR was not redeployed in Cook Strait in 2017. PSD dips at 1500 and 2000 Hz at Wairarapa and Kaikōura were due to destructive interference and were not true features of the soundscape.

around 80 and 100 dB re 1 $\mu$Pa$^2$ Hz$^{-1}$ (figure 3*c* and *d*). This implied that some of the data from Wairarapa and Kaikōura contained low-frequency sound that was approximately 20 dB re 1 $\mu$Pa$^2$ Hz$^{-1}$ louder than data recorded at other times during the deployments.

## 3.1. Biophony

### 3.1.1. Baleen whales

Calls from two sub-species of blue whale were recorded in the region and contributed significantly to the marine soundscape. Pygmy blue whale (*B. m. brevicauda*) calls contributed sound that peaked at 16/17

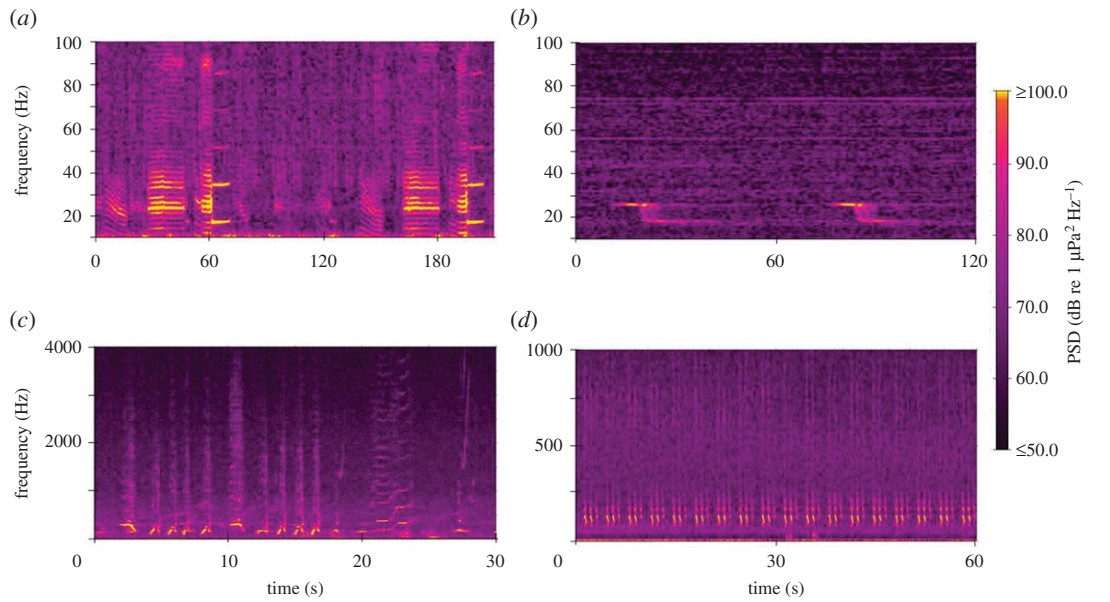

**Figure 4.** Baleen whale vocalization spectrograms. (*a*) Pygmy blue whale calls, South Taranaki Bight, 16 July 2016. (*b*) Antarctic blue whale Z calls, Kaikōura, 27 June 2016. (*c*) Humpback whale song, South Taranaki Bight, 22 July 2016. (*d*) Antarctic minke whale 'bio-duck', South Taranaki Bight, 2 October 2016. (*a,b*) Spectrograms computed with frequency resolution of 0.244 Hz, 2 s Hamming-windowed data and 75% overlap. (*c,d*) Spectrograms computed with frequency resolution of 1.95 Hz, 0.128 s Hamming-windowed data and 75% overlap. Note scale differences on all axes.

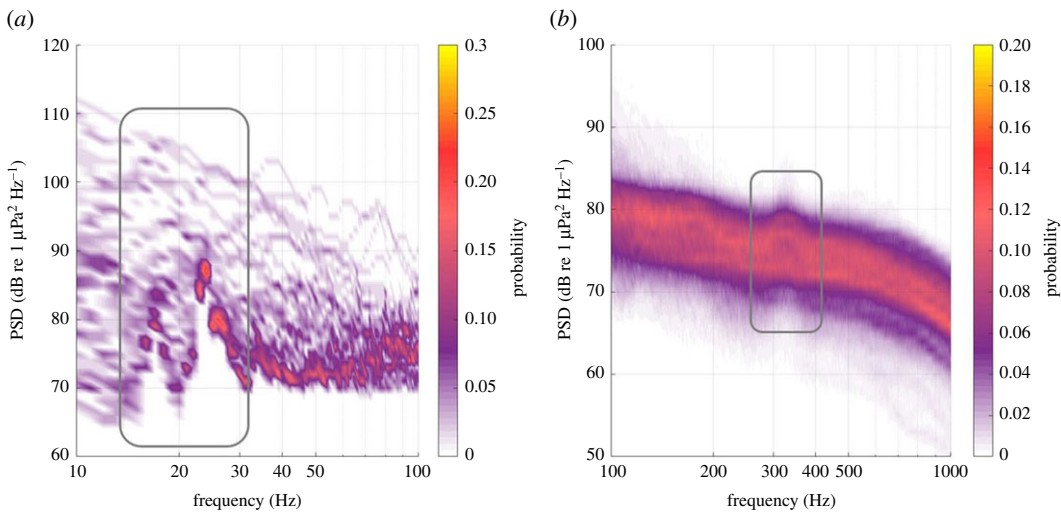

**Figure 5.** Monthly spectral probability densities of data recorded at South Taranaki Bight featuring baleen whale calls. The probability of a given power spectral density (PSD) at a given frequency is represented as colour. (*a*) In April 2017, pygmy blue whale calls generated peaks in PSD at 16/17 and 23 Hz (grey box). (*b*) In July 2016, humpback whale song generated a peak in PSD between 250 and 400 Hz (grey box). Note scale differences on both axes.

and 23 Hz (figures 4*a* and 5*a*, electronic supplementary material, file S1). These calls were most apparent in the STB soundscape, where they were evident throughout both deployment periods (figures 2 and 3*a*), particularly during March and April 2017 (austral autumn) (figures 6*b* and 7*b*). Pygmy blue whale calls were also apparent at Kaikōura during March 2017 (figure 6*e*). Antarctic blue whale calls (known as 'Z' calls) contributed sound at 17/18 and 26 Hz (figure 4*b*, electronic supplementary material, file S2) and were notable in the soundscape recorded at Kaikōura at the end of June 2016 (figure 6*e*). Song produced by humpback whales (*Megaptera novaeangliae*) was evident in the soundscape of the STB between 250 and 400 Hz during July 2016 (figures 4*c*, 5*b* and 6*b*, electronic supplementary material, file S3). The 'bio-duck' vocalization of the Antarctic minke whale (*Balaenoptera bonaerensis*) [49] was

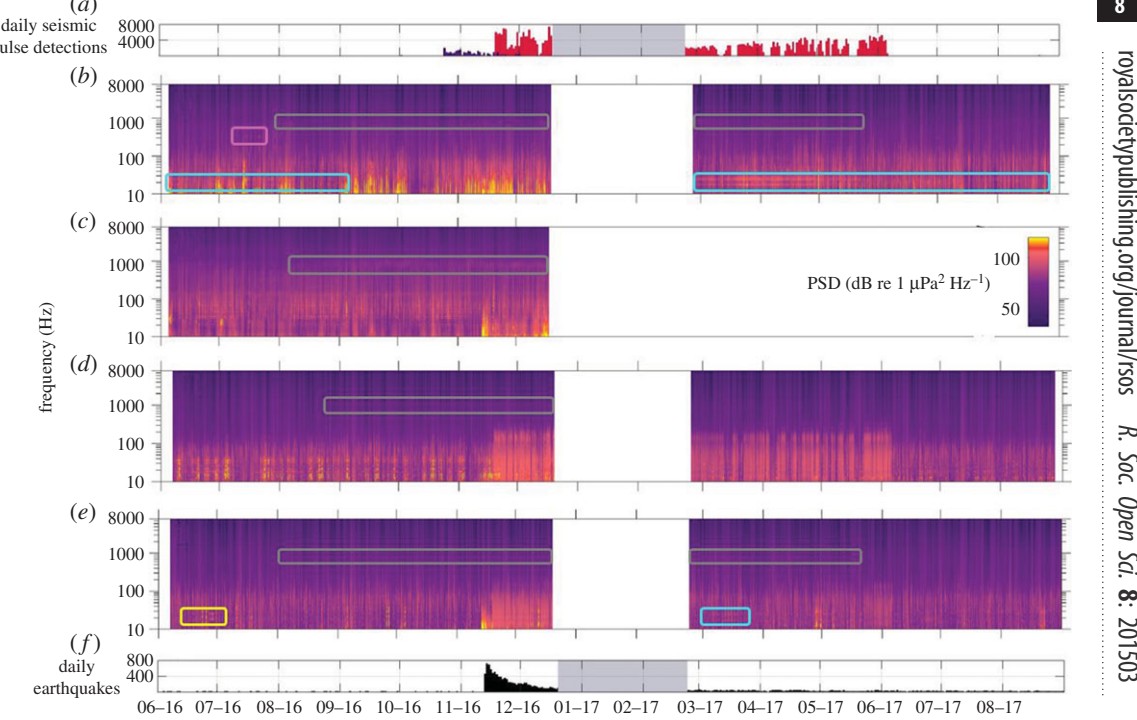

**Figure 6.** Long-term spectral averages of (*b*) South Taranaki Bight (STB), (*c*) Cook Strait, (*d*) Wairarapa and (*e*) Kaikōura for both deployments. White areas indicate periods when the AMARs were not deployed. The power spectral density (PSD) of a given frequency at a given date/time is represented as colour (colour bar in (*c*) is applicable to all axes). (*a*) Illustrates the output of the seismic survey pulse detector for the STB data (purple bars) and the Wairarapa data (red bars) for the 16 kHz files (70% duty cycle). Detections at STB are of the PGS Taranaki South three-dimensional seismic survey, and detections at Wairarapa are of the Schlumberger Pegasus Basin three-dimensional seismic survey. (*f*) Illustrates the total number of earthquakes (magnitude greater than 0.5) in the study area (red box, figure 1), per day (obtained from https://www.geonet.org.nz/). Contributions from biophonic sources are indicated: humpback whales at STB (pink box (*b*)); pygmy blue whales at STB and Kaikōura (blue boxes (*b*,*e*)); Antarctic blue whales at Kaikōura (yellow box (*e*)); and putative biological chorus at all locations (grey boxes (*b*,*c*,*d* and *e*)).

also noted (figure 4*d*, electronic supplementary material, file S4), as the broadband pulsed nature of this sound caused false positive detections by the seismic survey detector. 'Bio-duck' detections occurred at STB in early October 2016 and at Kaikōura in August 2016 and July 2017. It is not known how reflective these detections are of the actual spatial and temporal presence of Antarctic minke whales.

### 3.1.2. Other biophonic sources

During spring, summer and autumn, PSD was raised between approximately 600 and 1100 Hz at all recording locations to varying extents (grey boxes, figure 6). As well as seasonal variation, diel variation was also evident in this soundscape contributor; when this sound was present, the 1000 Hz-centred decidecade band level was higher during daylight hours than during hours of darkness (figure 8). The individual calls associated with the raised frequency band were not evident in the recordings, but the seasonal and diel patterns of the sound implied that it was produced by one or more biophonic sources, and the occupied frequency band did not match the frequency of known marine mammal vocalizations.

## 3.2. Anthrophony

### 3.2.1. Shipping

The sound of shipping was present in the data recorded at all locations. Shipping sound consisted of persistent tonal sounds or broadband cavitation sounds which increased in received level and bandwidth when vessels passed near the AMARs; the Lloyd mirror effect was commonly evident

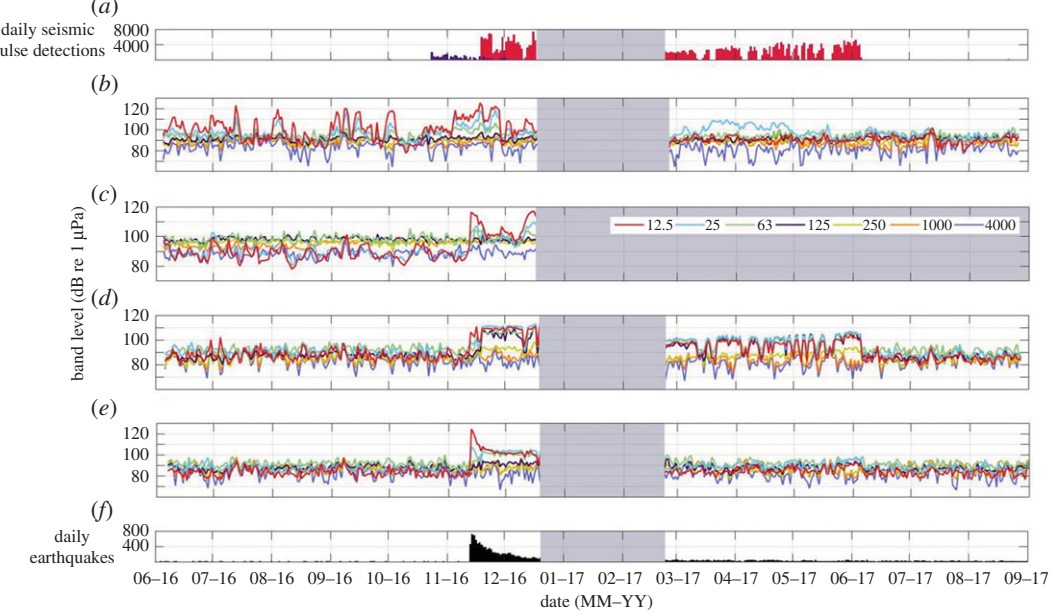

**Figure 7.** Daily median decidecade band levels over both deployments at (*b*) South Taranaki Bight (STB), (*c*) Cook Strait, (*d*) Wairarapa and (*e*) Kaikōura. Centre frequencies of the decidecade bands are provided in (*c*). Grey areas indicate periods when the AMARs were not deployed. (*a*) and (*f*) are the same as described in figure 6.

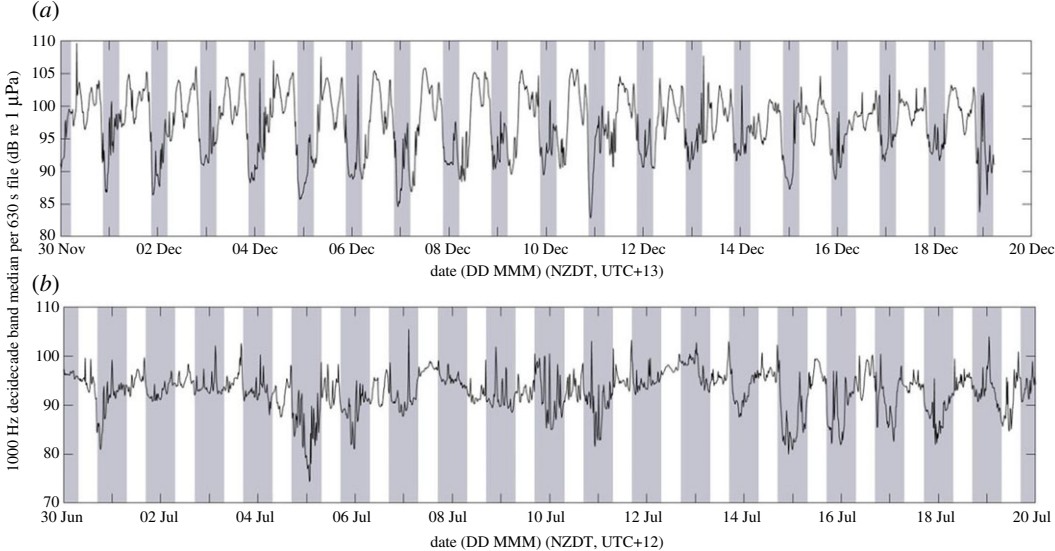

**Figure 8.** (*a*) The median value of the 1000 Hz decidecade band at Cook Strait from 30 November to 20 December 2016. Grey areas indicate night (21.00–5.00, NZDT). PSD is raised in the 1000 Hz decidecade band during daylight as a result of suspected biological chorusing. (*b*) The median value of the 1000 Hz decidecade band at Cook Strait from 30 June to 20 July 2016. Grey areas indicate night (17.00–7.00, NZDT) and demonstrate the absence of diel pattern in the 1000 Hz decidecade band due to a lack of suspected biological chorusing.

during close ship passes (figure 9). Figure 9 illustrates an example where the sound produced by a passing ship occupied the frequency bands being used by one or more Antarctic blue whales (yellow box, figure 9). In LTSAs, the ship passes appeared as short-duration, broadband peaks in intensity, and these were most abundant at Cook Strait (figure 6c). Persistent frequency-specific tonal sounds associated with vessels were also evident below 100 Hz at this recording location (figures 3b and 6c). At Cook Strait, the 63 and 125 Hz-centred decidecade bands, which are proportional to vessel sound, were the loudest decidecade bands, except during November and December 2016 when earthquake sound raised bands centred at lower frequencies (figure 7c). Outside of November 2016 to June 2017, when a seismic survey influenced band levels, the 63 Hz-centred band was commonly loudest at

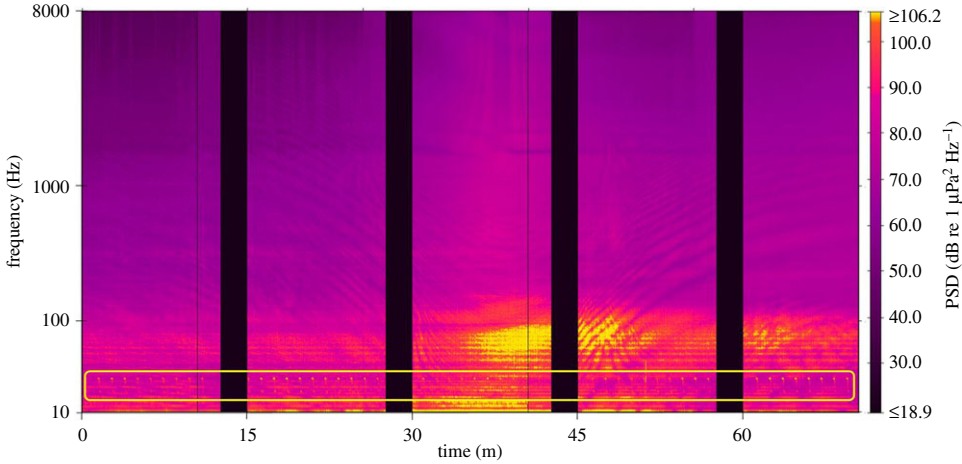

**Figure 9.** Eighty minute spectrogram of a ship pass recorded at Kaikōura, 18 June 2016. Antarctic blue whale 'Z' calls are evident as vertical signals between 18 and 26 Hz throughout the illustrated period (yellow box). The spectrogram includes data sampled at both 16 and 250 kHz, and black vertical bars correspond with duty cycled gaps between recordings (spectrogram computed with frequency resolution of 0.244 Hz, 2 s Hamming-windowed data and 75% overlap).

Wairarapa and Kaikōura, although the band levels here were approximately 5 dB re 1 µPa quieter than their equivalents at Cook Strait (figure 7*d* and *e*).

### 3.2.2. Seismic surveys

Two seismic surveys occurred in the region during the acoustic deployments (figure 1) and impulses from both surveys were detected, with close alignment to known periods of seismic surveying (illustrated in figures 6*a* and 7*a*). In addition to true seismic impulses, the detector made a small number of false detections outside of survey periods that were caused by Antarctic minke whale 'bio-duck' pulses, and unknown sound sources.

The PGS Taranaki South three-dimensional seismic survey (23 October to 2 December 2016) occurred closest to the STB recorder (figure 1) and was only detected in the soundscape recorded at STB. Although airgun pulses from the PGS Taranaki South three-dimensional seismic survey were detected in the STB (figure 6*a*), PSD at low frequencies (less than 100 Hz) was highly variable during the first deployment in this location, and the contribution of the seismic survey was not apparent in the LTSA or the decidecade band levels for this location (figures 6*b* and 7*b*).

The Schlumberger Pegasus Basin three-dimensional seismic survey was conducted along the east coast of central New Zealand (figure 1) and was detected in both the Kaikōura and Wairarapa data from 18 November 2016 to 6 June 2017 (figure 6*a*). Detections were fewer at Kaikōura (maximum of 1400 detections during a day, compared with a maximum of 7500 during a day at Wairarapa) but covered the same temporal range. The Schlumberger Pegasus Basin three-dimensional seismic survey was evident in the soundscape at Wairarapa between approximately 100 and 300 Hz (figure 6*d*) and to a lesser extent at Kaikōura over the same frequency range (figure 6*e*). This seismic survey contributed to the upper SPD mode at 100 dB re 1 µPa$^2$ Hz$^{-1}$ at Wairarapa during both deployment periods (figure 3*c*). Throughout its duration, the Schlumberger Pegasus Basin three-dimensional seismic survey influenced the decidecade bands centred at 250 Hz and below at Wairarapa, with band level increases of up to 20 dB re 1 µPa (figure 7*d*). At Kaikōura, decidecade bands centred at 125 Hz and below were raised during the period of the survey (figure 7*e*). During the survey period, received sound was not consistent due to the varying distance of the survey from the acoustic recorders, and survey operations, such as line turns and shutdowns.

## 3.3. Geophony

### 3.3.1. Weather

No 16 kHz acoustic data were recorded during periods of 'mid' or 'high' rainfall at any recording location. Compared with 'no' rainfall, 'low' rainfall corresponded with sound levels that were approximately 2–5 dB re 1 µPa$^2$ Hz$^{-1}$ louder at frequencies above 200 Hz at all four recording locations (figure 10). The influence

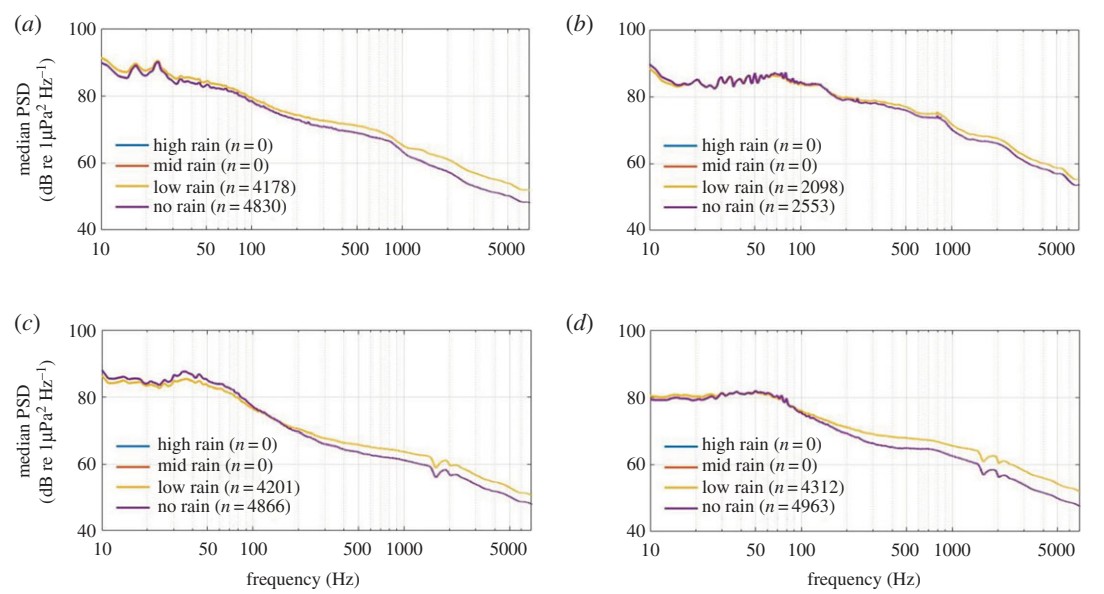

**Figure 10.** Median power spectral densities (PSDs) corresponding to given rainfall scenarios for (a) South Taranaki Bight, (b) Cook Strait, (c) Wairarapa and (d) Kaikōura. PSD dips at 1500 and 2000 Hz at Wairarapa (c) and Kaikōura (d) were due to destructive interference and were not true features of the soundscape.

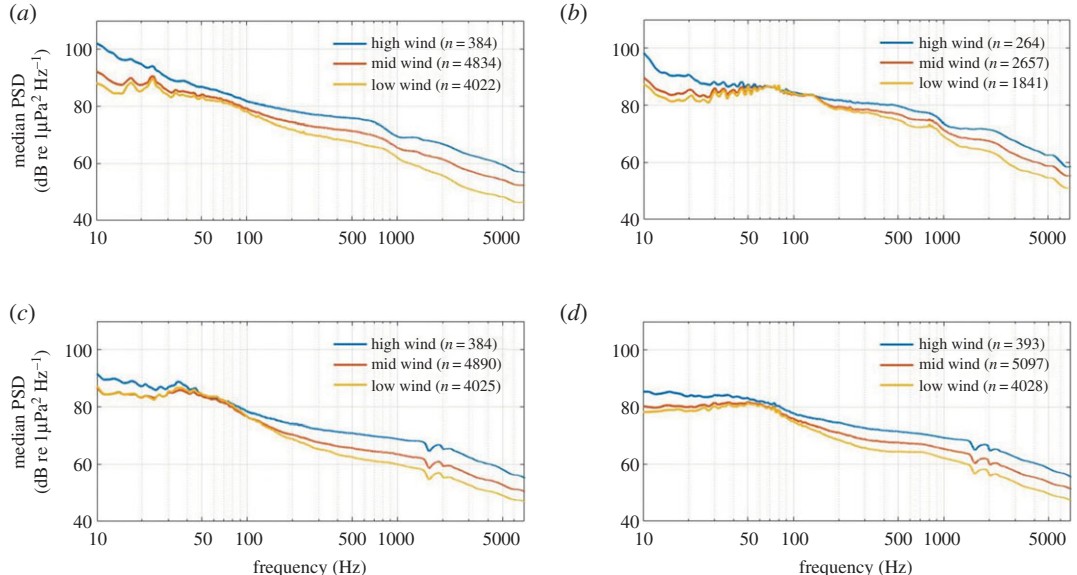

**Figure 11.** Median power spectral densities (PSDs) corresponding to given wind speed scenarios for (a) South Taranaki Bight, (b) Cook Strait, (c) Wairarapa and (d) Kaikōura. PSD dips at 1500 and 2000 Hz at Wairarapa (c) and Kaikōura (d) were due to destructive interference and were not true features of the soundscape.

of wind speed on the soundscape of the area was also evident in PSD levels, across a wider range of examined frequencies. In general, 'high' wind resulted in the largest PSD values across all frequencies, approximately 3–5 dB re 1 $\mu$Pa$^2$ Hz$^{-1}$ louder than 'mid' wind scenarios, with 'low' wind generating the lowest PSD scenarios (figure 11). Some acoustic convergence occurred among the scenarios between 30 and 150 Hz across all recording locations (figure 11).

### 3.3.2. Wave activity

The highest minimum and maximum SWHs were noted at Wairarapa (minimum = 0.6 m, maximum = 10.2 m), while the lowest minimum and maximum SWHs occurred at Cook Strait (minimum = 0.1 m, maximum = 5.4 m). While the range of SWH varied between stations, the influence of SWH was

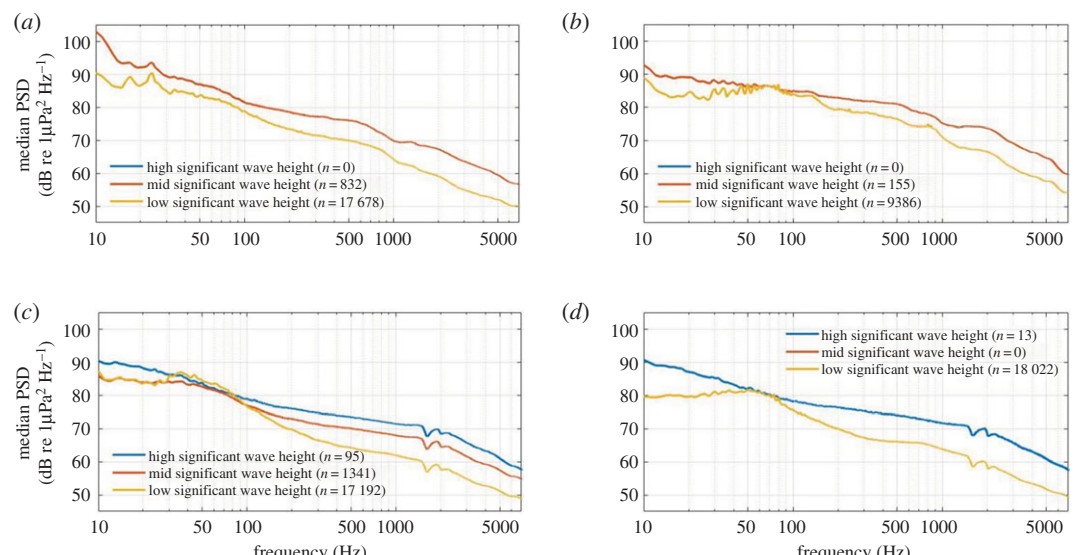

**Figure 12.** Median power spectral densities (PSDs) corresponding to given modelled significant wave height scenarios for (*a*) South Taranaki Bight, (*b*) Cook Strait, (*c*) Wairarapa and (*d*) Kaikōura. PSD dips at 1500 and 2000 Hz at Wairarapa (*c*) and Kaikōura (*d*) were due to destructive interference and were not true features of the soundscape.

similar at all stations. Excepting some convergence between 50 and 100 Hz, larger SWHs corresponded with louder PSD values across all frequencies and locations (figure 12). Some of the convergence between 50 and 100 Hz at Wairarapa (figure 12*c*) was due to the presence of seismic survey pulses which increased the median PSD value of the 'low' SWH scenario in this frequency band. The 50–100 Hz bandwidth at which the wave scenarios converged was congruent with the convergence displayed by the wind scenarios, probably due to the physical relationship that exists between wind and wave elements.

### 3.3.3. Tidal state

At STB and Cook Strait, median PSD levels below 50 Hz were up to 15 dB re 1 µPa$^2$ Hz$^{-1}$ higher during flooding and ebbing tidal periods than during high and low tide periods (figures 13*a* and *b*), while median PSD was similar across all tidal states at Kaikōura (figure 13*d*). High tide generated the loudest median sound levels below 100 Hz at Wairarapa, with PSD up to 15 dB re 1 µPa$^2$ Hz$^{-1}$ higher than during low, flooding or ebbing tide (figure 13*c*). Above 100 Hz, median PSD did not differ between tidal scenarios at any recording location (figure 13).

### 3.3.4. Natural seismic activity

During the first acoustic deployment, 12 122 earthquakes occurred within the study area with a maximum magnitude of 7.8 (median = 2.5). A total of 6508 earthquakes occurred during the second deployment period, with a maximum magnitude of 5.2 (median = 2.0). Earthquakes in the study region, particularly those of higher magnitude, were concentrated along the east coast of the South Island. Within a 50 km radius of STB and Wairarapa, approximately the same number of earthquakes occurred during both deployments (312 and 390 at STB, and 25 and 12 at Wairarapa); however, 3.4 times as many earthquakes occurred within a 50 km radius of Kaikōura during the first deployment compared with the second deployment (2778 compared with 816).

Earthquakes were evident in the acoustic data as sudden onset, high-energy, low-frequency (generally less than 100 Hz) broadband sound lasting a few seconds to a few minutes. Earthquakes occurring outside of the study region probably also contributed to recorded sound levels. A magnitude 7.8 earthquake occurred on 14 November 2016 (NZDT). This large earthquake, and consequent aftershocks that persisted into December 2016, were evident in data recorded at all locations, except STB, and were most apparent at Kaikōura due to its closest proximity to the epicentre (figures 6*e* and *f*, and 7*e* and *f*). The magnitude 7.8 earthquake resulted in sound levels that briefly exceeded the dynamic range of the AMAR at Kaikōura. Decidecade bands centred at 12.5, 25 and 63 Hz at Cook Strait, Wairarapa and

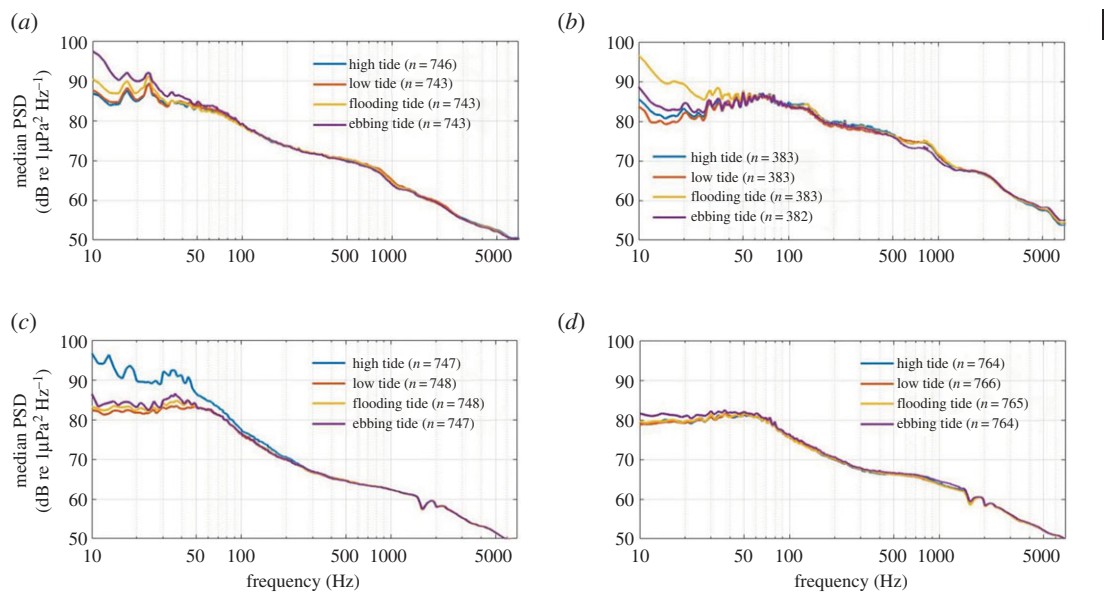

**Figure 13.** Median power spectral densities (PSDs) corresponding to given tidal states for (*a*) South Taranaki Bight, (*b*) Cook Strait, (*c*) Wairarapa and (*d*) Kaikōura. PSD dips at 1500 and 2000 Hz at Wairarapa (*c*) and Kaikōura (*d*) were due to destructive interference and were not true features of the soundscape.

Kaikōura were raised by earthquake activity (figure 7); at Kaikōura, the decidecade band centred at 12.5 Hz was raised from approximately 85 to 124 dB re 1 µPa by the initial earthquake (figure 7*e* and *f*). This period of earthquake activity contributed to the upper SPD mode (around 100 dB re 1 µPa$^2$ Hz$^{-1}$) that occurred below 100 Hz at Kaikōura during the first deployment (figure 3*d*).

# 4. Discussion

Soundscape analysis is a non-invasive research approach that can elucidate the spatio-temporal distributions of specific sound contributors in a holistic manner that is integrated with information about other sound emitters and the wider environment. Such studies can be used to identify areas where free-ranging species could face risks as a result of spatial, temporal, or spectral overlap between contributors [57], and findings can be used to inform and direct management decisions [58]. Limited knowledge regarding the spatio-temporal distributions of baleen whales and baseline ambient sound levels around central New Zealand made the region an ideal case study to demonstrate the utility of soundscape analysis. Characterization of the marine soundscapes at four recording locations around central New Zealand revealed the spatio-temporal distributions of migratory and resident baleen whales, as well as their overlap in time, space and frequency with natural phenomena, shipping and seismic survey activity. We demonstrated that an acoustic approach can increase understanding of data-deficient areas, which is essential for effective management.

## 4.1. Spatio-temporal distributions of baleen whales

Passive acoustic monitoring (PAM) proved to be a valuable tool to elucidate the spatio-temporal distributions of baleen whales in offshore waters where sea conditions are notoriously poor [33,34,40]. Here, notable acoustic contributions were made by pygmy blue whales, Antarctic blue whales and humpback whales. Previous observations of blue whales in New Zealand waters have often conflated pygmy and Antarctic blue whales as it is challenging to visually distinguish between the two sub-species at sea (see [59]). However, conflation is not an optimal approach as sub-species specific management of the 'critically endangered' Antarctic blue whale is vitally important [37,60]. An acoustic approach enabled blue whale sub-species to be distinguished with ease (see also [3,9,43,61]). The pulsed calls of Antarctic minke whales were also identified in the data presented here, and to the best of our knowledge, this is the first documentation of the 'bio-duck' sound [49] in New Zealand waters.

Antarctic blue whale calls and humpback whale song were temporally constrained to austral winter, indicating that central New Zealand is a migratory corridor for both species that feed around Antarctica in austral summer and breed in warmer waters during austral winter [38,48,62]. Humpback whales that migrate through New Zealand waters have been linked to breeding grounds in East Australia and Oceania [42,63–66]. The breeding areas of Antarctic blue whales in the western South Pacific are largely unknown [67]. At the soundscape level presented here, Antarctic blue whales were detected along the shelf edge to the east of central New Zealand in a deep-water environment. Movements along shelf edges are common for blue whales [8,67–69] and may result from a desire to communicate with distant conspecifics via the acoustic propagation properties afforded by deep-water environments [70]. Conversely, humpback whales were detected in the shallow, sheltered bay of the South Taranaki Bight (STB), suggesting that their northbound migration route followed the opposite coast to the Antarctic blue whales. However, when the calls of these two species were examined in the data using a targeted approach rather than an overall soundscape-level approach, both Antarctic blue whales and humpback whales were found to use migration routes along both the east and west coasts of central New Zealand [42,43]. Antarctic minke whales were detected on both coasts, during winter and spring, and this species is also assumed to conduct seasonal migrations [71]. In the present study, neither Antarctic blue whale calls or humpback whale song were noted in the soundscape during austral spring when southbound migration would be expected [38]. It is possible that these species migrated southward through the study area without producing an abundance of calls and were therefore not evident in the overall soundscape; indeed, Antarctic blue whale calls were detected in the region during austral spring when the data were analysed using a targeted approach [43]. Migratory routes are assumed to be stable, at least over short evolutionary timescales [72]; however, neither humpback nor Antarctic blue whale calls were notable in this soundscape-level analysis during winter 2017. New Zealand recorded an extreme heatwave in 2017 [73], and it is possible that oceanographic conditions in 2017 may have been unusual, causing the whales to favour alternative migration routes. Only male blue and humpback whales are thought to produce the calls and song reported here, and detections may not be representative of female, juvenile and non-vocal whales [4,74,75].

In general, pygmy blue whales do not migrate to Antarctica to feed [67], and their distributions are thought to be determined by prey availability [76–78]. In central New Zealand, upwelled water flows into the South Taranaki Bight (STB) year-round, making it a highly productive area [32] and blue whale feeding behaviour in the STB has been widely reported [76,79–81]. Accordingly, the soundscape of the STB featured near-consistent contributions from pygmy blue whales. Prey availability may have also underpinned humpback whale presence in the STB if the whales undertook opportunistic feeding during migration [82]. The acoustic contribution of pygmy blue whales in the STB was greatest in austral autumn (March and April), suggesting that the whales were calling more frequently, or aggregating in the area, during this time [43]. Calls from pygmy blue whales were also evident in the soundscape recorded at Kaikōura during austral autumn, promoting further study within the vicinity of the Kaikōura deployment location to understand how pygmy blue whales use the area, and whether their presence is an annual event.

## 4.2. Marine soundscape variation and overlaps between contributors

Fine spatial-scale variation was demonstrated in the marine soundscape of this relatively small region in central New Zealand. Environmental conditions within the detection area of an acoustic receiver can directly influence the reception of sound [83,84] and the four recording locations studied here differed in sediment type, propagation environment, exposure and depth. For example, Wairarapa featured the greatest contribution from airguns due to its close proximity to the Schlumberger Pegasus Basin three-dimensional seismic survey, but deep water along the east coast enabled the sound of the survey to propagate effectively, and sound from this contributor was also detected in the soundscape recorded at Kaikōura. Outside of the seismic survey periods, ambient sound levels recorded at the deep-water AMARs off the east coast were generally lower than those recorded in the shallower Cook Strait and STB. For sounds with an abiotic origin, particularly those related to weather events, the lower sound levels at the deeper recorders probably resulted from increased propagation loss due to the greater distance from the surface.

Overall, sound levels above 70 Hz were approximately 5 dB re 1 $\mu Pa^2 Hz^{-1}$ louder at Cook Strait compared with sound levels recorded in deep water off the east coast, or in the STB. All of the soundscape contributors recorded at Cook Strait were also recorded at the other locations, but in environments that were deeper or less enclosed, individual soundscape elements were less concentrated. Cook Strait features a coarse pebble seabed [40,85], which is associated with acoustic wave resonance and decreased transmission loss [84]. For example, sound levels in the 63 and 125 Hz-centred decidecade

bands were disproportionally high at Cook Strait, implying that the acoustic contribution of vessel traffic was concentrated here [55]. Cook Strait is a key shipping route, but it is also a known migratory corridor for large whale species (e.g. [39]); therefore, the potential for ship strike in the narrow channel is possible. Ship strike is a global threat to large whales [86–88], and sound resonation in the Strait could make it difficult for whales to localize and consequently avoid ship traffic. In the Northern Hemisphere, North Atlantic right whales (*Eubalaena glacialis*) are particularly vulnerable to ship strike as they do not exhibit behavioural responses to ship sound [89]. Although southern right whales (*Eubalaena australis*) were not acoustically evident in the present data, their populations are recovering from the effects of whaling and they are returning to former habitats, including central New Zealand [90,91]. Southern right whales could be at risk of ship strike in the narrow channel of Cook Strait if they demonstrate a similar lack of response to ship sound—their responses have not been quantified. As global shipping activity increases and whale populations recover from the effects of whaling, this potential interaction could become more pertinent, both in the study area and elsewhere. Vessel speed restrictions have been shown to reduce the lethality of ship strikes, and slower speeds can also have positive implications for sound pollution produced by shipping [22,92–94].

Natural variation in ambient sound levels can influence the ranges over which sounds are detectable, which can have consequences for the communication abilities of vocal species [18,22,95,96]. Here, increases in wind, rain and wave conditions corresponded with increased sound levels across the analysed frequency range in both shallow water and deep water (as also demonstrated by [97–100]). Tidal movement also increased sound levels below 100 Hz at most recording locations (also reported by Boebel *et al.* [101]), although it should be noted that moving receivers, such as swimming animals, are unlikely to perceive tidal sound in the same way as the moored recorders examined here. Additional acoustic contributions from other sources, such as anthropogenic sources, can further influence the perception of sound by biological organisms [22,25,95]. For example, the communication space of Bryde's whales (*Balaenoptera edeni brydei*) in the Hauraki Gulf, northern New Zealand, was estimated to fluctuate in size with natural variation in ambient noise levels, but was significantly reduced by shipping sound contributions [22]. In the present study, the ability of vessel sound to reduce the detectability of baleen whale calls at the hydrophones was demonstrated directly (figure 9). Moreover, the Schlumberger Pegasus Basin three-dimensional seismic survey and earthquakes along the east coast contributed cumulative sound across shared low frequencies in November and December 2016, raising levels of decidecade bands centred below 250 Hz by more than 30 dB re 1 µPa at times. The low-frequency bands influenced by earthquakes and seismic survey activity were used by all noted baleen whale species, and the events may therefore have affected the communication abilities of the whales. For humpback and Antarctic blue whales that conduct seasonal migrations through New Zealand waters, missed communication opportunities could cause individuals to become separated from conspecifics, and may hamper acoustic mate attraction that occurs prior to arrival at the breeding areas [4,75]. However, communication implications can only be considered in light of the auditory processing capabilities of the receiving animal, which for many marine mammal species, particularly baleen whales, is unknown (see [20] for a review). In addition to potential communication consequences, impulsive sounds, such as those emitted by seismic surveying, can elicit behavioural responses from marine mammals, as well as broad-scale displacements and changes in their own sound emissions [15,17,27,102–104]. The high-intensity sound produced by earthquakes could also elicit similar effects, particularly as earthquakes occur without warning or ramp-up period [105].

In the present study, a chorus was detected between 600 and 1100 Hz that featured seasonal and diel trends. The sound was deemed to be of biological origin due to the evident seasonal and diel variation in intensity [53], and a similar chorus, recorded in the eastern Arabian Sea, was suspected to be produced by planktivorous fish [98]. All soniferous species can be vulnerable to acoustic disturbance. Raised natural and anthropogenic sound levels in the frequency band of the chorus may mask the chorusing to conspecifics: in some locations, fish choruses have been found to mask each other [106]; while anthropogenic sounds can also reduce communication space of fish species through masking [22]. The presence of the chorus at all recording locations, despite differences in the local environment (including water depth and proximity to shore), promotes further investigation.

Knowing how specific events impact soundscapes and other contributors can help tailor planned anthropogenic activities to cause the least interference with the habitats and acoustic niches of wild species [29]. Soundscape planning is a modern approach that encourages anthropogenic activities to be targeted in locations, times and frequency bands that cause the least impact to wildlife [15,29]. In order to achieve this, it is essential to understand the relative distributions of both stressors and receivers (e.g. baleen whales) [26]. Here, humpback whale song was detected in the STB, where pygmy blue whales

and a large amount of marine industry (as discussed by [81]) also occurred. Pygmy blue and humpback whales produced calls at notably different frequencies and at differing times of the year in central New Zealand, highlighting the need for thorough planning regarding the timing and sound emission characteristics of anthropogenic activities to minimize risk to these species. The fitness and vital rates of individual animals can be affected by exposure to ambient noise, and chronic exposure can lead to population-level consequences [17,26,28]. This study adds to a growing body of literature demonstrating that PAM should be considered an important part of environmental analyses as soundscape analysis is a useful tool to collect the required holistic data to inform management decisions.

# 5. Conclusion

The case study presented here demonstrated that PAM is an invaluable tool to increase understanding of the spatio-temporal distributions of baleen whale species in a region where previous research has been constrained by the inaccessibility of offshore animals and challenges in identification. Soundscape monitoring provided a relatively large-scale, continuous solution to study whale distributions non-invasively, while concurrently sampling the ambient soundscape to investigate the environment in which the whales occurred in an integrated manner. Overlaps between soundscape contributors revealed potential conflicts, both in terms of physical risks and acoustic interference. The information gained from soundscape analysis can be invaluable to inform conservation and management decisions.

Data accessibility. Open access NIWA data (matrices of per-file 1 Hz PSD, for each recording location) available from Dryad, doi:10.5061/dryad.vmcvdncpj [107].

Authors' contributions. V.E.W. conducted all analyses and wrote the manuscript. All authors contributed to study design, interpreted data and critically revised the manuscript. K.T.G., G.G. and V.E.W. assisted with data collection. All authors gave final approval for publication.

Competing interests. The authors declare no competing interests.

Funding. This work was supported by OMV New Zealand Ltd, Chevron New Zealand Holdings LLC, Marlborough District Council, and Woodside Energy [Woodside Marine Mammal Research Grant, awarded to V.E.W.]. V.E.W. is also funded by the University of Auckland Doctoral Scholarship. G.G. and K.T.G. received co-funding by NIWA Coasts and Oceans Programme 4.

Acknowledgements. Thanks to Christopher Whitt (JASCO Applied Sciences) for training and deployment advice, and to the NIWA acoustic deployment and retrieval teams: Sarah Searson, Mike Brewer, Olivia Price, Fiona Elliott, and the crews of the RVs Tangaroa, Kaharoa and Ikatere. Richard Gorman (NIWA, Hamilton) provided wave statistics from the NZWAVE-NZCSM wave forecast and contributed the associated paragraph in Materials and methods. Further thanks to Rochelle Constantine (University of Auckland), Craig Stevens (University of Auckland and NIWA, Wellington), Malcolm Francis (NIWA, Wellington), Matt Pinkerton (NIWA, Wellington) and two anonymous reviewers for feedback that improved the manuscript. The findings and conclusions in this paper are those of the authors and do not necessarily represent the views of the National Marine Fisheries Service, NOAA. Mention of trade names and commercial firms does not imply endorsement by the National Marine Fisheries Service, NOAA.

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
