## [Peer Review File · Royal Society Open Science]

Review History

RSOS-200117.R0 (Original submission)

Review form: Reviewer 1

Is the manuscript scientifically sound in its present form?

No

Are the interpretations and conclusions justified by the results?

No

Is the language acceptable?

Yes

Do you have any ethical concerns with this paper?

No

Have you any concerns about statistical analyses in this paper?

Yes

Recommendation?

Reject

Comments to the Author(s)

A large amount of work clearly went into making the measurements and processing the data of this paper. The figures are well done, and the paper is well written. I had multiple technical and grammatical details that are captured in the sticky notes of the reviewed pdf (Appendix A). My main concern is in the messaging and packaging of the manuscript content. At present, the manuscript is more a description of the soundscape in New Zealand at a specific time. There are many papers in the literature that describe a soundscape at a particular place and time. It is already well known that soundscapes vary with biotic, abiotic, and anthropogenic contributions. What this manuscript needs is a focusing question that is answered or a testing of an ecological concept. In both the Intro and Discussion, the authors correctly point out that soundscapes can provide information about the ecological function of a soundscape, but this was not successfully accomplished in the paper. A more comprehensive synthesis or application to a specific need or question would greatly increase the value of this work. It may be more appropriate for a regional regulatory or management journal.

Review form: Reviewer 2

Is the manuscript scientifically sound in its present form?

Yes

Are the interpretations and conclusions justified by the results?

Yes

Is the language acceptable?

Yes

Do you have any ethical concerns with this paper?

No

Have you any concerns about statistical analyses in this paper?

No

Recommendation?

Accept with minor revision (please list in comments)

Comments to the Author(s)

See attached file (Appendix B).

Decision letter (RSOS-200117.R0)

23-Mar-2020

Dear Miss Warren:

Manuscript ID RSOS-200117 entitled "Marine soundscape variation in an oceanographically diverse region, central New Zealand" which you submitted to Royal Society Open Science, has been reviewed. The comments from reviewers are included at the bottom of this letter.

In view of the criticisms of the reviewers, the manuscript has been rejected in its current form. However, a new manuscript may be submitted which takes into consideration these comments.

Please note that resubmitting your manuscript does not guarantee eventual acceptance, and that your resubmission will be subject to peer review before a decision is made.

Your resubmitted manuscript should be submitted by 20-Sep-2020. If you are unable to submit by this date please contact the Editorial Office.

on behalf of Dr Asha de Vos (Associate Editor) and Pietro Cicuta (Subject Editor)
openscience@royalsociety.org

Associate Editor Comments to Author (Dr Asha de Vos):

The comments provided by the reviewer are incredibly comprehensive and will no doubt guide what seems like a rewrite of the existing paper. I hope you will see this as an opportunity to improve what has been put together. I think it is a valuable addition, in the event that these comments are addressed and as such I am rejecting but allowing resubmission (at which point we will send the paper for review once again).

Thank you

Reviewers' Comments to Author:

Reviewer: 1
Comments to the Author(s)

A large amount of work clearly went into making the measurements and processing the data of this paper. The figures are well done, and the paper is well written. I had multiple technical and grammatical details that are captured in the sticky notes of the reviewed pdf. My main concern is in the messaging and packaging of the manuscript content. At present, the manuscript is more a description of the soundscape in New Zealand at a specific time. There are many papers in the literature that describe a soundscape at a particular place and time. It is already well known that

soundscapes vary with biotic, abiotic, and anthropogenic contributions. What this manuscript needs is a focusing question that is answered or a testing of an ecological concept. In both the Intro and Discussion, the authors correctly point out that soundscapes can provide information about the ecological function of a soundscape, but this was not successfully accomplished in the paper. A more comprehensive synthesis or application to a specific need or question would greatly increase the value of this work. It may be more appropriate for a regional regulatory or management journal.

Reviewer: 2

Comments to the Author(s)

See attached file.

Author's Response to Decision Letter for (RSOS-200117.R0)

See Appendix C.

RSOS-201503.R0

Review form: Reviewer 2

Is the manuscript scientifically sound in its present form?

Yes

Are the interpretations and conclusions justified by the results?

Yes

Is the language acceptable?

Yes

Do you have any ethical concerns with this paper?

No

Have you any concerns about statistical analyses in this paper?

No

Recommendation?

Accept with minor revision (please list in comments)

Comments to the Author(s)

I appreciated the new version of the manuscript and the approach that the authors took. The authors now frame the study to describe the acoustic presence of baleen whales and changes in the soundscape that they may perceive, which includes an in-depth look at sources of variation. I believe that the authors have addressed all my previous comments from the first review. I only have a few minor comments, which I detail below.

Line 1: "varied" I would recommend using the present tense here (vary) as you have not introduced the results yet.

Line 16: "with the hydrophones mounted to the AMARs." This statement is not necessary, correct? You already stated that the AMARs were moored on vertical line moorings. I am assuming that the AMARs already incorporate hydrophones?

Line 30: I do not think that you need a comma before "and per month" nor before "at each station."

Line 39: "energy" is not the appropriate term here. I recommend replacing energy with levels or inputs.

Page 4

Line 2: This sentence would benefit from the addition of a citation or two.

Lines 5-7: Decidecades are exactly the same as 1/3 octave bands assuming that they are base ten bands, if I am not mistaken. These two sentences open the door for confusing the reader. I would recommend not even mentioning 1/3 octave bands here and just refer to the measurements as decade bands as they are understood by the community and the preferred term by Reviewer 1 in the previous review.

Page 5

Lines 33-35: Given the bandwidth of the biological sound, it is likely that the rise in PSD levels might originate from more than one source. Consider adding in this possibility. Additionally in regards to Figure 8, the duration of daily increases in the 1000 Hz decade is much longer than one might expect from a single species if these daily increases are definitely biologic in origin, especially if it was produced by a fish species. However, it could be related to feeding sounds of mixed-fish-species origin. Another possibility would be from invertebrates I suppose. In any event, it would be good to include that it might originate from multiple sources.

Page 7

Line 10: Any idea why the high tide at Wairarapa would yield the loudest median sound levels below 100 Hz? Could another source of sound or pressure contribute to this result?

Figure 4: In response to my suggestion for including .wav files as supplementary material for this figure, the authors stated that "We do not believe this is necessary - similar sounds can easily be found online." I would suggest that they reconsider as I cannot find an example of a pygmy blue whale online, or at least easily.

Figure 6: It is difficult to see the components of the LTSAs as presented. Would it be possible to have two figures; one for the first deployment and one for the second deployment? Alternatively, could this figure be presented in "landscape" format. I would like to be able to see the details a bit better. For example, the biological chorus is virtually undetectable in the LTSAs. Maybe this can be addressed in the proofing stage.

Review form: Reviewer 3

Is the manuscript scientifically sound in its present form?

Yes

Are the interpretations and conclusions justified by the results?

Yes

Is the language acceptable?

Yes

Do you have any ethical concerns with this paper?

No

Have you any concerns about statistical analyses in this paper?

No

Recommendation?

Accept with minor revision (please list in comments)

Comments to the Author(s)

This is a nice paper giving a rarely done soundscape analysis of a particular study region - the details are obviously pertinent to that region, which is quite specific, but the notion of how one does a soundscape analysis, what it is good for, and what it can potentially show, is general and therefore worth publishing in a general journal like this. The analyses are not statistical, although they are clearly highly quantitative. I am mindful that I am reviewing a revised manuscript, and have tried to do so in the context of how the authors have responded to the original comments, rather than as a new submission per se. It's clear to me that the authors have responded carefully and fully to the original reviewer reports, and I don't think it's fair to ask for further major revisions in the light of that. If it were coming to me as an original submission I might have asked the authors to reflect on how they might provide statistical support for some of the observations they make from the quantitative patterns, but I don't think it's fair in this case - there's nothing wrong with what they did, and so perhaps the authors can take my comment here as something to reflect on when planning future work of this kind. Other than that I have only minor comments.

I am kind of puzzled by the insistence of one reviewer to use the obscure term 'decidecade' instead of the widely understood and used 1/3 octave - the bands referred to are clearly using the base 2 octave as they are the standard ISO / Acoustical Society bands, so it's not clear that this is correct in any case. I realise this is perhaps frustratingly contradictory for the authors so I would leave it up to them but to my mind the previous reviewer's comment is unnecessary and impedes understanding/communication. It would particularly eliminate the confusion in P20 L5-8.

P1 L20 'contributors' is unclear

L23 strike 'and'

P18 L25 additional to what? suggest removing

P19 L38 'periods of raised PSD' is a bit unclear - can the authors specify was this in a frequency band or broadband, and what was used to identify 'raised'? Some quantitative idea would help (e.g. over a certain level, or percentile of the data)

P19 L40 would be good practise to just mention that spectrogram parameters were chosen to best display the species vocalisations here?

Is Fig 7 really needed given Fig 6? In general the authors could consider cutting down the number of figures

Decision letter (RSOS-201503.R0)

Dear Miss Warren

On behalf of the Editors, we are pleased to inform you that your Manuscript RSOS-201503 "Marine soundscape variation reveals insights into baleen whales and their environment: a case study in central New Zealand" has been accepted for publication in Royal Society Open Science subject to minor revision in accordance with the referees' reports. Please find the referees' comments along with any feedback from the Editors below my signature.

Please submit your revised manuscript and required files (see below) no later than 7 days from today's (ie 03-Feb-2021) date. Note: the ScholarOne system will 'lock' if submission of the revision is attempted 7 or more days after the deadline. If you do not think you will be able to meet this deadline please contact the editorial office immediately.

on behalf of Prof Pietro Cicuta (Subject Editor)
openscience@royalsociety.org

Associate Editor Comments to Author:
Please accept our apologies for the unusual delay in completing review, but we hope the reviewers' comments are helpful in revising your paper.

Reviewer comments to Author:
Reviewer: 2

Comments to the Author(s)
I appreciated the new version of the manuscript and the approach that the authors took. The authors now frame the study to describe the acoustic presence of baleen whales and changes in the soundscape that they may perceive, which includes an in-depth look at sources of variation. I

believe that the authors have addressed all my previous comments from the first review. I only have a few minor comments, which I detail below.

Page 3

Line 1: "varied" I would recommend using the present tense here (vary) as you have not introduced the results yet.

Line 16: "with the hydrophones mounted to the AMARs." This statement is not necessary, correct? You already stated that the AMARs were moored on vertical line moorings. I am assuming that the AMARs already incorporate hydrophones?

Line 30: I do not think that you need a comma before "and per month" nor before "at each station."

Line 39: "energy" is not the appropriate term here. I recommend replacing energy with levels or inputs.

Page 4

Line 2: This sentence would benefit from the addition of a citation or two.

Lines 5-7: Decidecades are exactly the same as 1/3 octave bands assuming that they are base ten bands, if I am not mistaken. These two sentences open the door for confusing the reader. I would recommend not even mentioning 1/3 octave bands here and just refer to the measurements as decidecade bands as they are understood by the community and the preferred term by Reviewer 1 in the previous review.

Page 5

Lines 33-35: Given the bandwidth of the biological sound, it is likely that the rise in PSD levels might originate from more than one source. Consider adding in this possibility. Additionally in regards to Figure 8, the duration of daily increases in the 1000 Hz decidecade is much longer than one might expect from a single species if these daily increases are definitely biologic in origin, especially if it was produced by a fish species. However, it could be related to feeding sounds of mixed-fish-species origin. Another possibility would be from invertebrates I suppose. In any event, it would be good to include that it might originate from multiple sources.

Page 7

Line 10: Any idea why the high tide at Wairarapa would yield the loudest median sound levels below 100 Hz? Could another source of sound or pressure contribute to this result?

Figure 4: In response to my suggestion for including .wav files as supplementary material for this figure, the authors stated that "We do not believe this is necessary - similar sounds can easily be found online." I would suggest that they reconsider as I cannot find an example of a pygmy blue whale online, or at least easily.

Figure 6: It is difficult to see they components of the LTSAs as presented. Would it be possible to have two figures; one for the first deployment and one for the second deployment? Alternatively, could this figure be presented in "landscape" format. I would like to be able to see the details a bit better. For example, the biological chorus is virtually undetectable in the LTSAs. Maybe this can be addressed in the proofing stage.

Reviewer: 3

Comments to the Author(s)

This is a nice paper giving a rarely done soundscape analysis of a particular study region - the details are obviously pertinent to that region, which is quite specific, but the notion of how one does a soundscape analysis, what it is good for, and what it can potentially show, is general and therefore worth publishing in a general journal like this. The analyses are not statistical, although

they are clearly highly quantitative. I am mindful that I am reviewing a revised manuscript, and have tried to do so in the context of how the authors have responded to the original comments, rather than as a new submission per se. It's clear to me that the authors have responded carefully and fully to the original reviewer reports, and I don't think it's fair to ask for further major revisions in the light of that. If it were coming to me as an original submission I might have asked the authors to reflect on how they might provide statistical support for some of the observations they make from the quantitative patterns, but I don't think it's fair in this case - there's nothing wrong with what they did, and so perhaps the authors can take my comment here as something to reflect on when planning future work of this kind. Other than that I have only minor comments.

I am kind of puzzled by the insistence of one reviewer to use the obscure term 'decidecade' instead of the widely understood and used 1/3 octave - the bands referred to are clearly using the base 2 octave as they are the standard ISO / Acoustical Society bands, so it's not clear that this is correct in any case. I realise this is perhaps frustratingly contradictory for the authors so I would leave it up to them but to my mind the previous reviewer's comment is unnecessary and impedes understanding/communication. It would particularly eliminate the confusion in P20 L5-8.

P1 L20 'contributors' is unclear

L23 strike 'and'

P18 L25 additional to what? suggest removing

P19 L38 'periods of raised PSD' is a bit unclear - can the authors specify was this in a frequency band or broadband, and what was used to identify 'raised'? Some quantitative idea would help (e.g. over a certain level, or percentile of the data)

P19 L40 would be good practise to just mention that spectrogram parameters were chosen to best display the species vocalisations here?

Is Fig 7 really needed given Fig 6? In general the authors could consider cutting down the number of figures

===PREPARING YOUR MANUSCRIPT===

While not essential, it will speed up the preparation of your manuscript proof if you format your references/bibliography in Vancouver style (please see

<https://royalsociety.org/journals/authors/author-guidelines/#formatting>). You should include DOIs for as many of the references as possible.

===PREPARING YOUR REVISION IN SCHOLARONE===

<https://royalsociety.org/journals/authors/author-guidelines/#data>. You should ensure that you cite the dataset in your reference list. If you have deposited data etc in the Dryad repository,

please only include the 'For publication' link at this stage. You should remove the 'For review' link.

Author's Response to Decision Letter for (RSOS-201503.R0)

See Appendix D.

Decision letter (RSOS-201503.R1)

Dear Miss Warren,

It is a pleasure to accept your manuscript entitled "Marine soundscape variation reveals insights into baleen whales and their environment: a case study in central New Zealand" in its current form for publication in Royal Society Open Science.

Please see the Royal Society Publishing guidance on how you may share your accepted author manuscript at <https://royalsociety.org/journals/ethics-policies/media-embargo/>. After

publication, some additional ways to effectively promote your article can also be found here <https://royalsociety.org/blog/2020/07/promoting-your-latest-paper-and-tracking-your-results/>.

on behalf of Prof Pietro Cicuta (Subject Editor)
openscience@royalsociety.org

Appendix A**ROYAL SOCIETY
OPEN SCIENCE****Marine soundscape variation in an oceanographically
diverse region, central New Zealand**

Journal:	Royal Society Open Science
Manuscript ID	RSOS-200117
Article Type:	Research
Date Submitted by the Author:	21-Jan-2020
Complete List of Authors:	Warren, Victoria; Institute of Marine Science, Leigh Marine Laboratory; National Institute of Water and Atmospheric Research Wellington, McPherson, Craig; JASCO Applied Sciences (Australia) Pty Ltd Giorli, Giacomo; National Institute of Water and Atmospheric Research Wellington Goetz, Kim; NOAA, National Oceanic and Atmospheric Administration, National Marine Fisheries Service Radford , Craig; University of Auckland, Marine Science
Subject:	Acoustics < PHYSICS, Oceanography < EARTH SCIENCES, Meteorology < EARTH SCIENCES
Keywords:	passive acoustic monitoring, oceanography, bioacoustics, meteorology, soundscape
Subject Category:	Physics and Biophysics

Author-supplied statements

Relevant information will appear here if provided.

Ethics

Does your article include research that required ethical approval or permits?:

This article does not present research with ethical considerations

Statement (if applicable):

CUST_IF_YES_ETHICS :No data available.

Data

It is a condition of publication that data, code and materials supporting your paper are made publicly available. Does your paper present new data?:

Yes

Statement (if applicable):

Open access data (matrices of per-file 1 Hz PSD, for each recording location) available from Dryad, DOI: <https://doi.org/10.5061/dryad.vmcvdncpj>

The private data review URL is:

<https://datadryad.org/stash/share/uRglAdTX2mhoqeuvAhKkzBajGdSluSlqDCErAnhlowk>

Conflict of interest

I/We declare we have no competing interests

Statement (if applicable):

CUST_STATE_CONFLICT :No data available.

Authors' contributions

This paper has multiple authors and our individual contributions were as below

Statement (if applicable):

VEW conducted all analyses and wrote the manuscript. All authors contributed to study design, interpreted data and critically revised the manuscript. KTG, GG and VEW assisted with data collection. All authors gave final approval for publication.

Marine soundscape variation in an oceanographically diverse region, central New Zealand

Victoria E. Warren^{1,2,*}, Craig McPherson³, Giacomo Giorli², Kimberly T. Goetz⁴, Craig A. Radford¹

1. Institute of Marine Science, Leigh Marine Laboratory, University of Auckland, 160 Goat Island Road, Leigh 0985, New Zealand

2. National Institute of Water and Atmospheric Research, 301 Evans Bay Parade, Hataitai, Wellington 6021, New Zealand

3. JASCO Applied Sciences (Australia) Pty Ltd, 14 Hook Street, Unit 1, Capalaba QLD 4157, Australia

4. National Oceanic and Atmospheric Administration, National Marine Fisheries Service, Alaska Fisheries Science Center, National Marine Mammal Laboratory, 7600 Sand Point Way NE, Seattle, Washington 98115, USA

Keywords: passive acoustic monitoring, oceanography, bioacoustics, weather, soundscape

1. Summary

Despite relatively easy access, high anthropogenic use and a richness of biodiversity, the marine soundscape of central New Zealand has not been quantified. To address this, passive acoustic monitoring (PAM) was conducted at four recording locations around central New Zealand during 2016 and 2017. Median power spectral density was highest below 100 Hz (75 – 97 dB re $1\mu\text{Pa}^2/\text{Hz}$), declining by 3.6 - 4.3 dB re $1\mu\text{Pa}^2/\text{Hz}$ per kHz above 100 Hz. Sound levels were positively correlated with wind speed, wave height, rainfall and tidal movement. Earthquakes increased $1/3$ -octave-bands centered at 63 Hz and below by up to 30 dB re $1\mu\text{Pa}^2/\text{Hz}$. Baleen whales contributed sound year-round, or seasonally, at specific frequencies. Fish chorusing dominated 600 – 1100 Hz, prior to dusk during summer. Two non-concurrent seismic surveys raised $1/3$ -octave-bands centered at 250 Hz and below by up to 20 dB re $1\mu\text{Pa}^2/\text{Hz}$. Vessel noise was ubiquitous, and acoustically concentrated within Cook Strait. Above 70 Hz, broadband sound levels were highest in Cook Strait by approximately 5 dB re $1\mu\text{Pa}^2/\text{Hz}$ due to echoic, shallow water and concentrated contributors. The results demonstrate that PAM is an effective, long-term, holistic monitoring option pertinent to a wide array of ecological and environmental functions.

*Author for correspondence (vwar775@aucklanduni.ac.nz).

2. Introduction

The ambient acoustic environment, or soundscape, consists of cumulative contributions from abiotic (geophonic), biotic (biophonic) and man-made (anthrophonic) sound sources (1). Variation in soundscape characteristics over time and space can act as proxies for geographical, biological and anthropogenic events occurring within an environment, and soundscape analysis provides a non-invasive method to study whole ecosystems.

In the marine environment, geophonic elements of the soundscape commonly correlate with oceanographic conditions. Increased sea state and wind speed lead to higher sound intensities across frequencies ranging from 500 Hz to 30 kHz, *via* sound produced by breaking waves, cavitation, surface flow noise and pressure changes (2, 3). Rainfall elevates sound levels in the 1 – 15 kHz frequency range, *via* surface impacts and bubble entrainment (4-6). The specific frequency band affected by rainfall depends on rain strength and droplet size (7). Flow noise, caused by water movement, is generated as a byproduct of pressure eddies and vortices moving past an acoustic receiver which can be detected as sound; it is not considered to be part of a marine soundscape (8, 9). However, flow noise intensity can be indicative of current strength (10). Abiotic acoustic contributions are often unpredictable or irregular (9), for example, significant low frequency acoustic energy can be contributed to marine soundscapes by earthquakes and sea ice movement (11, 12). On the other hand, biophonic contributions often feature seasonal and diel activity patterns (e.g. 13).

Biophonic contributors to marine soundscapes include tonal and pulsive vocalisations produced by marine mammals, birds, fish and invertebrates to communicate, orientate and feed. Such sounds often feature specific time/frequency characteristics, which can be used to identify the species responsible. Seasonal trends in biophonic sounds can act as proxies for behaviours, such as migration (e.g. 14). Other sounds of animal origin that contribute to marine soundscapes include byproducts of behavior, such as the snaps produced by snapping shrimp (*Alpheus heterochaelis*) when catching prey (15). When a large number of sound-producing animals are present, both voluntary and involuntary sounds can combine to generate choruses where individual sounds cannot be distinguished (e.g. 14). In northern New Zealand, feeding sea urchins are known to produce a resonant chorus which can potentially act as a proxy for reef health and diversity (16-18). Diel trends in choruses can be indicative of time-specific behaviours, such as crepuscular or nocturnal fish activity (19-22). Biophonic contributors to the soundscape have evolved to produce sounds with minimal overlap, either in frequency, time or space (acoustic niche hypothesis (23)).

Anthropogenic sounds are relatively recent additions to soundscapes; unlike biophonic contributors, often overlap in frequency, space or time (24). As such, man-made sound sources can affect organisms living in the marine environment (25-28). Anthropogenic contributors to global ocean noise include vessel traffic (commercial and recreational) at frequencies in the decade 50 to 500 Hz (9), and marine industries such as construction, oil and gas exploration and extraction (29). The presence and intensity of anthropogenic sound varies over time and space (9).

Passive acoustic monitoring (PAM) in marine environments provides a non-invasive means to study natural and abiotic elements of an environment which can be analysed over a variety of temporal and spatial scales.

With ever-growing concerns about the impact of anthropophony, it is essential to understand baseline acoustic
conditions. Consistent monitoring of underwater sound improves the foundation of knowledge available to
make informed management decisions. This is especially important in areas with growing anthropogenic
activity that overlaps with rich biodiversity and diverse ecosystems.

New Zealand lies on the fault line between the east-bound Indo-Australian plate and the west-bound Pacific
plate and therefore experiences frequent seismic activity (30). The two plates encounter each other most notably
around central New Zealand and the natural tectonic activity in this area has led to a diverse array of marine
environments including deep-water canyons, drowned river valleys and large shallow embayments (30). These
environments are rich in marine life, hosting important fish spawning grounds (31), seasonally migrating whale
species (32), and pygmy blue whales (*Balaenoptera musculus brevicauda*) that are present year-round (33). At the
centre of this region, Cook Strait connects the Tasman Sea with the Pacific Ocean and separates the North and
South Islands of New Zealand. The narrow Strait is notorious for poor weather and rough sea conditions (34-
37). Nonetheless, as the main central waterway, Cook Strait is an important commercial shipping area and also
hosts fourteen scheduled inter-island ferry crossings per day. Furthermore, the interaction of the tectonic plates
in central New Zealand generated a sedimentary basin in the South Taranaki Bight (STB). This location currently
supports all of New Zealand's active oil and gas extraction industry (38). As a consequence of high biodiversity,
complex oceanography and anthropogenic activities, a diverse array of sound types are emitted into the marine
environment in this region.

Tidal flows in Cook Strait exhibit the dynamics of a standing wave due to antithetical tidal cycles at the east
and west entrances, resulting in high levels of entropy (30, 35), further adding to turbulent sound generated by
weather and wave activity in the channel. Bathymetry, geology and sedimentology have direct effects on
underwater sound (39), and Cook Strait features a coarse pebble seabed (30, 40), likely to propagate and resonate
sound (41). In contrast, STB has a sand and mud seabed (30, 40), reducing sound propagation distances (41).
There are a large number of acoustic contributors in the region, but there is a fundamental lack of baseline
information about the marine soundscape of this area. Around the northern coast of New Zealand, the marine
soundscapes of Kaipara Harbour and the Hauraki Gulf have been quantified (18, 22, 42, 43). However, the
dynamic nature of environments in central New Zealand are quite different to the sheltered, shallow bays of
northern New Zealand. Extrapolating soundscapes from these northern areas is unlikely to be appropriate, as
the strong tidal forces and poor weather that affect the Cook Strait region will not be reflected in marine
soundscapes elsewhere. Moreover, soundscape variation is likely to exist on a small scale between the differing
marine environments within central New Zealand (e.g. 22, 44).

Given the prevalence and concentration of geological, meteorological, biological and increasing
anthropogenic activity around central New Zealand, there has been a growing need to quantify the sound type
of this region. Autonomous acoustic recorders were deployed to quantify the marine soundscapes around
central New Zealand and identify the relative spatial and temporal contributions of geophonic, biophonic, and
anthropophonic sources.

3. Materials and Methods

Data collection

Four Autonomous Multi-channel Acoustic Recorders (AMARs, JASCO Applied Sciences) were deployed from early June to mid-December 2016 around central New Zealand: in the South Taranaki Bight; in the narrows of Cook Strait; north-east of Kaikoura; and off the east coast of Wairarapa (henceforth referred to as STB, Cook Strait, Kaikoura and Wairarapa) (Table 1, Figure 1 (45)). Three AMARs were redeployed between late-February and early September 2017: two in the same relative Kaikoura and Wairarapa locations and one in STB, 25.2 km southeast of the first STB deployment location (Table 1, Figure 1). A recorder was not redeployed in Cook Strait in 2017. The AMARs in Cook Strait and STB were bottom-mounted on metal baseplates, with the hydrophone 75 cm off the seafloor in a flow-shield, in water depths less than 300 m. The Kaikoura and Wairarapa ultra-deep AMARs were deployed in water depths exceeding 1000 m and were moored on vertical line moorings, approximately 10 m above the seabed, with the hydrophones mounted to the AMARs and contained in a flow-shield. All AMARs were retrieved *via* the use of acoustic releases and buoyancy aids.

Acoustic sampling was duty cycled over 900 seconds: 630 seconds at a sampling rate of 16 kHz, 125 seconds at a sampling rate of 250 kHz, 45 seconds of sleep. Only files with 16 kHz sampling rate were considered in this study. Recorder sensitivity was calibrated *via* pistonphone, and all data were corrected by frequency- and recorder-specific sensitivity curves. Sensitivity featured a flat frequency response of -164.8 dB re 1 V/ μ Pa (\pm 0.5 dB re 1 V/ μ Pa) between 100-8000 Hz, with a low frequency roll-off below 51.2 Hz. Analyses focused on frequency values between 10 - 7000 Hz to omit high energy infrasound, and the effect of Nyquist roll-off close to 8000 Hz.

Acoustic Processing

Acoustic analyses were conducted in MATLAB (46). Power spectral density (PSD) was calculated with 1 Hz resolution for each 630 s, 16 kHz file using a Fast Fourier Transform with window length of 16,000 samples (Hanning window) and 80% overlap. PSD outputs were utilised to compute spectral probability density (SPD) (47) with PSD bin widths of 1 dB re $1\mu\text{Pa}^2$ / Hz. SPDs were calculated per deployment, and per month, at each station. Exceedance percentiles of 1%, 5%, 50% (median), 95% and 99% were also calculated. In order to investigate temporal trends and broadband sounds, long-term spectral averages (LTSAs) were generated by concatenating the 1 Hz resolution PSD averages for all data files and displaying as scaled colours. Daily 1/3-octave-band medians centred at 12.5, 25, 63, 125, 250, 1000 and 4000 Hz were calculated from the per-file PSD outputs per station and deployment. Seven-day moving averages were subsequently applied. Specific features within the data were visualised as spectrograms in PAMlab-Lite (48).

Analysis of soundscape contributors

I. Geophonic

Weather data were obtained from the CliFlo database (<https://cliflo-niwa.niwa.co.nz/>) from five weather stations around central New Zealand (**Figure 1**). Hourly rainfall and average hourly wind speed were obtained for both acoustic deployment periods. Twenty-four-point (one day) moving averages were placed over the hourly rain and wind data to clarify periods of persistent weather. Within the averaged data, the total ranges of wind and rain for each station were split into three and classified as 'low', 'medium' or 'high' (**Tables 1 and 3**). Rain was also classified as 'no rain' if the average value was 0 mm (**Table 2**). Rain and wind scenarios were drawn from these classifications when at least three of the weather stations observed the same scenario (e.g. 'low wind' or 'mid rain'), implying that the whole region was experiencing consistent weather. The median PSDs of the files recorded closest in time to each wind or rain scenario were plotted to examine the impact of varying wind speed and rainfall on the marine soundscape at each station.

Wave statistics were obtained from the NZWAVE-NZCSM wave forecast, an implementation of the third generation wave model Wavewatch III® Version 4.18 (45), part of NIWA's operational environmental forecasting system. Significant wave height (SWH, mean height of the highest 33% of waves) output from NZWAVE-NZCSM at 30 minute intervals within the subdomain 173.0 to 176.5, 43.0°S to 39.5°S were extracted and accumulated for the months June 2016 to September 2017, taking the first 6 hours of successive forecasts. As per the weather data, 'low', 'mid' and 'high' SWH scenarios were extracted for each recording location (**Table 4**), and the median PSDs of corresponding acoustic files were compared to examine the influence of wave activity on the soundscape.

Modelled sea level (tidal) data from the two deployment periods were obtained for acoustic recording locations from NIWA's online tide forecaster (<https://www.niwa.co.nz/services/online-services/tide-forecaster>). Timestamps of high tide, low tide, mid-flood tide (low tide + 3 hours) and mid-ebb tide (low tide - 3 hours) were extracted and the median PSDs of the acoustic data at each station were computed and visualised per tidal state.

The timings and magnitudes of all earthquakes (exceeding magnitude 0.5) that occurred during the two deployment periods, between 40 - 43°S, 173 - 177°E (**Figure 1**), were extracted from the Geonet database (<https://www.geonet.org.nz/>) and used to quantify the effect of natural seismic activity on the soundscape of the region.

II. Biophonic

Data from the four recording locations were reviewed manually to investigate the acoustic presence of marine mammals. The data were sub-sampled: one random 630 s file was extracted from every two hours of recording and displayed as a spectrogram with 0.244 Hz frequency resolution, 2 s time window, 0.5 s time step, Hamming window. Data were viewed between 0 - 100 Hz, over the total file duration. All cetacean vocalisations evident within these limits were annotated. Cetacean species were identified by comparing the spectrograms with those found for the same or similar species elsewhere (49-55). PSD peaks at frequencies below 100 Hz were then compared with annotated marine mammal vocalisations. For PSD peaks at frequencies above 100 Hz, data files corresponding to periods of raised PSD were examined to determine the origin of the increased energy.

Given that biophonic contributions can feature diel cycles, specific times (e.g. dusk) were obtained for the
 location of Wellington (<https://www.gaisma.com/en/location/wellington.html>).
III. Anthrophonic

Information regarding seismic survey activity conducted around central New Zealand during recording
 periods was derived from Marine Mammal Impact Assessments (MMIA) available from the New Zealand
 Department of Conservation website ([https://www.doc.govt.nz/our-work/seismic-surveys-code-of-](https://www.doc.govt.nz/our-work/seismic-surveys-code-of-conduct/marine-mammal-impact-assessments/)
 [conduct/marine-mammal-impact-assessments/](https://www.doc.govt.nz/our-work/seismic-surveys-code-of-conduct/marine-mammal-impact-assessments/)). Operational area coordinates were obtained from the MMIA's,
 or Land Information New Zealand 'Notices to Mariners' ([https://www.linz.govt.nz/sea/maritime-safety/notices-](https://www.linz.govt.nz/sea/maritime-safety/notices-mariners)
 [mariners](https://www.linz.govt.nz/sea/maritime-safety/notices-mariners)). Seismic survey pulse sequences were detected in the data using an automated detector (full details
 provided by Martin, 2013 (56)). Detections were made on a daily scale and compared against the known periods
 of seismic survey activity from the above sources.

Following recommendations in the EU Marine Strategy Framework Directive (2008/56/EC),  1/3-octave-
 bands centred at 63 Hz and 125 Hz were examined to investigate the contribution of shipping to the soundscape
 of the region (57). Such directives are not available for the South Pacific, hence the application of European
 guidelines. Vessel contributions were also illustrated by the overview analyses.

4. Results

At all recording locations, median PSDs were highest below 100 Hz, ranging from 75 – 97 dB re $1\mu\text{Pa}^2 / \text{Hz}$
 (Figure 2). Above 100 Hz, median PSD values decreased by 3.6 - 4.3 dB re $1\mu\text{Pa}^2 / \text{Hz}$ per kHz, to values of 49 –
 55 dB re $1\mu\text{Pa}^2 / \text{Hz}$ at 7000 Hz. At Cook Strait, PSD at frequencies above 70 Hz exceeded other stations by
 approximately 5 dB re $1\mu\text{Pa}^2 / \text{Hz}$ while following the same overall trend (Figure 2). PSD dipped at 1500 and
 2000 Hz consistently during both deployments at Kaikoura and Wairarapa (Figure 2).  This was an artefact
 resulting from destructive interference due to the separation distance between the AMAR and hydrophone and
 the spherical glass floats deployed at these deep-water locations and was not a true feature of the soundscape.

Above 300 Hz, the probability of a given PSD value showed central tendencies for each 1 Hz value, at all
 recording locations (Figure 3). Below 100 Hz, SPD varied within respective PSD ranges. At STB and Cook Strait,
 this variation occurred across PSDs of 60 to 120 dB re $1\mu\text{Pa}^2 / \text{Hz}$ (Figure 3A and 3B), while at Wairarapa and
 the first deployment at Kaikoura, the data became bimodal,  centered around 80 dB re $1\mu\text{Pa}^2 / \text{Hz}$ and 100 dB re
 $1\mu\text{Pa}^2 / \text{Hz}$ (Figure 3C and 3D). This implied that a minority of files contained loud, low frequency sound
 energy. During the first deployment, some of the data centred around 100 dB re $1\mu\text{Pa}^2 / \text{Hz}$ were a result of a
 magnitude 7.8 earthquake on 14 November (NZDT) and associated aftershocks, which contributed significantly
 to the marine soundscape. At Kaikoura, Wairarapa and Cook Strait, this earthquake contribution raised the 1/3-
 octave-band averages centred at 12.5, 25 and 63 Hz (Figure 4C, 4D, 4E and 4F). At Kaikoura, the 1/3-octave-
 band centred at 12.5 Hz was raised from approximately 75 dB re $1\mu\text{Pa}^2 / \text{Hz}$ to 108 dB re $1\mu\text{Pa}^2 / \text{Hz}$ by the initial
 quakes (Figure 4D and 4F).

In addition to earthquake contribution, the Schlumberger Pegasus Basin 3D seismic survey off the coast of
 Wairarapa (Figure 1) also resulted in increased PSD at low frequencies, further contributing to the bimodal

PSDs below 100 Hz during both deployment periods at Wairarapa (**Figure 3D**). The seismic survey was
conducted from mid-November 2016 until early June 2017 (across both acoustic deployment periods), and was
most evident in the soundscape at Wairarapa between 100 and 300 Hz. The seismic survey influenced the 1/3-
octave-bands centred at 250 Hz and below at Wairarapa, throughout its duration, with PSD increases of up to
20 dB re $1\mu\text{Pa}^2 / \text{Hz}$ (**Figure 4A and 4E**). During the second deployment period, when earthquake contribution
was reduced (**Figure 4F**), the 12.5 and 25 Hz centred 1/3-octave-bands at Wairarapa were less influenced (**Figure**
**4E**). The seismic survey also influenced 1/3-octave-bands centred at 125 Hz and below at Kaikoura, to a lesser
extent than at Wairarapa (**Figure 4A and 4D**).

Neither the earthquakes nor the Schlumberger Pegasus Basin 3D seismic survey affected 1/3-octave-bands
at STB (**Figure 4B**). Another seismic survey, the PGS Taranaki South 3D seismic survey, was conducted from
mid-October 2016 until early December 2016, closest to the STB recording location (**Figures 1 and 4A**). PSD at
low frequencies (< 100 Hz) was highly variable at STB during the first deployment (**Figures 3A and 4B**), and
although seismic pulses were detected in the data (**Figure 4A**), the contribution of the seismic survey was not
obvious (**Figure 4B**). At STB, the 12.5, 25 and 63 Hz centred bands were quieter during the second deployment
than the first deployment, likely due to the relocation of the recording equipment within the STB (**Figure 4B**).

When viewed in the time/frequency domain, the contribution of Schlumberger Pegasus Basin 3D seismic
survey airgun impulses to the soundscape was notable between 0 – 250 Hz at Wairarapa between mid-
November 2016 until mid-June 2017 (**Figure 5D**), and to a lesser extent at Kaikoura (**Figure 5C**). Throughout the
seismic period, sound energy in the frequency range was not constant due to survey operations, such as line
turns, shut-downs and varying distance from the acoustic recorders. The magnitude 7.8 earthquake on 14
November (NZDT) was evident in the LTSA of all stations (**Figure 5**), and the Schlumberger Pegasus Basin 3D
seismic survey occurred concurrently with aftershocks at Kaikoura and Wairarapa (**Figure 5C and 5D**).

Geophony

I.

Weather

Across the region, the strongest winds and heaviest rainfall were recorded at Akitio and Wellington weather
stations (**Figure 1, Tables 2 and 3**). Preliminary analyses of weather data indicated that heavy rain often
followed heavy wind, rather than occurring simultaneously. PSD levels corresponding to periods of 'high'
rainfall were absent at Cook Strait and limited in number ($N = 21$) at the other three recording locations (**Figure**
**6**). Examples of 'mid' rainfall were also few ($N = 15$ or 21). At Cook Strait, Kaikoura and Wairarapa, the 'mid'
rain scenarios included PSD peaks between 600 – 1100 Hz, and it is likely that the small number of files were all
from a similar time when fish chorusing was occurring (see Biophony section). Outside of these limitations,
there was no clear trend in the effect of rainfall on the marine soundscape at Cook Strait (**Figure 6B**). At the
remaining recording locations, 'high' rain generated the greatest sound levels at frequencies above 200 or
300 Hz, while 'no' rain generated the quietest sound levels across the same bandwidth (**Figure 6A, 6C and 6D**).

At STB, the positive correlation of rainfall on sound levels was most evident, with PSD levels above 200 Hz
positively correlated with heavier rainfall (high > mid > low > no) (**Figure 6A**).

The influence of wind speed on the soundscape of the area was strongly evident in PSD levels. In general,
'high' wind resulted in the largest PSD values across all frequencies, followed by 'mid' wind, with 'low' wind
generating the lowest PSD scenario (**Figure 7**). At Cook Strait, the sound levels of all three wind scenarios
converged between 30 and 130 Hz (**Figure 7B**), while at Wairarapa, 'mid' and 'low' wind scenarios converged
between 30 and 100 Hz (**Figure 7D**). Overall, these results show that wind speed influences the soundscape
more greatly than rainfall, and across all frequencies, while the addition of rain increases higher frequency
sound levels (above 300 Hz) and is most strongly correlated with sound levels in shallow water.

II. Wave activity

The highest minimum and maximum SWH were noted at Wairarapa (minimum = 0.56 m, maximum = 10.19
12 m), while the lowest minimum and maximum SWH occurred at Cook Strait (minimum = 0.10 m, maximum =
5.43 m) (**Table 4**). Excepting some convergence between 50 and 100 Hz, the median PSD of 'high' wave activity
was the loudest scenario at all recording stations, approximately 3 – 5 dB re $1\mu\text{Pa}^2 / \text{Hz}$ louder than 'mid' wave
activity, which was in turn 3 – 5 dB re $1\mu\text{Pa}^2 / \text{Hz}$ louder than 'low' wave activity, which generated the lowest
average PSD values (**Figure 8**). While the range of SWH varied between stations, the influence of SWH was
consistent at all stations. Some of the convergence at 50 – 100 Hz at Wairarapa was due to the presence of seismic
survey pulses which increased the median value of the 'low' SWH most significantly. However, the 50 – 100 Hz
bandwidth at which the wave scenarios converged was congruent with the convergence displayed by the wind
scenarios, implying correlation between the wind and wave elements.

III. Tidal state

At STB and Cook Strait, median PSD levels below 50 Hz were up to 15 dB re $1\mu\text{Pa}^2 / \text{Hz}$ higher during
flooding and ebbing tidal periods than during high and low tide periods (**Figure 9A and 9B**), while median PSD
was similar across all tidal states at Kaikoura (**Figure 8C**). High tide generated the loudest median sound levels
below 100 Hz at Wairarapa, with PSD up to 15 dB re $1\mu\text{Pa}^2 / \text{Hz}$ higher than during low, flooding or ebbing tide
(**Figure 9D**). The reason for highest PSD levels at high tide at Wairarapa is unknown and warrants further
investigation. Above 100 Hz, median PSD did not differ between tidal scenarios at any station (**Figure 9**).

IV. Natural seismic activity

During the first acoustic deployment, 12,122 earthquakes occurred within the study area (**Figure 1**) with a
maximum magnitude of 7.8 (median = 2.5). A total of 6,508 earthquakes occurred during the second deployment
period, with a maximum magnitude of 5.2 (median = 2.04). Earthquakes in the study region, particularly those
of higher magnitude, were concentrated along the east coast of the South Island. Within a 50 km radius of STB
and Wairarapa, approximately the same number of earthquakes occurred during both deployments (312 and
390 at STB, and 25 and 12 at Wairarapa). However, 3.4 times as many earthquakes occurred within a 50 km
radius of Kaikoura during the first deployment compared to the second deployment (2778 compared to 816).

Earthquakes were evident in the acoustic data as sudden onset, high energy, low frequency (generally less
than 100 Hz) broadband sound lasting a few seconds to a few minutes. Earthquakes occurring outside of the
study region likely also contributed to recorded sound levels. The influence of earthquake activity on the marine

soundscape was most evident at Kaikoura subsequent to the magnitude 7.8 earthquake on 14 November
(NZDT) (**Figure 4**). The epicentre of this large earthquake was closest to the Kaikoura recorder and generated
surface displacements of up to 12 m, and resulted in mud slides and sediment movement across a 10,000 km²
area (58). Both the initial earthquake and associated effects contributed to the soundscape of the area and
resulted in sound levels that briefly exceeded the dynamic range of the AMAR at Kaikoura.

**Biophony**

**I. Cetaceans**

Song from two sub-species of blue whale were recorded in the region and contributed significantly to the
marine soundscape. Pygmy blue whale song contributed sound energy that peaked at 16/17 Hz and 23 Hz
(**Figure 10A**). This song was recorded at all stations but was most apparent in the STB soundscape where it was
present year-round, particularly from February to May 2017 (**Figures 5A and 11A**). Pygmy blue whale song was
evident in median PSD and SPD for both STB deployments (**Figures 2 and 3A**) and raised the 25 Hz-centred
1/3-octave-band during March and April 2017 (**Figure 4B**). Antarctic blue whale (*Balaenoptera musculus*
*intermedia*) song units (also known as 'Z' calls) were recorded seasonally during migration and contributed
energy to the soundscape at 17/18 Hz and 26 Hz (**Figure 10B**). This sound was evident in the Kaikoura
soundscape at the end of June 2016 (**Figure 5D**), and similarly raised the 25 Hz-centred 1/3-octave-band during
this time (**Figure 4D**). Another sound, similar to the first part of a 'Z' call, was also recorded in the region and
may be an example of an 'M' call produced by Antarctic blue whales (59) or 'spot' call of which the origin
remains unknown (60) (**Figure 10C**). Blue whale 'D' calls (downsweeps between 60 and 45 Hz with a duration
of one second (61)) were also present in the data and most abundant at STB (**Figure 10D**).

Possible sei whale (*Balaenoptera borealis*) downsweep calls were noted in the data (**Figure 10E**). These were
very similar to the 'D' calls of blue whales, but with slightly lower maximum frequencies and often in pairs or
triplets (54). In addition, downsweep, back-beat and 89 Hz vocalisations of fin whales (*Balaenoptera physalus*)
(53) were recorded (**Figure 10F**). The song produced by migrating male humpback whales (*Megaptera*
*novaeangliae*) contributed significant energy between 250 and 400 Hz to the soundscape of the region (**Figure**
**10G**). Humpback song had greatest influence on the soundscape during July 2016, at the STB station (**Figure**
**11B**). Southern right whale (*Eubalaena australis*) vocalisations can be difficult to distinguish from humpback
whales due to their similar temporal presence and frequency range (62), but possible southern right whale
upcalls were noted in the data during August 2016 (**Figure 10H**). The 'bio-duck' sound of the Antarctic minke
whale (*Balaenoptera bonaerensis*) (52) was noted within the data (**Figure 10I**) and this is the first documentation
of this pulsive sound in New Zealand waters.

The whistles of unidentified odontocetes were abundant in the data, although some components featured
frequencies above 8 kHz. Furthermore, the 16 kHz data examined here only resolves the low frequency
components of odontocete clicks, with the 250 kHz recorded data (not analysed) included to cover higher

frequency components. Despite this, clicks from unidentified odontocetes were abundant at all recording
locations, and sperm whale (*Physeter macrocephalus*) clicks were evident at the deep-water moorings (63).

II. Fish

A seasonal PSD peak occurred between 600 and 1100 Hz at all stations to varying extents (**Figures 3 and 5**)
and is assumed to be a result of fish chorusing. Higher PSD levels between 600 – 1100 Hz were evident at Cook
Strait between September and December 2016 (**Figure 5B**), at Kaikoura and Wairarapa between November 2016
and April 2017 (**Figure 5C and 5D**) and at STB in every month of recording except May, June and July (austral
winter) (**Figure 5A**). As well as seasonal variation, diel variation was also evident in this soundscape contributor.
During austral summer, the 1000 Hz-centred 1/3-octave-band level peaked prior to dusk (20:00 – 22:00) (**Figure**
**12A**). During austral winter, when the 600-1100 Hz chorus was not present, there was no cyclic diel variation in
the 1000 Hz-centred 1/3-octave-band level at dusk (17:00 – 18:00) or any other time of day (**Figure 12B**). In
addition to the suspected fish chorus, other sounds, likely produced by fish, were noted within the data, but did
not contribute significantly to the overall soundscape (e.g. **Figure 13**).

Anthrophony

I. Seismic surveys

Two seismic surveys occurred in the same region as the acoustic deployments: the PGS Taranaki South 3D
survey (23 October to 2 December 2016); and the Schlumberger Pegasus Basin 3D survey (10 November 2016 to
June 2017) (**Figure 1**). The effect of the seismic surveys on the soundscape (PSD, SPD and LTSA) was previously
discussed. Impulses from both surveys were detected in the data, with close alignment to known periods of
seismic surveying (**Figure 4A**). The PGS Taranaki South 3D seismic survey was only detected in data from STB.
The Schlumberger Pegasus Basin 3D seismic survey was detected from the 18th November 2016 to the 6th June
2017 in both the Kaikoura and Wairarapa data. **Figure 4A** displays detector results for the Schlumberger survey
for Wairarapa only. Detections from Kaikoura were fewer (maximum of 1400 detections during a day, compared
to a maximum of 7500 during a day for Wairarapa) but covered the same period, except no detections were
made during March 2017. In addition to true seismic impulses, the seismic detector made some false detections
outside of survey periods. Probable fish calls (**Figure 12**) and Antarctic minke whale 'bio-duck' pulses caused
false positive detections at STB in early October 2016 and at Kaikoura in August 2016 and July 2017. Six
detections at Cook Strait on 3 August 2016 were due to the presence of an artefact repeatedly knocking on the
hydrophone.

II. Shipping

The sound of shipping was present in the data at all stations as persistent tonal sounds or broadband
cavitation sounds. Ship sound increased in received level and bandwidth when ships passed near the acoustic
recorders, and the Lloyd mirror effect was commonly evident during close ship passes (**Figure 14**). **Figure 14**
also illustrates an example of possible acoustic masking where the sound produced by a passing ship occupied
the frequency bands being utilized by Antarctic blue whales. In LTSAs, ship passes appeared as short-duration,
broadband peaks in intensity. The greatest number of such broadband peaks was evident at Cook Strait, where

they were almost continuous (**Figure 5B**). Persistent frequency-specific tonal sounds associated with vessels were also evident below 100 Hz at this recording location (**Figures 3B and 5B**).

At Cook Strait, the 63 and 125 Hz centred 1/3-octave-bands, known to be proportional to vessel sound, were louder than the other 1/3-octave-bands until earthquake contribution occurred in November 2016, and did not follow the trend of the other stations where an increase in centre frequency generally correlated with a decrease in PSD (**Figure 4C**). This implied that shipping sound was most concentrated in Cook Strait, resulting from the bottleneck nature of the Strait for vessel traffic, multiple daily ferry crossings, and the propagation environment of the channel which has a coarse pebble substrate that resonated sound. Outside of the Schlumberger Pegasus Basin 3D seismic survey, the 63 and 125 Hz-centred 1/3-octave-bands were lowest at Wairarapa, approximately 5 - 10 dB re $1\mu\text{Pa}^2 / \text{Hz}$ quieter than at Cook Strait (**Figure 4E**).

5. Discussion

Soundscape quantification is useful for investigating ecosystem functioning in a non-invasive manner, while also revealing the origins and quantities of anthropogenic sound and how such contributors could affect natural processes (1). Here, the marine soundscapes at four seabed recording locations in an oceanographically diverse area around central New Zealand were characterised during 2016 and 2017. The four recording locations differed in depth, sediment type, exposure and oceanography. Values below 50 Hz varied by up to 30 dB re $1\mu\text{Pa}^2 / \text{Hz}$ within and between stations and deployments (**Figure 3**). Applying a simple spherical-spreading transmission loss model implied that transmission loss between the surface and the STB recorder was 40.9 dB (average 111 m depth), 48.0 dB at Cook Strait (252 m depth), 62.0 dB at Kaikoura (average 1252 m depth) and 63.4 dB at Wairarapa (average 1483 m depth). Surface sound therefore experienced an extra 22.5 dB of transmission loss *en route* to the Wairarapa recorder, compared to the STB recorder, helping explain why both Kaikoura and Wairarapa soundscapes were generally quieter than STB and Cook Strait (**Figure 2**). Average PSD values above 70 Hz were greatest at the seabed in Cook Strait, by approximately 5 dB re $1\mu\text{Pa}^2 / \text{Hz}$, but were otherwise similar across both deployments and all stations. The combination of soundscape contributors recorded around central New Zealand emphasises the complexity and diversity of the region.

Geophony

Wind and wave activity are largely perceived to affect marine soundscapes at frequencies above 500 Hz (2, 3), however soundscapes exposed to persistent wind noise and large swells have demonstrated the influence of these contributors at frequencies as low as 10 - 100 Hz (64, 65) and with variation as large as 30 dB re $1\mu\text{Pa}^2/\text{Hz}$ (65). Here, the sound levels at all four recording locations were positively correlated with wave height and wind speed across the full range of frequencies up to 7 kHz, with variation exceeding 20 dB re $1\mu\text{Pa}^2/\text{Hz}$ (**Figures 6 and 7**). Therefore, the concentration of strong winds and persistent wave action around central New Zealand (30, 35) was reflected in the soundscape of the region as high ambient sound levels. In the tropical and sub-tropical Pacific, where winds were not noted to be strong, sound levels of less than 60 dB re $1\mu\text{Pa}^2 / \text{Hz}$ were

1 recorded at depths of 600 – 1100 m for wind and wave dominated frequencies above 200 Hz (66). Conversely,
in the present study, median PSD levels dropped below 60 dB re $1\mu\text{Pa}^2 / \text{Hz}$ only at frequencies above 1000 Hz,
even at the deep water (> 1km) recorders at Kaikoura and Wairarapa. As anthroponic and biophonic
contributions generally featured peak intensity at lower frequencies, a large proportion of the sound energy
above 1000 Hz was due to natural processes.

While this study reports the influence of wind speed and significant wave height (SWH) on the soundscape
separately, the two elements are correlated as wind can generate waves and bubble entrainment. Across all
weather stations and recording locations, a basic comparison revealed that months with high average wind
speed were coincident with high average SWH, and low average wind speeds were coincident with low average
SWHs. These correlations imply that both wind speed and SWH effects contribute to the soundscape
simultaneously, and this was emphasised by PSD convergence at similar frequencies for both parameters. The
two elements were considered separately here because swell can also be generated by non-local sources such as
strong winds offshore.

Sound levels above 200 Hz correlated positively with rainfall at all recording locations except Cook Strait.
The lack of correlation between rainfall and PSD levels at Cook Strait may be a result of wind and wave activity
dominating the soundscape in the narrow channel, where the surface area for rainfall is limited compared to
the open-water scenarios at other recording locations. Sound from surface weather conditions can propagate to
the sea floor (e.g. 4, 22, 67). Here, the shallow-water STB data demonstrated this phenomenon, but wind, wave
and rainfall conditions also affected the soundscapes recorded at the deep-water instruments. Erbe *et al.* (68)
also noted the influence of weather in deep water, with wind influencing the marine soundscape recorded at
the seabed of the Perth Canyon, western Australia, between 200 – 3000 Hz, and heavy rain affecting frequencies
as low as 500 Hz.

Tidal movement increased sound levels below 100 Hz at all recording locations except Kaikoura, as has
been reported at other locations (69). During the first deployment, sound levels at very low frequencies (~10 Hz)
at STB were high and related to periods of high wind, rain and wave activity, as well as ebbing tidal movements.
Both STB moorings were made on sand/mud seabed where sediment resuspension and consequent noise was
expected during tidal movement. The first deployment location at STB was highly affected by this pseudo noise
at low frequencies, but the second deployment did not record these high values at low frequencies, possibly
indicating subtle differences in local sediment composition.

Earthquake activity was a significant source of geophonic sound in central New Zealand, contributing low
frequency ambient noise and rare periods of high intensity sound. Earthquakes affected the marine soundscape
off the north-eastern South Island the most, due to its location on an active tectonic fault. The 12.5 Hz 1/3-octave-
band recorded at Kaikoura increased by more than 30 dB re $1\mu\text{Pa}^2 / \text{Hz}$ during the period of high earthquake
activity during November and December 2016. The influence of earthquake sound was enough to contribute
towards bimodal SPDs below 100 Hz at both Kaikoura and Wairarapa. Previously, bimodal SPDs below 100 Hz
have been demonstrated in Antarctic waters, where summer and winter soundscapes are markedly different
due to the presence or absence of ice cover and associated decoupling of wind effects on the sea surface (70).

In a New Zealand context, the soundscape of STB may be expected to be comparable to that of the Hauraki
Gulf, described by Putland *et al.* (22), as both are large shallow environments with sand and mud sediments.
However, while PSD levels between 50 – 200 Hz at Horn Rock in the Hauraki Gulf (18 m depth) ranged from
60 – 80 dB re $1\mu\text{Pa}^2 / \text{Hz}$ (22), the median PSD value between 50 – 200 Hz at STB ranged from 73 – 84 dB re $1\mu\text{Pa}^2$
5 / Hz, which was somewhat higher and in spite of additional transmission loss occurring at the deeper STB
recorder. Moreover, in the 50 – 200 Hz range, Cook Strait PSD values were greatest, from 79 – 87 dB re $1\mu\text{Pa}^2 /$
7 Hz, markedly higher than the Hauraki Gulf, and this was expected due to the narrow channel and busy shipping
lane within Cook Strait. The coarse pebble seabed in the Strait likely creates a propagation environment that
emphasises the acoustic contribution of certain contributors, such as ship sound (41). Similarly, the resonant
environment of the Strait caused echoing of humpback whale song units; echoes were not evident in the data
recorded at any other location.

13 **Biophony**

Central New Zealand is known to host an array of marine mammal species, which was confirmed by the
marine soundscape of the region. At the largest temporal scale, pygmy blue whales influenced the soundscape
the most, with ubiquitous contributions at 16/17 and 23 Hz at the STB recording location. Torres (71), described
the STB as a 'hotspot' for pygmy blue whales, but song from this sub-species was also recorded at the other
three locations, suggesting that the whole region provides important habitat. Pygmy blue whale distribution in
New Zealand waters is thought to be driven by water temperature and consequent prey availability (72, 73).
Antarctic blue whale 'Z' calls were also noted in data recorded throughout the region, in addition to 'D' calls.
'D' calls are produced by both pygmy and Antarctic blue whales and have been associated with feeding and
reproductive behaviour (74, 75). This combination of blue whale activity emphasises the ecological importance
of central New Zealand as a whole and documents the year-round presence of cetaceans (see also 63, 76).

Contributions from humpback whale song was seasonal, with acoustic energy centred around 300 Hz.
Humpback whales migrate northwards through New Zealand waters between May and August to winter
breeding grounds, and southwards between September to November to summer feeding grounds (32). Within
this monitoring project, humpback song was a major soundscape contributor during July only. It should be
noted that only male humpback whales are thought to produce song, and only vocal animals can be detected
*via* an acoustic methodology. A July peak in song may not fully represent the timing of the northbound
migration if non-singing female or juvenile animals pass through the region with demographic separation (77).
Based on these findings, male humpback whales may not sing on their southbound migration past New
Zealand, or do not follow the same return route. Humpback song only contributed to the soundscape notably
during the northbound migration in one of the two sampled years (2016). Inter-annual differences between 2016
and 2017 were further emphasised by a lack of strong Antarctic blue whale calling at Kaikoura at the end of
June 2017, as there was at the end of June 2016 (**Figure 5C**). Water temperatures or weather conditions may have
differed between the two years, causing animals to favour alternative migration routes, away from the recording

locations. For example, the austral summer of 2017 featured the most intense heatwave on record in New
Zealand (78).

Other baleen whale species contributed to the soundscape more sporadically. Sei, fin and Antarctic minke
whales are all considered data deficient in New Zealand waters (79). Little is known about the fine-scale
movements of these species, but all are expected to undertake seasonal migrations (80). The conservation status
of southern right whales in New Zealand is considered to be 'at risk – recovering' (79), and their seasonal
presence around mainland New Zealand is becoming more common (81). Much of the current knowledge of
New Zealand's marine mammal community is biased by effort in areas where sea conditions are generally calm
and wildlife is expected to be present. PAM provides a method to monitor marine mammals, as a more cost-
effective, non-invasive, larger-scale and longer-term option than visual surveying allows.

The biophonic elements of marine soundscapes recorded in shallow coastal northern New Zealand included
an array of invertebrates and fish that exhibit strong temporal patterns (22, 42, 43). In the central New Zealand
soundscapes characterised here, invertebrate sounds were not noted and the variety of fish sound was less, but
fish chorusing nonetheless featured temporal trends, with highest energy prior to sunset. The individual calls
associated with the chorusing were not evident in the recordings, but the chorus reported here was assumed to
originate from fish as the diel pattern implied a biophonic source, which did not match with known marine
mammal vocalisation frequencies. Its presence during spring to autumn implied a temperature-driven
relationship (82), and the peak prior to dusk suggests a crepuscular preference related to light availability or
diel activity of the fish species or its prey (82). Fish choruses have been reported from other locations (19, 21,
83), and a similar chorus was recorded in the eastern Arabian Sea, with 200 – 2000 Hz pulses that combined into
a chorus with a peak at 900 Hz that occurred immediately after sunset (67). In that instance, the origin was
suspected to be planktivorous fish (67). Further investigation of the fish chorus reported here would be
beneficial, as fine-scale variation of ambient levels of this sound may indicate its specific location in the region,
allowing precise targeting for research.

26 **Anthrophony**

The influence of shipping activity on marine soundscapes around central New Zealand was considered to
be approximately typical of an environment close to the continental shelf near a moderate sized port, with the
sound of shipping influencing the soundscape at low frequencies year-round, with greatest contributions when
ships passed close to a recorder. This demonstrates a number of similarities to the soundscape of the Perth
Canyon, Australia (68). Further investigation would be required to quantify the patterns of contribution from
shipping on the soundscape of central New Zealand as, over the two recording periods, it was a constant
contributor with some variation. Sound levels in the 63 and 125 Hz 1/3-octave-bands, which the EU Marine
Strategy Framework Directive deems relevant to vessel sound (57), were disproportionately high at Cook Strait,
implying that the acoustic contribution of vessel traffic was concentrated in the Strait.

In addition to the recording locations reported here, further acoustic moorings were placed in the central
New Zealand marine environment during the study periods (63, 84-86). During the first deployment period, an
AMAR was located in Queen Charlotte Sound, an arm of the Marlborough Sounds in the north of the South

Island. The soundscape at this location was dominated by vessel traffic between 100 – 1000 Hz due to
recreational, fishing and passenger boats that transited the Sound for an average of twelve hours per day (86).
Small vessels, which are not required to utilise AIS tracking, can contribute significant noise across large
frequency bands in shallow water (87). However, unlike the deep water environments at Kaikoura and
Wairarapa where vessel passes were detectable for many minutes, or even hours, the narrow, shallow Sound
caused vessel noise to be detectable only for short periods during boat passes (86).

The influence of shipping on marine soundscapes depends upon relative distances to shipping lanes (88,
89). The 95% percentile of sound below 100 Hz in a shallow south Australian bay noted for its lack of shipping
activity was around 75 dB re 1 $\mu\text{Pa}^2 / \text{Hz}$ (65), however, in Cook Strait, the median value of PSD for frequencies
below 100 Hz ranged from 82 to 89 dB re 1 $\mu\text{Pa}^2 / \text{Hz}$ due to shipping activity, in addition to environmental
contributors. In polar regions, where shipping is not concentrated, marine soundscapes can be pristine with
little sound received from this anthropogenic contributor (69, 70, 90). In the Arctic, vessel sound increases during
summer when there is open water (89, 91), and diel variation in shipping activity has been recorded in the Pacific
North West, with highest levels during daylight hours (92). Both local and distant shipping can contribute to
ambient noise levels and such relationships can be directly correlated with bathymetry. In the north and central
Pacific Ocean, sound is able to radiate over vast distances, and the sound from Pacific shipping lanes can
influence coastal soundscapes as much as (93), or even more so than local ship traffic (66). At the deep water
Kaikoura and Wairarapa recorders, distant ship noise from the South Pacific was also likely received, in addition
to sound from close vessel passes.

Seismic surveys influence marine soundscapes globally, from the equator where seismic survey sound can
be near-constant, to the poles where airgun pulses are recorded seasonally or very occasionally (90, 91). Seismic
survey airgun pulses were present in the data recorded during this study, although it should be acknowledged
that recordings made during a different time period may have contained no seismic survey activity. The
contribution from the seismic surveys at each recording location was dependent upon the distance and
bathymetry between the surveys and the recorders. Wairarapa experienced the highest intensity contribution
due to its closest proximity to the Schlumberger Pegasus Basin 3D seismic survey area, and frequencies between
100 and 300 Hz were raised by up to 20 dB re 1 $\mu\text{Pa}^2 / \text{Hz}$ when compared to periods without seismic survey
presence. The reported data reveals a limitation of the PAM approach; the results presented here are a snapshot
of the marine soundscapes of central New Zealand, relevant only to the temporal span of data collection and
the acoustic listening area available to each recorder. If the AMARs had been located in different places, the
reported values would have differed. The quantitative results regarding the seismic surveys exemplify this
issue, both the Kaikoura and Wairarapa recording locations received sound energy from the seismic survey, but
at different intensities, and over different frequency bands, due to different distances from the source of the
sound.

Time and frequency overlap occurred between soundscape contributors. Marine fauna utilise mechanisms
to combat acoustic overlap, such as altering the timing, length and frequency of vocalisations (26), but intense

sounds can interfere with the way fauna use sound, and can overwhelm quieter sounds, known as masking.
Acoustic masking can lead to an inability to communicate, resulting in missed opportunities to maintain fitness,
reduced chance of reproducing, lack of appropriate behavioural responses, and increased risk (if a danger is
unable to be perceived) (26, 28). An illustrated example in the present study was the masking of Antarctic blue
whale 'Z' calls by the sound produced by a ship passing the Kaikoura recording location. It has been indicated
that vessel sound has the potential to reduce communication space of Bryde's whales (*Balaenoptera brydei edeni*)
(not recorded in this data) and fish in northern New Zealand (94), and for marine organisms around the world
(25, 95, 96). All of the baleen whale species recorded in the data produced vocal energy at frequencies below 300
9 Hz and their communication and listening spaces could have been influenced by the sound of shipping and
10 seismic survey activity (29), when it was present, as well as natural sounds such as earthquakes and
11 oceanographic noise.

In addition to masking caused by anthropogenic sound sources, natural sounds, such as earthquakes, can
also occupy frequency bands utilised by biological organisms. In this study, earthquake activity in November
and December 2016 was significant and prolonged, raising the ambient sound levels in the lowest 1/3-octave-
bands (centred at 12.5, 25 and 63 Hz). Seismic surveying concurrently contributed additional energy to these
frequency bands at Wairarapa and Kaikoura. A more in-depth study would be required to quantify the extent
and possible implications of multiple contributors to shared frequency bands in the soundscape of central New
Zealand.

**Future research recommendations**

Many data gaps exist regarding the marine environment around New Zealand. This study adds to a
growing body of literature demonstrating that PAM is a useful tool for monitoring marine soundscapes around
New Zealand as a means to collect long-term baseline data that can be analysed over a range of scales. Such
data helps to combat knowledge gaps to aid in informed management decisions, and acts as a foundation upon
which future data can be compared. Previous studies have generally focused on specific locations, times or
species, which have undoubted value, but a greater body of understanding should be considered a priority. In
this study, fine spatial-scale variation was demonstrated in the soundscape of this relatively small area, further
promoting the need for more work to understand variability. PAM should be considered an important part of
environmental analyses; bottom-mounted acoustic recorders provide a long-term, broad-range, holistic
methodology to gather data that is pertinent to a wide array of ecological and environmental functions. As such,
PAM can be a highly effective monitoring option resulting in high quality data, although care must be taken to
ensure data are collected appropriately using suitable equipment, knowledge and survey design. Indeed,
additional, fine-scale analyses of the data discussed in this study has, and will, elucidate further information
about the region and its marine inhabitants (63, 76, 84, 85).

Soundscape analysis is a relatively recent field of study and provides an array of possible future uses. PAM
is a viable method to provide a tool that could be used by managers to monitor a region. Detailed examination
of acoustic data could reveal whether anthropogenic users are keeping to agreed limits, such as temporal
restrictions on activities or sound-emission constraints.

This study demonstrates that weather and sea conditions affect the soundscape, often at particular
frequencies. Therefore, working backwards, it seems likely that recorded acoustic data could be used as a proxy
for weather or sea conditions (7). A model of the New Zealand soundscape could generalise these contributions,
incorporating weather, waves, tides and modelled examples of earthquakes, biophonic and anthrophonic
elements. Such a model could reveal gaps where other significant contributors to the soundscape should be
accounted for, or other phenomena should be modelled.

6. Conclusion

The soundscapes of central New Zealand demonstrated PSD levels that were positively correlated with
wind speed, rainfall, wave activity, tidal movement, ship passes, seismic surveys, the seasonal and year-round
presence of baleen whales and chorusing fish, and earthquakes, which contributed ambient sound throughout,
as well as rare periods of high intensity sound. Biophonic contributions largely featured temporal patterns in
occurrence, while geophonic contributions were less predictable and in the case of weather and wave activity,
likely cumulative. Anthrophonic contributions were persistent at all locations, although received intensities
varied with distance to sound sources.

Overall, the marine soundscape within Cook Strait was louder than that recorded in deep-water off the east
coast or in STB. The narrow aperture of Cook Strait channelled wind, waves and tidal flows, created a bottleneck
for shipping and wildlife, and combined with the coarse pebble sea-bed to resonate sound around the Strait. All
of these soundscape contributors were recorded at the three other locations, but in wider embayments and
deeper water, the elements contributed to the soundscape in a less concentrated manner. This study confirms
that it is not appropriate to extrapolate soundscapes from coastal, northern New Zealand to coastal and offshore
environments around central New Zealand. Moreover, the marine soundscape of the region is highly dynamic
and not directly comparable to other examples from around the world, even on a per-station basis.

Acknowledgments

Thanks to Christopher Whitt (JASCO Applied Sciences) for training and deployment advice, and to the NIWA acoustic
deployment and retrieval teams: Sarah Searson, Mike Brewer, Olivia Price, Fiona Elliott, and the crews of the RVs Tangaroa,
Kaharoa and Ikatere. Richard Gorman (NIWA, Hamilton) provided wave statistics from the NZWAVE-NZCSM wave
forecast and contributed the associated paragraph in Materials and Methods. Further thanks to Rochelle Constantine
(University of Auckland), Craig Stevens (University of Auckland and NIWA, Wellington), Malcolm Francis and Matt
Pinkerton (NIWA, Wellington) for feedback which improved the manuscript.

Funding Statement

This work was supported by OMV New Zealand Ltd, Chevron New Zealand Holdings LLC, Marlborough District Council,
and Woodside Energy [Woodside Marine Mammal Research Grant, awarded to VEW]. A University of Auckland Doctoral
Scholarship was also awarded to VEW. GG and KTG received co-funding by NIWA Coasts and Oceans Programme 4.

Data Accessibility

Open access data (matrices of per-file 1 Hz PSD, for each recording location) available from Dryad, DOI:
<https://doi.org/10.5061/dryad.vmcvdnpcj>
The private data review URL is: <https://datadryad.org/stash/share/uRglAdTX2mhoqeuVhKkzBajGdSluSlqDCErAnhlowk>

Competing Interests

The authors declare no competing interests.

Authors' Contributions

VEW conducted all analyses and wrote the manuscript. All authors contributed to study design, interpreted data and
critically revised the manuscript. KTG, GG and VEW assisted with data collection. All authors gave final approval for
publication.

References

1. Krause BL. Anatomy of the soundscape: Evolving perspectives. *Journal of the Audio Engineering Society*. 2008;56:73-80.
2. Knudsen VO, Alford RS, Emling JW. Underwater ambient noise. *J Mar Res*. 1948;7(410).
3. Wenz GM. Acoustic ambient noise in the ocean: Spectra and sources. *J Acoust Soc Am*. 1962;34(12):1936-56.
4. Bom N. Effect of rain on underwater noise level. *J Acoust Soc Am*. 1969;45(1):150-6.
5. Heindsmann TE, Smith RH, Arneson AD. Effect of rain upon underwater noise levels. *J Acoust Soc Am*. 1955;27(2):378-9.
6. Scrimger JA, Evans DJ, McBean GA, Farmer DM, Kerman BR. Underwater noise due to rain, hail, and snow. *J Acoust Soc Am*. 1987;81(1):79-86.
7. Ma BB, Nystuen JA, Lien R-C. Prediction of underwater sound levels from rain and wind. *J Acoust Soc Am*. 2005;117(6):3555.
8. Strasberg M. Nonacoustic noise interference in measurements of infrasonic ambient noise. *J Acoust Soc Am*. 1979;66(5):1487-93.
9. Urick RJ. Principles of underwater sound. 3rd ed. New York: McGraw-Hill; 1983.
10. Willis J, Dietz FT. Effect of tidal currents on 25 cps shallow water ambient noise measurements. *J Acoust Soc Am*. 1961;33(11):1659-.
11. Matsumoto H, Bohnenstiehl DR, Tournadre J, Dziak RP, Haxel JH, Lau TKA, et al. Antarctic icebergs: A significant natural ocean sound source in the Southern Hemisphere. *Geochem Geophys Geosyst*. 2014;15(8):3448-58.
12. Urick RJ. Sea-bed motion as a source of the ambient noise background of the sea. *J Acoust Soc Am*. 1974;56(3):1010-1.
13. Matthews LP, Mccordic JA, Parks SE. Remote acoustic monitoring of North Atlantic right whales (*Eubalaena glacialis*) reveals seasonal and diel variations in acoustic behavior. *PLoS ONE*. 2014;9(3).
14. Leroy EC, Samaran F, Bonnel J, Royer J-Y. Seasonal and diel vocalization patterns of Antarctic blue whale (*Balaenoptera musculus intermedia*) in the southern Indian Ocean: A multi-year and multi-site study. *PLoS ONE*. 2016;11(11).
15. Versluis M, Schmitz B, Von Der Heydt A, Lohse D. How snapping shrimp snap: Through cavitating bubbles. *Science*. 2000;289(5487):2114-7.
16. Harris SA, Shears NT, Radford CA. Ecoacoustic indices as proxies for biodiversity on temperate reefs. *Methods Ecol Evol*. 2016;7(6):713-24.
17. Radford CA, Jeffs A, Montgomery JC, Tindle C. Resonating sea urchin skeletons create coastal choruses. *Mar Ecol Prog Ser*. 2008;362:37-43.
18. Radford CA, Jeffs AG, Tindle CT, Montgomery JC. Temporal patterns in ambient noise of biological origin from a shallow water temperate reef. *Oecologia*. 2008;156:921-9.
19. D'Spain GL, Batchelor HH. Observations of biological choruses in the Southern California Bight: A chorus at midfrequencies. *J Acoust Soc Am*. 2006;120(4):1942-55.
20. McCauley RD, Cato DH. Patterns of fish calling in a nearshore environment in the Great Barrier Reef. *Philos Trans R Soc Lond B Biol Sci*. 2000;355(1401):1289-93.
21. McCauley RD, Cato DH. Evening choruses in the Perth Canyon and their potential link with Myctophidae

- fishes. *J Acoust Soc Am*. 2016;140(4):2384-98.
22. Putland RL, Constantine R, Radford CA. Exploring spatial and temporal trends in the soundscape of an ecologically significant embayment. *Sci Rep*. 2017;7(1).
23. Krause BL. The habitat niche hypothesis: a hidden symphony of animal sounds. *The Literary Review*. 1992;36(1):40-5.
24. van Opzeeland IC, Boebel O. Marine soundscape planning: Seeking acoustic niches for anthropogenic sound. *J Ecoacoustics*. 2018;2.
25. Erbe C, Marley SA, Schoeman R, Smith J, Trigg L, Embling C. The effects of ship noise on marine mammals - A review. *Front Mar Sci*. 2019;6.
26. Erbe C, Reichmuth C, Cunningham K, Lucke K, Dooling R. Communication masking in marine mammals: A review and research strategy. *Mar Pollut Bull*. 2016;103(1-2):15-38.
27. Kavanagh AS, Nykänen M, Hunt W, Richardson N, Jessopp MJ. Seismic surveys reduce cetacean sightings across a large marine ecosystem. *Sci Rep*. 2019;9(19164).
28. Tyack PL. Implications for marine mammals of large-scale changes in the marine acoustic environment. *J Mammal*. 2008;89(3):549-58.
29. Nowacek DP, Clark CW, Mann D, Miller PJO, Rosenbaum HC, Golden JS, et al. Marine seismic surveys and ocean noise: time for coordinated and prudent planning. *Front Ecol Environ*. 2015;13:378-86.
30. Harris TFW. Greater Cook Strait: form and flow. Wellington, N.Z.: DSIR Marine and Freshwater; 1990.
31. Paul LJ. Seasonal fishing patterns in the commercial fishery for groper in New Zealand, with notes on reproduction and apparent migration in *Polyprion oxygeneios* and *P. americanus*: results of a questionnaire sent to commercial fishers. New Zealand Fisheries Assessment Report. 2005;62(2005).
32. Dawbin WH. The migrations of humpback whales which pass the New Zealand coast. *Trans R Soc N Z*. 1956;84:147-96.
33. Barlow DR, Torres LG, Hodge KB, Steel D, Baker CS, Chandler TE, et al. Documentation of a New Zealand blue whale population based on multiple lines of evidence. *Endanger Species Res*. 2018;36:27-40.
34. Chiswell SM, Zeldis JR, Hadfield MG, Pinkerton MH. Wind-driven upwelling and surface chlorophyll blooms in Greater Cook Strait. *NZ J Mar FreshWater Res*. 2017;51(4):465-89.
35. Stevens CL, O'Callaghan JM, Chiswell SM, Hadfield MG. Physical oceanography of New Zealand/Aotearoa shelf seas – a review. *NZ J Mar FreshWater Res*. 2019:1-40.
36. Stevens CL, Smith MJ, Grant B, Stewart CL, Divett T. Tidal energy resource complexity in a large strait: The Karori Rip, Cook Strait. *Cont Shelf Res*. 2011;33:100-9.
37. Walters R, Gillibrand P, Bell R, Lane E. A study of tides and currents in Cook Strait, New Zealand. *Ocean Dynamics*. 2010;60(6):1559-80.
38. Gregg R, Walrond C. 'Oil and gas - The Māui gas field' <http://www.teara.govt.nz/en/map/8934/taranaki-oil-and-gas-fields-2006>: Te Ara - the Encyclopedia of New Zealand; 2006 [12/11/2019].
39. Buckingham MJ, Jones SAS. A new shallow-ocean technique for determining the critical angle of the seabed from the vertical directionality of the ambient noise in the water column. *J Acoust Soc Am*. 1987;81(4):938-46.
40. Bostock H, Jenkins C, Mackay K, Carter L, Nodder S, Orpin A, et al. Distribution of surficial sediments in the ocean around New Zealand/Aotearoa. Part B: continental shelf. *NZ J Geol Geophys*. 2019;62:24-45.
41. Wille PC. Sound images of the ocean in research and monitoring. SpringerLink, editor. Berlin New York: Springer; 2005.
42. Pine MK, Radford CA, Jeffs AG. Eavesdropping on the Kaipara Harbour: characterising underwater soundscapes within a seagrass bed and a subtidal mudflat. *NZ J Mar FreshWater Res*. 2015:247-58.
43. Radford CA, Stanley JA, Montgomery CT, Jeffs JC, Tindle AG. Localised coastal habitats have distinct underwater sound signatures. *Mar Ecol Prog Ser*. 2010;401:21-9.
44. Freeman L, Freeman S. Rapidly obtained ecosystem indicators from coral reef soundscapes. *Mar Ecol Prog Ser*. 2016;561:69-82.
45. QGIS Development Team. QGIS Geographic Information System. Open Source Geospatial Foundation Project <http://qgis.org>. 2.18.9 ed 2017.
46. MATLAB. Natick, Massachusetts, United States: The MathWorks, Inc.; R2016b.
47. Merchant ND, Barton TR, Thompson PM, Pirotta E, Dakin DT, Dorocicz J. Spectral probability density as a tool for

- ambient noise analysis. *J Acoust Soc Am.* 2013;133(4):EL262-EL7.
48. PAMLab-Lite. version 8.3.3 ed: JASCO Applied Sciences; 2017.
49. McDonald MA, Mesnick SL, Hildebrand JA. Biogeographic characterisation of blue whale song worldwide: using song to identify populations *J Cetacean Res Manag.* 2006;8(1):55-65.
50. Miller BS, Collins K, Barlow J, Calderan S, Leaper R, McDonald M, et al. Blue whale vocalizations recorded around New Zealand: 1964–2013. *The Journal of the Acoustical Society of America.* 2014;135(3):1616-23.
51. Payne RS, McVay S. Songs of humpback whales. *Science.* 1971;173(3997):585-97.
52. Risch D, Gales NJ, Gedamke J, Kindermann L, Nowacek DP, Read AJ, et al. Mysterious bio-duck sound attributed to the Antarctic minke whale (*Balaenoptera bonaerensis*). *Biol Lett.* 2014;10(4).
53. Širović A, Hildebrand JA, Wiggins SM, Thiele D. Blue and fin whale acoustic presence around Antarctica during 2003 and 2004. *Mar Mamm Sci.* 2009;25(1):125-36.
54. Tremblay CJ, Van Parijs SM, Cholewiak D. 50 to 30-Hz triplet and singlet down sweep vocalizations produced by sei whales (*Balaenoptera borealis*) in the western North Atlantic Ocean. *J Acoust Soc Am.* 2019;145(6):3351-8.
55. Webster TA, Dawson SM, Rayment WJ, Parks SE, Van Parijs SM. Quantitative analysis of the acoustic repertoire of southern right whales in New Zealand. *J Acoust Soc Am.* 2016;140(1):322-33.
56. Martin B. Computing cumulative sound exposure levels from anthropogenic sources in large data sets. *ICA* 2013; Montreal: Proceedings of Meetings on Acoustics; 2013.
57. Tasker ML, Amundin M, Andre M, Hawkins A, Lang W, Merck T, et al. Task Group 11 Report: Underwater noise and other forms of energy. 2010.
58. Kaiser A, Balfour N, Fry B, Holden C, Litchfield N, Gerstenberger M, et al. The 2016 Kaikoura, New Zealand, Earthquake: Preliminary Seismological Report. *Seismol Res Lett.* 2017;88(3).
59. Leroy EC, Samaran F, Bonnel J, Royer J-Y. Identification of two potential whale calls in the southern Indian Ocean, and their geographic and seasonal occurrence. *J Acoust Soc Am.* 2017;142(3):1413-27.
60. Ward R, Gavrilov AN, McCauley RD. "Spot" call: A common sound from an unidentified great whale in Australian temperate waters. *J Acoust Soc Am.* 2017;142(2):EL231-EL6.
61. McDonald MA, Calambokidis J, Teranishi AM, Hildebrand JA. The acoustic calls of blue whales off California with gender data. *J Acoust Soc Am.* 2001;109(4):1728.
62. Baumgartner MF, Fratantoni DM, Hurst TP, Brown MW, Cole TVN, Van Parijs SM, et al. Real-time reporting of baleen whale passive acoustic detections from ocean gliders. *J Acoust Soc Am.* 2013;134(3):1814-23.
63. Giorli G, Goetz KT. Foraging activity of sperm whales (*Physeter macrocephalus*) off the east coast of New Zealand. *Sci Rep.* 2019;9(1):1-9.
64. McDonald MA, Hildebrand JA, Wiggins SM, Ross D. A 50 Year comparison of ambient ocean noise near San Clemente Island: A bathymetrically complex coastal region off Southern California. *J Acoust Soc Am.* 2008;124(4):1985-92.
65. Ward R, McCauley RD, Gavrilov AN, Charlton C. Underwater Sound Sources and Ambient Noise in Fowlers Bay, South Australia, during the Austral Winter. *Acoust Aust.* 2019;47(1):21-32.
66. Širović A, Wiggins SM, Oleson EM. Ocean noise in the tropical and subtropical Pacific Ocean. *J Acoust Soc Am.* 2013;134(4):2681-9.
67. Mahanty MM, Ganeshan L, Govindan R. Soundscapes in shallow water of the eastern Arabian Sea. *Prog Oceanogr.* 2018;165:158-67.
68. Erbe C, Verma A, McCauley RD, Gavrilov AN, Parnum I. The marine soundscape of the Perth Canyon. *Prog Oceanogr.* 2015;137:38-51.
69. Boebel O, Klinck H, Kindermann L, Naggar SEDE. PALAOA: Broadband recordings of the Antarctic coastal soundscape. *Bioacoustics.* 2008;17(1-3):18-21.
70. Menze S, Zitterbart DP, Van Opzeeland IC, Boebel O. The influence of sea ice, wind speed and marine mammals on Southern Ocean ambient sound. *R Soc Open Sci.* 2017;4(1):160370.
71. Torres LG. Evidence for an unrecognised blue whale foraging ground in New Zealand. *NZ J Mar FreshWater Res.* 2013;47(2):235-48.
72. Barlow DR, Bernard K, Escobar-Flores P, Palacios DM, Torres LG. Cloudy with a chance of whales: Forecasting blue whale occurrence based on tiered, bottom-up models to mitigate industrial impacts. *World Marine Mammal Conference 2019; Barcelona2019.*
73. Goetz K, Childerhouse S, Paton D, Ogle M, Hupman K, Constantine R, et al. Satellite

- tracking of blue whales in New Zealand waters, 2018 voyage report. 2018:11.
74. Oleson EM, Calambokidis J, Burgess W, McDonald MA, Leduc C, Hildebrand JA. Behavioral context of call production by eastern North Pacific blue whales. *Mar Ecol Prog Ser.* 2007;330:269-84.
75. Schall E, Di Iorio L, Berchok C, Filún D, Bedriñana-Romano L, Buchan SJ, et al. Visual and passive acoustic observations of blue whale trios from two distinct populations. *Mar Mamm Sci.* 2019:1-10.
76. Warren VE, Constantine R, Širović A, McPherson C, Radford CA, Goetz KT. Spatio-temporal distribution of pygmy and Antarctic blue whales in central New Zealand, via passive acoustic monitoring. *World Marine Mammal Conference; Barcelona2019.*
77. Dawbin WH. Temporal segregation of humpback whales during migration in Southern Hemisphere waters. *Mem Queensl Mus.* 1997;42:105-38.
78. Salinger MJ, Renwick J, Behrens E, Mullan AB, Diamond HJ, Sirguy P, et al. The unprecedented coupled ocean-atmosphere summer heatwave in the New Zealand region 2017/18: drivers, mechanisms and impacts. *Environ Res.* 2019;14(4):044023.
79. Baker CS, Boren LJ, Childerhouse S, Constantine R, van Helden A, Lundquist D, et al. Conservation status of New Zealand marine mammals, 2019. *New Zealand Threat Classification Series* 29. 2019.
80. Stern SJ. Migration and movement patterns. In: Perrin WF, Würsig B, Thewissen JGM, editors. *Encyclopedia of marine mammals.* Academic Press; 2002. p. 742-8.
81. Carroll EL, Rayment WJ, Alexander AM, Baker CS, Patenaude NJ, Steel D, et al. Reestablishment of former wintering grounds by New Zealand southern right whales. *Mar Mamm Sci.* 2014;30(1):206-20.
82. Jamie NM, Robert DM, Christine E, Miles JGP. Patterns of biophonic periodicity on coral reefs in the Great Barrier Reef. *Sci Rep.* 2017;7(1):1-13.
83. Parsons MJG, Salgado-Kent CP, Marley SA, Gavrilov AN, McCauley RD. Characterizing diversity and variation in fish choruses in Darwin Harbour. *ICES J Mar Sci.* 2016;73(8):2058-74.
84. Giorli G, Goetz KT. Acoustically estimated size distribution of sperm whales (*Physeter macrocephalus*) off the east coast of New Zealand. *NZ J Mar Freshwater Res.* 2019:1-12.
85. Giorli G, Goetz KT, Delarue J, Maxner E, Kowarski KA, Bruce Martin S, et al. Unknown beaked whale echolocation signals recorded off eastern New Zealand. *The Journal of the Acoustical Society of America.* 2018;143(4):EL285-EL91.
86. Goetz KT, Hupman K. Passive Acoustic Monitoring in the greater Cook Strait region with particular focus on Queen Charlotte Sound / Tōtaranui. <https://www.marlborough.govt.nz/repository/libraries/id:1w1mps0ir17q9sgxanf9/hierarchy/Documents/Environment/Coastal/Scientific%20Investigations%20List/Queen%20Charlotte%20Sound%20underwater%20soundscape:NIWA;2017.Report.No.:2017216WN>.
87. Line H, Lonnie M, Jakob T, Kristian B, Mark J, Peter TM. Recreational vessels without Automatic Identification System (AIS) dominate anthropogenic noise contributions to a shallow water soundscape. *Sci Rep.* 2019;9(1):1-10.
88. Curtis KR, Howe BM, Mercer JA. Low-frequency ambient sound in the North Pacific: Long time series observations. *The Journal of the Acoustical Society of America.* 1999;106(6):3189-200.
89. Martin SB, Morris C, Bröker K, O'Neill C. Sound exposure level as a metric for analyzing and managing underwater soundscapes a). *J Acoust Soc Am.* 2019;146(1):135-49.
90. Haver SM, Klinck H, Nieukirk SL, Matsumoto H, Dziak RP, Miksis-Olds JL. The not-so-silent world: Measuring Arctic, Equatorial, and Antarctic soundscapes in the Atlantic Ocean. *Deep-Sea Res Part I Oceanogr Res Pap.* 2017;122:95-104.
91. Ahonen H, Stafford KM, de Steur L, Lydersen C, Wiig Ø, Kovacs KM. The underwater soundscape in western Fram Strait: Breeding ground of Spitsbergen's endangered bowhead whales. *Mar Pollut Bull.* 2017;123:97-112.
92. Williams R, Clark CW, Ponirakis D, Ashe E. Acoustic quality of critical habitats for three threatened whale populations. *Anim Conserv.* 2014;17(2):174-85.
93. Haxel JH, Dziak RP, Matsumoto H. Observations of shallow water marine ambient sound: The low frequency underwater soundscape of the central Oregon coast. *The Journal of the Acoustical Society of America.* 2013;133(5):2586-96.
94. Putland RL, Merchant ND, Farcas A, Radford CA. Vessel noise cuts down

communication space for
vocalizing fish and marine
mammals. *Glob Change Biol.*
2018;24(4):1708-21.
95. Cholewiak D, Clark
CW, Ponirakis D, Frankel AS,
Hatch LT, Risch D, et al.
Communicating amidst the

noise: Modeling the aggregate
influence of ambient and vessel
noise on baleen whale
communication space in a
national marine sanctuary.
Endanger Species Res.
2018;36:59-75.

96. Dunlop R. The effects of
vessel noise on the
communication network of
humpback whales. *R Soc Open
Sci.* 2019;6:190967.

9 1
10 2

Table 1. Deployment information per recording location (time standard: UTC).

	South Taranaki Bight	Cook Strait	Kaikoura	Wairarapa
Deployment 1				
Position	40.42°S, 174.50°E	41.09°S, 174.55°E	42.31°S, 174.21°E	41.61°S, 175.90°E
Water depth (m)	110	252	1251	1481
Deployment	4 June 2016	4 June 2016	6 June 2016	5 June 2016
Retrieval	20 Dec 2016	19 Dec 2016	21 Dec 2016	21 Dec 2016
Deployment 2				
Position	40.53°S, 174.76°E	-	42.31°S, 174.21°E	41.61°S, 175.90°E
Water depth (m)	112	-	1254	1486
Deployment	23 Feb 2017	-	21 Feb 2017	22 Feb 2017
Retrieval	29 Aug 2017	-	8 Sept 2017	30 Aug 2017

Table 2. Raw values for each rainfall scenario, per weather station.

Weather Station	Rainfall (mm)			
	No	Low	Mid	High
Whanganui	0	0 - 0.8	0.8 - 1.5	1.5 - 2.3
Akitio	0	0 - 1.3	1.3 - 2.6	2.6 - 3.9
Wellington	0	0 - 1.3	1.3 - 2.6	2.6 - 3.9
Blenheim	0	0 - 0.4	0.4 - 0.9	0.9 - 1.3
Waipara	0	0 - 0.9	0.9 - 1.8	1.8 - 2.7

Table 3. Raw values for each wind speed scenario, per weather station.

Weather Station	Wind speed (m/s)		
	Low	Mid	High
Whanganui	0.9 - 4.0	4.0 - 7.2	7.2 - 10.3
Akitio	1.8 - 10.3	10.3 - 18.8	18.8 - 27.3
Wellington	2.1 - 10.9	10.9 - 19.8	19.8 - 28.6
Blenheim	0.4 - 2.9	2.9 - 5.3	5.3 - 7.8
Waipara	0.9 - 4.5	4.5 - 8.1	8.1 - 11.6

Table 4. Raw values for each modelled significant wave height scenario, per recording location.

Recording Location	Modelled significant wave height (m)		
	Low	Mid	High
STB	0.2 - 2.4	2.4 - 4.6	4.6 - 6.8
Cook Strait	0.1 - 1.9	1.9 - 3.7	3.7 - 5.4
Kaikoura	0.4 - 2.8	2.8 - 5.2	5.2 - 7.6
Wairarapa	0.6 - 3.8	3.8 - 7.0	7.0 - 10.2

COLOUR FIGURE Figure 1. Map of the Cook Strait region, New Zealand showing AMAR deployment locations (first deployment (June – December 2016) = black circles, second deployment (February – September 2017) = white triangles), weather stations (yellow circles), boundaries of earthquake data (open red box), operation borders of the PGS Taranaki South 3D seismic survey (purple filled area), and Schlumberger Pegasus Basin 3D seismic survey (red filled area).

COLOUR FIGURE Figure 2. Median PSD levels per station, per deployment (solid line = June to December 2016, dotted line = February to September 2017). A recorder was only deployed in Cook Strait (orange) in 2016. 1 Hz frequency resolution.

COLOUR FIGURE Figure 3. Average SPD of **A)** STB, **B)** Cook Strait, **C)** Kaikoura, and **D)** Wairarapa for the 2016 (**left**) and 2017 (**right**) deployments. An AMAR was not redeployed in Cook Strait in 2017. The probability of a given PSD at a given frequency is represented as colour (colour bar is applicable to all axes). 1 Hz frequency resolution.

COLOUR FIGURE Figure 4. Median 1/3-octave-band values over both deployments at **B)** STB, **C)** Cook Strait, **D)** Kaikoura and **E)** Wairarapa. The legend for the 1/3-octave-band centre frequencies is provided in panel **C)**. PSD values are daily averages following the application of a 7-day moving average. Grey areas indicate periods when the AMARs were not deployed. Panel **A)** illustrates the output of the seismic pulse detector for the STB data (purple bars) and the Wairarapa data (red bars). Detections at STB are of the PGS Taranaki South 3D seismic survey, and detections at Wairarapa are of the Schlumberger Pegasus Basin 3D seismic survey. Panel **F)** illustrates the total number of earthquakes (magnitude >0.5) in the study area (**red box, Figure 1**), per day.

Figure 5. Long-term spectral averages of **A)** STB, **B)** Cook Strait, **C)** Kaikoura, and **D)** Wairarapa for both deployments. The PSD of a given frequency at a given date/time is represented as colour (colour bar is applicable to all axes). White areas indicate periods when the AMARs were not deployed.

COLOUR FIGURE Figure 6. Median PSDs corresponding to given rainfall scenarios for **A)** STB, **B)** Cook Strait, **C)** Kaikoura, and **D)** Wairarapa. PSD dips at 1500 and 2000 Hz at Kaikoura (**C)** and Wairarapa (**D)** are due to destructive interference and are not true features of the soundscape.

COLOUR FIGURE Figure 7. Median PSDs corresponding to given wind speed scenarios for **A)** STB, **B)** Cook Strait, **C)** Kaikoura, and **D)** Wairarapa. PSD dips at 1500 and 2000 Hz at Kaikoura (**C)** and Wairarapa (**D)** are due to destructive interference and are not true features of the soundscape.

COLOUR FIGURE Figure 8. Median PSDs corresponding to given modelled significant wave height scenarios for **A)** STB, **B)** Cook Strait, **C)** Kaikoura, and **D)** Wairarapa. PSD dips at 1500 and 2000 Hz at

1 Kaikoura (C) and Wairarapa (D) are due to destructive interference and are not true features of the
2 soundscape.

**COLOUR FIGURE Figure 9.** Median PSDs corresponding to given tidal states for A) STB, B) Cook Strait, C)
Kaikoura, and D) Wairarapa. PSD dips at 1500 and 2000 Hz at Kaikoura (C) and Wairarapa (D) are due to
destructive interference and are not true features of the soundscape.

**Figure 10.** Baleen whale vocalization spectrograms. A) Pygmy blue whale song, STB, 16 July 2016. B) Antarctic
blue whale Z calls, Kaikoura, 27 June 2016. C) Possible 'spot' call, Kaikoura, 9 May 2017. D) Blue whale 'D'
calls, Kaikoura, 11 March 2017. E) Possible sei whale downsweeps, STB, 12 April 2017. F) Fin whale calls,
Kaikoura, 27 June 2017. G) Humpback whale song, STB, 22 July 2016. H) Possible southern right whale
upcalls, Cook Strait, 24 August 2016. I) Antarctic minke whale 'bio-duck', STB, 2 October 2016. A-F) 0.244 Hz
frequency resolution, 2 s time window, 0.5 s time step, Hamming window. G- I) 1.95 Hz frequency resolution,
0.128 s time window, 0.032 s time step, Hamming window. Note that x- and y-axes differ per spectrogram.

**COLOUR FIGURE Figure 11.** Monthly SPDs with exceedance percentiles (from bottom: 99, 95, 50, 5, 1%)
featuring baleen whale calls recorded at South Taranaki Bight. A) In April 2017, pygmy blue whale song
generated peaks in SPD and 95% of all data at 16/17 and 23 Hz. B) In July 2016, humpback whale vocalisations
generated a peak in SPD and all percentiles between 250 and 400 Hz. Note scale differences on both axes.

**Figure 12.** A) The median value of the 1000 Hz 1/3-octave-band at Wairarapa from 30 November to 20
December 2016. Grey areas indicate dusk (8-10pm, NZDT); periods of raised PSD in the 1000 Hz 1/3-octave-
band as a result of chorusing fish occur prior to dusk. B) The median value of the 1000 Hz 1/3-octave-band at
Wairarapa from 30 June to 20 July 2016. Grey areas indicate dusk (5-6pm, NZDT) and demonstrates the
absence of diel pattern in the 1000 Hz 1/3-octave-band due to a lack of fish chorusing.

**Figure 13.** Spectrogram of an unidentified, but probable, fish sounds recorded at STB, 1 October 2016 (1.95 Hz
frequency resolution, 0.128 s time window, 0.032 s time step, Hamming window).

**Figure 14.** Spectrogram of ship pass recorded at Kaikoura, 18 June 2016. Antarctic blue whale Z calls are
evident as vertical signals between 18 and 26 Hz throughout the illustrated period. The spectrogram includes
data sampled at both 16 and 250 kHz, and black vertical bars correspond with duty cycled gaps between
recordings (0.244 Hz frequency resolution, 2 s time window, 0.5 s time step, Hamming window).

Figure 1. Map of the Cook Strait region, New Zealand showing AMAR deployment locations (first deployment (June – December 2016) = black circles, second deployment (February – September 2017) = white triangles), weather stations (yellow circles), boundaries of earthquake data (open red box), operation borders of the PGS Taranaki South 3D seismic survey (purple filled area), and Schlumberger Pegasus Basin 3D seismic survey (red filled area).

254x190mm (300 x 300 DPI)

Figure 2. Median PSD levels per station, per deployment (solid line = June to December 2016, dotted line = February to September 2017). A recorder was only deployed in Cook Strait (orange) in 2016. 1 Hz frequency resolution.

254x190mm (300 x 300 DPI)

Figure 3. Average SPD of A) STB, B) Cook Strait, C) Kaikoura, and D) Wairarapa for the 2016 (left) and 2017 (right) deployments. An AMAR was not redeployed in Cook Strait in 2017. The probability of a given PSD at a given frequency is represented as colour (colour bar is applicable to all axes). 1 Hz frequency resolution.

254x190mm (300 x 300 DPI)

Figure 4. Median 1/3-octave-band values over both deployments at B) STB, C) Cook Strait, D) Kaikoura and E) Wairarapa. The legend for the 1/3-octave-band centre frequencies is provided in panel C). PSD values are daily averages following the application of a 7-day moving average. Grey areas indicate periods when the AMARs were not deployed. Panel A) illustrates the output of the seismic pulse detector for the STB data (purple bars) and the Wairarapa data (red bars). Detections at STB are of the PGS Taranaki South 3D seismic survey, and detections at Wairarapa are of the Schlumberger Pegasus Basin 3D seismic survey. Panel F) illustrates the total number of earthquakes (magnitude >0.5) in the study area (red box, Figure 1), per day.

254x190mm (300 x 300 DPI)

Figure 5. Long-term spectral averages of A) STB, B) Cook Strait, C) Kaikoura, and D) Wairarapa for both deployments. The PSD of a given frequency at a given date/time is represented as colour (colour bar is applicable to all axes). White areas indicate periods when the AMARs were not deployed.

254x190mm (300 x 300 DPI)

Figure 6. Median PSDs corresponding to given rainfall scenarios for A) STB, B) Cook Strait, C) Kaikoura, and D) Wairarapa. PSD dips at 1500 and 2000 Hz at Kaikoura (C) and Wairarapa (D) are due to destructive interference and are not true features of the soundscape.

254x190mm (300 x 300 DPI)

Figure 7. Median PSDs corresponding to given wind speed scenarios for A) STB, B) Cook Strait, C) Kaikoura, and D) Wairarapa. PSD dips at 1500 and 2000 Hz at Kaikoura (C) and Wairarapa (D) are due to destructive interference and are not true features of the soundscape.

254x190mm (300 x 300 DPI)

Figure 8. Median PSDs corresponding to given modelled significant wave height scenarios for A) STB, B) Cook Strait, C) Kaikoura, and D) Wairarapa. PSD dips at 1500 and 2000 Hz at Kaikoura (C) and Wairarapa (D) are due to destructive interference and are not true features of the soundscape.

254x190mm (300 x 300 DPI)

Figure 9. Median PSDs corresponding to given tidal states for A) STB, B) Cook Strait, C) Kaikoura, and D) Wairarapa. PSD dips at 1500 and 2000 Hz at Kaikoura (C) and Wairarapa (D) are due to destructive interference and are not true features of the soundscape.

254x190mm (300 x 300 DPI)

Figure 10. Baleen whale vocalization spectrograms. A) Pygmy blue whale song, STB, 16 July 2016. B) Antarctic blue whale Z calls, Kaikoura, 27 June 2016. C) Possible 'spot' call, Kaikoura, 9 May 2017. D) Blue whale 'D' calls, Kaikoura, 11 March 2017. E) Possible sei whale downsweps, STB, 12 April 2017. F) Fin whale calls, Kaikoura, 27 June 2017. G) Humpback whale song, STB, 22 July 2016. H) Possible southern right whale upcalls, Cook Strait, 24 August 2016. I) Antarctic minke whale 'bio-duck', STB, 2 October 2016. A-F) 0.244 Hz frequency resolution, 2 s time window, 0.5 s time step, Hamming window. G- I) 1.95 Hz frequency resolution, 0.128 s time window, 0.032 s time step, Hamming window. Note that x- and y-axes differ per spectrogram.

254x190mm (300 x 300 DPI)

Figure 11. Monthly SPDs with exceedance percentiles (from bottom: 99, 95, 50, 5, 1%) featuring baleen whale calls recorded at South Taranaki Bight. A) In April 2017, pygmy blue whale song generated peaks in SPD and 95% of all data at 16/17 and 23 Hz. B) In July 2016, humpback whale vocalisations generated a peak in SPD and all percentiles between 250 and 400 Hz. Note scale differences on both axes.

254x190mm (300 x 300 DPI)

Figure 12. A) The median value of the 1000 Hz 1/3-octave-band at Wairarapa from 30 November to 20 December 2016. Grey areas indicate dusk (8-10pm, NZDT); periods of raised PSD in the 1000 Hz 1/3-octave-band as a result of chorusing fish occur prior to dusk. B) The median value of the 1000 Hz 1/3-octave-band at Wairarapa from 30 June to 20 July 2016. Grey areas indicate dusk (5-6pm, NZDT) and demonstrates the absence of diel pattern in the 1000 Hz 1/3-octave-band due to a lack of fish chorusing.

254x190mm (300 x 300 DPI)

Figure 13. Spectrogram of an unidentified, but probable, fish sounds recorded at STB, 1 October 2016 (1.95 Hz frequency resolution, 0.128 s time window, 0.032 s time step, Hamming window).

254x190mm (300 x 300 DPI)

Figure 14. Spectrogram of ship pass recorded at Kaikoura, 18 June 2016. Antarctic blue whale Z calls are evident as vertical signals between 18 and 26 Hz throughout the illustrated period. The spectrogram includes data sampled at both 16 and 250 kHz, and black vertical bars correspond with duty cycled gaps between recordings (0.244 Hz frequency resolution, 2 s time window, 0.5 s time step, Hamming window).

254x190mm (300 x 300 DPI)

Appendix B

Marine soundscape variation in an oceanographically diverse region, central New Zealand
Warren, V.E. et al. Royal Society Open Science

General Comments

After reviewing the manuscript, I commend the authors for their research and valuable contribution to the field of underwater acoustics and soundscape analysis. The study examines the components of and contributors to the soundscapes of four spatially separated locations in the region of central New Zealand. As the authors state, this region of the ocean has been understudied, and the authors present and make a strong case that passive acoustic monitoring can help fill in data gaps at this location and elsewhere. I was especially impressed with the holistic approach that the authors took, and their ability to tease out different acoustic sources through comparisons of different acoustic metrics with ancillary data types. The inclusion of earthquakes was a nice addition. Very rarely do studies have the ability to document biophonic, geophonic, and anthropogenic contributions to soundscapes in a single study. However, this manuscript exemplifies this potential and will serve as a value contribution and reference for many ongoing studies in the field of soundscape analysis. Thus, after minor edits and further explanations of methods, which I detail below, I would expect this manuscript to be of great interest to the readers of Royal Society Open Science.

Specific Comments

Page 1

Line 21-22: "Median power spectral density." The use of "power" when referring to this term is widely used, but the authors may consider the use of "median pressure spectral density" instead, as nowhere in PSD is there included power, which has the units of watts. PSD can be seen as proportional to power. This comment is just a recommendation to use the appropriate terms; although I know others may disagree as the use of Power Spectral Density is a legacy term. If accepted, the authors would want to make changes throughout the manuscript.

Line 23: The authors use "correlated" throughout the manuscript, yet one would need to also provide a statistical test and results if statements about correlations are made. Consider using a different term throughout the manuscript or provide a statistical test.

Page 2

Line 18: The use of "vocalisations" is not appropriate for fish and invertebrates. Consider changing this word to "sounds."

Page 3

Line 15: An abbreviation is provided for South Taranaki Bight in this line, yet this site is also written out fully on Page 4 (Line 5-6) of the methods. I would suggest omitting the abbreviation from this line so that readers can refer to a full list of abbreviations on the following page.

Page 4

Line 11: I would appreciate some further details about the flow shield. What was it made out of? Can you include a graphic of your different mooring designs? Have the effects of the flow shield on received levels been assessed? While this comment maybe be selfish, I foresee other readers being interested in incorporating flow shields into their studies, so some further details are warranted. Additionally, without knowing about possible effects of the flow shield, one could speculate that the results of the study might have been affected by its presence at two of the four locations.

Lines 24-33: While this section on “Acoustic Processing” is nice and concise, I think that some details need to be expanded upon or clarified given its importance to the rest of the study.

Line 24: “Power” → “Pressure” ?

Line 26: 80% overlap seems like a strange choice over the typical 0%, 50%, 75% options. Was there any reason why this was chosen? If so, include it. If it is just what the authors routinely use, then they can ignore this comment.

Line 28: Please indicate from what data the exceedance percentiles were calculated from. I am assuming that they were calculated from the PSD values per each 630 s file, but this is not 100% clear given that this sentence follows a sentence about SPD calculations. Also were these calculated for the duration of each deployment, or per month, etc.? It would seem different time periods were selected based off the analysis, e.g. per deployment vs. per geophonic event period.

Line 30: “...concatenating the 1 Hz resolution PSD averages for all data files...” The inclusion of the word averages leads the reader to question the temporal resolution of the LTSA. Did you actually average PSD values across multiple files? Or were PSD values for each file simply concatenated? Please clarify.

Lines 31-32: It is unclear how daily 1/3-octave-band medians were calculated given the units provided later in the manuscript and figures (I detail these below). I would assume that one would integrate the PSD values per file over the appropriate frequencies of the 1/3 Octaves in the units of uPa^2/Hz and multiply by the frequency resolution of the PSD, which in this case is 1 Hz, resulting in the units of pressure squared amplitude (uPa^2). Then when converting to dB this would be done as $10 \cdot \log_{10}(\text{pressure squared amplitude}/\text{uPa}^2)$, yielding the units of dB re 1 uPa . If I am not mistaking these units of dB are the standard form for band levels. Then for daily medians you would just bin all calculated values per day per 1/3 octave band and find the 50th percentile. Is this correct? I am asking this because, for example in Figure 4 and in the text, you refer to 1/3-octave-band medians as PSD in the units of dB re 1 uPa^2/Hz , which seems to deviate from the normal method of calculating and reporting these values. Please clarify.

Line 32: Why were seven-day moving averages necessary in this case? Please state that they were applied to the daily 1/3 octave band medians if this is true.

Page 5

Line 9: The authors state that the median PSDs of files recorded closes in time to each wind and rain scenario were plotted. Is it possible to clarify what this means? Were a certain number of samples selected before and after each event? I am just curious about the temporal window selection for these comparisons and how many samples were included when calculating these medians. Please clarify.

Line 14: Should there be an “E” after 173.0 and 176.5?

Line 17: Similar to wind and rain, how many files were selected when calculating the medians of these periods used to compare to wave height?

Line 32: Is there a reason that data were not reviewed for cetaceans above 100 Hz? If there is exists a reasoning for this, such as known species or target species, please include this information.

Line 34-35: Was there a criteria used to identify “PSD peaks” or was this just subjective based off the results in PSD plots? Basically, I am curious about what a peak is and how they were identified.

Page 6

Line 15: What are the “overview analyses?”

Line 19: “Above 100 Hz, median PSD values decreased by 3.6-4.3 dB re 1 $\mu\text{Pa}^2/\text{Hz}$ per kHz...” This might be the case for values above 1 kHz, but in Figure 2 it is clear that values dropped a lot more than that from 100 Hz to 1000 Hz. Please adjust the wording appropriately.

Lines 22-24: I would like to see more information about how the authors deduced that these dips were recorded due to the mooring components itself. Was this tested for in another experiment? Is there a reference that can be cited or perhaps a personal communication? It would seem like this information would be useful for others who use deep water moorings with subsurface buoys.

Line 27: “...across PSDs of 60 to 120 dB...” Is this an approximation or absolute range? Please clarify.

Lines 29-30: How does the presence of a bimodal result indicate that a “minority of files contained loud, low frequency sound energy?” Please elaborate as to why this is true. Seems like this pattern would not come out if it was a minority. Also the use of energy is not the appropriate term. Acoustic energy, intensity, and power are specific measurements with their own units.

Page 7

Line 4: The use of PSD to describe the units of 1/3-octave-bands might not be correct unless the authors used methods that are not 100% clear. Typically the units of bands are dB re 1 uPa and are not referred to as PSD levels either. This comment is related to other comments above.

Lines 5-7: "...1/3-octave-bands at Wairarapa were less influenced" by what? Right now it is a fairly empty sentence. Please clarify.

Line 16: Include (LTSA) after "time/frequency domain" if this is what you are referring to.

Page 8

Line 9: The use of "correlated" here and throughout, especially statements about "strongly correlated" should only be used when accompanied by a statistical test and result in my opinion.

Page 10

Section on Fish: You state in the discussion that no signals were detected during periods of suspected fish chorusing in the discussion section. Also mention this here, as it is important to show that you at least tried to verify that the increases in those bands were from fish chorusing. Your presentation of diel patterns helps make the case that it could be from fish.

Page 11

Line 32: Use of correlated.

Page 12

Line 10: Use of "These correlations..." without statistical tests. I would recommend another choice of wording.

Page 14

Paragraph Lines 11-14: Given that you could not identify the source of the putative fish chorus, can you provide any explanations into why you could not identify the sources, i.e. range, propagation, etc., as well as provide other explanations as to what it could be from, i.e. invertebrates. If there are no other logical explanations than to attribute them to fish, then maybe just address the former comment briefly.

Page 15

Line 34: Somewhere towards the end of this preceding paragraph, it would be good to include a statement about the possible effects of the environment and acoustic propagation on the results and differences you saw. The differences seen might not only be a factor of distances from the source but also of the environment. Of course this is a complete separate study, but it would good to include a statement about this possible confounding factor.

Page 16

Lines 16-18: Excellent. It is always a challenge to assess contributions to different sources that occupy similar frequencies. I am glad you mentioned this as a challenge. Do you think that these factors may have influenced your results and conclusions?

Page 17

Lines 5-6: Following this sentence it would be good to include a statement that one really needs to better understand the environmental effects on propagation and the received sound field.

Line 9: Correlated can be replaced with something like tracked...

Lines 14-15: Improper use of "intensities." Consider replacing with "amplitude" or "levels." Also similar to other comments, received levels are not only affected by the distance to sound sources but also the environment. Given the large differences in habitats and depths, your results are undoubtedly affected by the environment and cannot be directly compared necessarily. You touched on this with your discussion about TL as a function of site depth in the first paragraph of the Discussion, which was great and not only applies to surface sources but also distant sources that propagate through the water column.

FIGURES

Figure 2. Include in the caption the limitations on data quality given the dips at higher frequencies at the deep sites due to the mooring design.

Figure 4. Unless I do not fully understand how 1/3 median octaves were calculated, I do not think that PSD are the appropriate units. Please consider this and other comments above.

Figure 5. Consider adding the subplots with earthquakes and seismic surveys in a similar manner as done in Figure 4. If you do not agree that is fine as it might be easy to refer to both Figure 4 and 5 in the manuscripts final published form.

Figure 10. Consider adding supplementary .wav files for each spectrogram so that readers can hear what they are looking at.

Figure 12. Again please consider the appropriate units of band levels. Add "putative" or "suspected" to fish chorusing.

Figure 13. Consider including .wav files of this spectrogram as supplementary material. Also there is a lot going on in this spectrogram. Is the reader supposed to know which signals are the probably fish sounds? All of them or just a few of them? It is strange that all the signals overlap temporally across frequencies ranging from 50-2700 Hz. That is an extremely broad bandwidth for fish sounds originating from one species.

Appendix C

Leigh Marine Laboratory
The University of Auckland
PO Box 349
Warkworth, New Zealand

160 Goat Island Road, Leigh
Telephone: +64 4 – 386 0526
Cellphone: +64 21- 0828 5241
Email: vwar775@aucklanduni.ac.nz

23rd August 2020

RE: Manuscript RSOS-200117 'Marine soundscape variation in an oceanographically diverse region, central New Zealand'.

Dear Dr de Vos,

My co-authors and I would like to thank you for considering our above-referenced manuscript. We are very grateful for the positive, detailed, and constructive feedback from the two reviewers, and we are extremely thankful to have been given the opportunity to revise the manuscript based on this feedback. We have studied the review comments carefully and believe that the revised manuscript is now markedly stronger and more appropriate for publication by Royal Society Open Science.

Reviewer 1 was complimentary of our work, but wished to see the manuscript provide a more comprehensive synthesis and specific aim. Reviewer 2 was also very positive, particularly in regard to the holistic approach we took in the original submission. Per this feedback, the revised paper no longer merely reports the acoustic contributors to the marine soundscape of central New Zealand. Instead, the paper has been rewritten to demonstrate the utility of holistic soundscape analysis to map the spatio-temporal distributions of vocal species, as well as potential stressors, using central New Zealand as an exemplar case study. We believe the paper now provides a first insight into the marine soundscape of central New Zealand, while providing a comprehensive summary as to how soundscape analysis can be a valuable technique to inform conservation and management. The title of the paper has been revised to: "Marine soundscape variation reveals insights into baleen whales and their environment: a case study in central New Zealand".

A detailed summary of the reviewer's feedback appears below, with our responses given in bold type. Thank you for your consideration of the revised manuscript. Please do not hesitate to contact me for any further information. We look forward to hearing from you.

Yours sincerely,

Victoria Warren

Reviewer 1

A large amount of work clearly went into making the measurements and processing the data of this paper. The figures are well done, and the paper is well written. I had multiple technical and grammatical details that are captured in the sticky notes of the reviewed pdf. My main concern is in the messaging and packaging of the manuscript content. At present, the manuscript is more a description of the soundscape in New Zealand at a specific time. There are many papers in the literature that describe a soundscape at a particular place and time. It is already well known that soundscapes vary with biotic, abiotic, and anthropogenic contributions. What this manuscript needs is a focusing question that is answered or a testing of an ecological concept. In both the Intro and Discussion, the authors correctly point out that soundscapes can provide information about the ecological function of a soundscape, but this was not successfully accomplished in the paper. A more comprehensive synthesis or application to a specific need or question would greatly increase the value of this work. It may be more appropriate for a regional regulatory or management journal.

- **Thank you for these kind words, we were pleased to hear that you liked the figures and writing style in general. To address the main concern of this reviewer, the paper has been rewritten to demonstrate the utility of soundscape analysis to map the spatio-temporal distributions of vocal species and potential stressors, using central New Zealand as an exemplar case study.**

Throughout

- Use Oxford commas
 - **Oxford commas inserted throughout manuscript. I hope we got them all!**

P1

- If you use the term 1/3 octave band, you must specify if this is 1/3 octave base 10 or 1/3 octave base 2. For clarity, the term decidecade band is being more widely used instead of 1/3 octave and is defined in ISO 18405. I have highlighted the next instance of this term, but it needs to be addressed throughout the paper.
 - **Switched to decidecade bands, instead of 1/3 octave bands**
- “long-term, holistic monitoring option” - or soundscapes? sources? both?
 - **This sentence is no longer included**

P2

- “rain strength” – would intensity be a better word
 - **No longer mentioned**

- “sea ice movement” - Wouldn't sea ice movement also be considered seasonal, like biotic contributions?
 - **No longer mentioned**
- “vocalisations produced by marine mammals, birds, fish and invertebrates to...” - Not all birds, fish, and inverts produce sound with vocal chords. More accurate to say sounds or sonations instead of vocalisations.
 - **This sentence is no longer included**
- “voluntary and involuntary sounds” - Suggest - both directly and indirectly produced sounds
 - **This sentence is no longer included**
- “Biophonic contributors to the soundscape have evolved to produce sounds with minimal overlap, either in frequency, time or space (acoustic niche hypothesis (23)).” - Some, not all, contributors.
 - **This sentence is no longer included**
- “Anthropogenic sounds are relatively recent additions to soundscapes” – over evolutionary time scales
 - **This sentence is no longer included**

P3

- “Given the prevalence and concentration of geological, meteorological, biological and increasing anthropogenic activity around central New Zealand, there has been a growing need to quantify the soundscape of this region. Autonomous acoustic recorders were deployed to quantify the marine soundscapes around central New Zealand and identify the relative spatial and temporal contributions of geophonic, biophonic, and anthropogenic sources.” - This was also stated in the Summary, but it doesn't say why it is needed. Just because a region is biodiverse or has lots of human activity is not justification for measuring soundscapes. What is the motivation or need to measure soundscapes in this region?
 - **The paper has been rewritten to demonstrate the utility of soundscape analysis to map the spatio-temporal distributions of vocal species as well as potential stressors, using central New Zealand as an exemplar case study.**

P4

- “Only files with 16 kHz sampling rate were considered in this study.” - Why were only the 16 kHz sampled recordings used? These are the lowest and most far ranging frequencies that have the potential to capture the same sources across recording sites. The higher frequency recordings would be more independent across sites and relate to a more "local" soundscape compared to the more "regional" low frequencies.

- **Unfortunately, due to time constraints, it was not feasible to analyse all of the data, and the 16 kHz data provided the largest sample. Also, as we now explain, the paper focuses on low frequency baleen whale calls, so frequencies exceeding 8 kHz would not have provided extra information.**
- “to omit high energy infrasound, and the effect of Nyquist roll-off” – remove comma
 - **Done**
- “Hamming window” - I believe this commonly referred to as a Hann window. Or do you mean Hamming window? They are two different things.
 - **We now state that PSD was calculated using the ‘pwelch’ function in MATLAB, where the default window function is ‘Hamming’.**

P5

- In Table 2, there are no consistent definitions for Low, Mid, and High rain accumulations. For example, 0.9-1.3 is considered High for Blenheim but Mid for Waipara. It would be more consistent to have clearly defined ranges for Low, Mid, and High. That may mean that a site like Blenheim has no periods with High rainfall, and that is acceptable. The same comment for consistent category definitions also applied to Table 3
 - **In response to this comment, we focused on only one weather station (Wellington) and split the ranges from this location into three: Low, Mid, and High. Furthermore, for wave heights, we took the full range of data across all recording locations and split the full range into Low, Mid, and High, instead of having different wave height bins per location.**
- Wind was always present as indicated in Table 3. There was no period when wind was zero. Hence, wind should be a consistent time series. Can you please clarify how the closest time to each wind scenario applies?
 - **Average wind speed was obtained for every hour, and was therefore discrete. When the wind speed was classified as High during an hour (for example), the 630 s acoustic file that was recorded closest to the hour (e.g. 12:00) was taken as the representative file and incorporated into the median.**
- As noted above for Tables 2 and 3, Table 4 should have consistent thresholds or definitions for the Low, Mid, and High. Also, waves are constantly present (no zeros in Table 4).
 - **See comment above relating to Table 2.**
- “Hamming window” - In Acoustic Processing, you used a Hann window. Here you are using a Hamming window. Why the difference?

- **The default window used for PSD processing in MATLAB was Hamming, not Hann, and we continued to use Hamming here too (in PAMlab-lite software)**
- Why was data only viewed up to 100 Hz when the sampling rate was 16kHz, giving a usable 8 kHz band? That seems like a lot of data to disregard. It would only reflect the low frequency baleen whales.
 - **We reviewed all data that caused noticeably raised PSD events in the 16 kHz files. It was a mistake to have worded the original sentence this way, and indeed, this sentence is no longer included.**
- Very often, quiet marine mammal calls do not result in an elevated mean PSD. In addition, during high levels of human activity, marine mammal calls can be observed between loud pulses. How did you account for these quiet and potentially masked calls?
 - **As should now be evident in the revised paper, we have focused on looking at the soundscape from a broad overview perspective (i.e. the boxed marine mammal sounds in Figure 6), rather than trying to detect and identify all individual sounds.**

P6

- “Given that biophonic contributions can feature diel cycles, specific times (e.g. dusk) were obtained for the location of Wellington (<https://www.gaisma.com/en/location/wellington.html>).” - Can you please clarify what you mean here? I'm assuming you mean times of dusk/dawn/sunset/etc. were obtained at the Wellington site and applied to all sites?
 - **“For other suspected biophonic sounds that were not thought to be produced by marine mammals, sound levels were examined against day and night time indicators (obtained for the location of Wellington and applied across all sites (<https://www.gaisma.com/en/location/wellington.html>)) as biological sounds often feature diel cycles as well as seasonal trends.”**
- There are many sources that vocalize and overlap with the 63 and 125 Hz decade bands (1/3 octave base 10). How did you separate out the contribution of vessels from other sources?
 - **We do not claim that other sources did not contribute to these bands, but we analysed the 63 and 125 Hz bands as these are globally accepted bands where ship sound concentrates. Quote from the European Marine Strategy Framework Directive (Tasker reference included in the manuscript): “Furthermore, focussing on bands where most shipping noise is concentrated (e.g. 63 and 125 Hz 1/3 octave bands) would be a cost effective approach.”**

- What was the noise floor of your recording system? This information was not provided in the Methods and is critical to accurate interpretation of the sound level values on the low end.
 - **Now included: “The pressure spectral density noise floor was limited by the hydrophone preamplifier at 32 dB re 1 μPa^2 / Hz.”**
- “This was an artefact resulting from destructive interference due to the separation distance between the AMAR and hydrophone and the spherical glass floats deployed at these deep-water locations and was not a true feature of the soundscape.” - Do you have a reference for this? or any other data to support this explanation?
 - **This was the manufacturers explanation for the artefact, I’m afraid I do not have a reference for this.**
- I'm not seeing a bimodal pattern. Maybe best to point this out on Figure 3 with arrows or symbols of some sort? Are you referring to the visible separation of the upper SPD as bimodal? I'm not sure that is the most accurate or technically correct term.
 - **Yes, this was the feature we were referring to. We have revised the explanation without using the word ‘bimodal’ now, so hopefully it is clearer. “At Wairarapa and the first deployment at Kaikōura, a visible separation occurred in the SPDs at frequencies below 250 Hz and 100 Hz, respectively (Figures 3C and 3D). At these locations, the two low frequency SPD modes were centred around 80 dB re 1 μPa^2 / Hz and 100 dB re 1 μPa^2 / Hz (Figures 3C and 3D). This implied that some of the data from Wairarapa and Kaikōura contained low frequency sound energy that was approximately 20 dB re 1 μPa^2 / Hz louder than data recorded at other times during the deployments.”**

P7

- These results are hard to interpret because there was not a consistent threshold or definition of low, mid, and high. This is probably why you saw no clear trend. If the values were mixed across sites, it would be impossible to see a trend.
 - **We have now revised the Low, Mid, and High definitions, and continue to see the clear trends that were presented in the original version of this paper: increased rainfall, wind speed and wave height lead to increased sound levels in specific frequency bands.**

P11

- Much of the Discussion content is either redundant with the Results or would be more appropriate in the Results section. I look for content that assimilates the variety of results into a more comprehensive message than just a list of observations.

- **Thank you for this honesty, this was perhaps the most useful review comment. Per the feedback, the paper has been rewritten to demonstrate the utility of soundscape analysis to map the spatio-temporal distributions of vocal species as well as potential stressors, using central New Zealand as an exemplar case study. We believe the Discussion now brings together the Results, makes interpretations, and provides a comprehensive summary of the key findings of the paper and soundscape analysis as a research technique.**
- **“Soundscape quantification is useful for investigating ecosystem functioning in a non-invasive manner” - Very true, but i don't see this assimilation in the present study. How does what you did relate to ecosystem function?**
 - **We no longer talk about ecosystem functioning in regard to the recorded soundscape, and instead have focused on the way in which soundscape analysis can provide insights into the spatio-temporal distributions of vocal species, as well as the spatio-temporal distributions of potential stressors.**
- This information is more appropriate for the Results section. The actual equation used should also be included. Given the coastal locations, I would argue a simple spherical spreading model is not appropriate. A PE model would better capture the propagation losses.
 - **Given that this model/calculation is not vital to the paper, we now provide a simplified comment in the Discussion: “For sounds with an abiotic origin, particularly those related to weather events, the lower sound levels at the deeper recorders likely resulted from increased propagation loss due to the greater distance from the surface.”**

Reviewer 2

General Comments

After reviewing the manuscript, I commend the authors for their research and valuable contribution to the field of underwater acoustics and soundscape analysis. The study examines the components of and contributors to the soundscapes of four spatially separated locations in the region of central New Zealand. As the authors state, this region of the ocean has been understudied, and the authors present and make a strong case that passive acoustic monitoring can help fill in data gaps at this location and elsewhere. I was especially impressed with the holistic approach that the authors took, and their ability to tease out different acoustic sources through comparisons of different acoustic metrics with ancillary data types. The inclusion of earthquakes was a nice addition. Very rarely do studies have the ability to document biophonic, geophonic, and anthropogenic contributions to soundscapes in a single study. However, this manuscript exemplifies this potential and will serve as a value contribution and reference for many ongoing studies in the field of soundscape analysis. Thus, after minor edits and further explanations of methods, which I detail below, I would expect this manuscript to be of great interest to the readers of Royal Society Open Science.

- **We are grateful for these kind words about the original paper. We believe the revised paper still provides the commended aspects (information about an understudied region), while meeting the recommendations of Reviewer 1 as well. We emphasised the holistic approach, which was praised by Reviewer 2, and we believe the paper is now strengthened in this regard.**

Specific Comments

Page 1

- Line 21-22: “Median power spectral density.” The use of “power” when referring to this term is widely used, but the authors may consider the use of “median pressure spectral density” instead, as nowhere in PSD is there included power, which has the units of watts. PSD can be seen as proportional to power. This comment is just a recommendation to use the appropriate terms; although I know others may disagree as the use of Power Spectral Density is a legacy term. If accepted, the authors would want to make changes throughout the manuscript.
 - **We now report the pressure spectral density noise floor of the recorders in the Methods, although we continue to use power spectral density in the Results as this is a commonly accepted term and we believe it will be most widely understood.**
- Line 23: The authors use “correlated” throughout the manuscript, yet one would need to also provide a statistical test and results if statements about

correlations are made. Consider using a different term throughout the manuscript or provide a statistical test.

- **Thank you for this insight – we have removed the word ‘correlated’ throughout the manuscript.**

Page 2

- Line 18: The use of “vocalisations” is not appropriate for fish and invertebrates. Consider changing this word to “sounds.”
 - **No longer included**

Page 3

- Line 15: An abbreviation is provided for South Taranaki Bight in this line, yet this site is also written out fully on Page 4 (Line 5-6) of the methods. I would suggest omitting the abbreviation from this line so that readers can refer to a full list of abbreviations on the following page.
 - **No longer included here**

Page 4

- Line 11: I would appreciate some further details about the flow shield. What was it made out of? Can you include a graphic of your different mooring designs? Have the effects of the flow shield on received levels been assessed? While this comment maybe be selfish, I foresee other readers being interested in incorporating flow shields into their studies, so some further details are warranted. Additionally, without knowing about possible effects of the flow shield, one could speculate that the results of the study might have been affected by its presence at two of the four locations.
 - **The flow shield was present on all four recorders, so would not have affected any location more than any other. To avoid confusion, we have removed the mention of it.**
- Lines 24-33: While this section on “Acoustic Processing” is nice and concise, I think that some details need to be expanded upon or clarified given its importance to the rest of the study.
 - **We have expanded and clarified all aspects of the Methods**
- Line 24: “Power” ◇ “Pressure” ?
 - **See previous comment. In this case: power**
- Line 26: 80% overlap seems like a strange choice over the typical 0%, 50%, 75% options. Was there any reason why this was chosen? If so, include it. If it is just what the authors routinely use, then they can ignore this comment.
 - **80% overlap was chosen to allow for a large amount of overlap. In hindsight, 75% may have been a better value to choose. It would require a significant amount of computing power to recalculate.**

- Line 28: Please indicate from what data the exceedance percentiles were calculated from. I am assuming that they were calculated from the PSD values per each 630 s file, but this is not 100% clear given that this sentence follows a sentence about SPD calculations. Also were these calculated for the duration of each deployment, or per month, etc.? It would seem different time periods were selected based off the analysis, e.g. per deployment vs. per geophonic event period.
 - **To avoid confusion, exceedance percentiles are no longer included in the paper.**
- Line 30: "...concatenating the 1 Hz resolution PSD averages for all data files..." The inclusion of the word averages leads the reader to question the temporal resolution of the LTSA. Did you actually average PSD values across multiple files? Or were PSD values for each file simply concatenated? Please clarify.
 - **The latter. Revised to: "long-term spectral averages (LTSAs) were generated by concatenating the 1 Hz resolution PSD data for all files and displaying as scaled colours."**
- Lines 31-32: It is unclear how daily 1/3-octave-band medians were calculated given the units provided later in the manuscript and figures (I detail these below). I would assume that one would integrate the PSD values per file over the appropriate frequencies of the 1/3 Octaves in the units of uPa^2/Hz and multiply by the frequency resolution of the PSD, which in this case is 1 Hz, resulting in the units of pressure squared amplitude (uPa^2). Then when converting to dB this would be done as $10 \cdot \log_{10}(\text{pressure squared amplitude}/\text{uPa}^2)$, yielding the units of dB re 1 uPa . If I am not mistaking these units of dB are the standard form for band levels. Then for daily medians you would just bin all calculated values per day per 1/3 octave band and find the 50th percentile. Is this correct? I am asking this because, for example in Figure 4 and in the text, you refer to 1/3-octave-band medians as PSD in the units of dB re 1 uPa^2/Hz , which seems to deviate from the normal method of calculating and reporting these values. Please clarify.
 - **Thank you for drawing this to our attention. This was indeed a mistake, and the band medians have now been correctly calculated, with the units of dB re 1 μPa . Values were calculated as follows:**
 - **PSD values were converted to linear space, integrated within each band, multiplied by the 1 Hz frequency resolution, and converted back to dB. Daily medians were obtained *via* the 50th percentile.**
- Line 32: Why were seven-day moving averages necessary in this case? Please state that they were applied to the daily 1/3 octave band medians if this is true.
 - **Seven-day moving averages have not been applied to the decidecade bands in the revised manuscript.**

- Line 9: The authors state that the median PSDs of files recorded closes in time to each wind and rain scenario were plotted. Is it possible to clarify what this means? Were a certain number of samples selected before and after each event? I am just curious about the temporal window selection for these comparisons and how many samples were included when calculating these medians. Please clarify.
 - **When the wind speed was classified as High during an hour (for example), the 630 s acoustic file that was recorded closest to the hour (e.g. 12:00) was taken as the representative file and incorporated into the median.**
- Line 14: Should there be an “E” after 173.0 and 176.5?
 - **There should have been, thank you**
- Line 17: Similar to wind and rain, how many files were selected when calculating the medians of these periods used to compare to wave height?
 - **See above comment**
- Line 32: Is there a reason that data were not reviewed for cetaceans above 100 Hz? If there is exists a reasoning for this, such as known species or target species, please include this information.
 - **No longer pertinent to the revised manuscript**
- Line 34-35: Was there a criteria used to identify “PSD peaks” or was this just subjective based off the results in PSD plots? Basically, I am curious about what a peak is and how they were identified.
 - **This was subjective, and in the revised manuscript, basically corresponds to the boxed events in Figure 6.**

Page 6

- Line 15: What are the “overview analyses?”
 - **This was referring to the PSDs, SPDs and LTSAs. This text is no longer included**
- Line 19: “Above 100 Hz, median PSD values decreased by 3.6-4.3 dB re 1 uPa²/Hz per kHz...” This might be the case for values above 1 kHz, but in Figure 2 it is clear that values dropped a lot more than that from 100 Hz to 1000 Hz. Please adjust the wording appropriately.
 - **We no longer present the data in terms of decline per kHz**
- Lines 22-24: I would like to see more information about how the authors deduced that these dips were recorded due to the mooring components itself. Was this tested for in another experiment? Is there a reference that can be cited or perhaps a personal communication? It would seem like this information would be useful for others who use deep water moorings with subsurface buoys.
 - **This was the manufacturers explanation for the artefact, I’m afraid I do not have a reference for this.**

- Line 27: “...across PSDs of 60 to 120 dB...” Is this an approximation or absolute range? Please clarify.
 - **We have clarified absolute and approximate ranges throughout the revised manuscript**
- Lines 29-30: How does the presence of a bimodal result indicate that a “minority of files contained loud, low frequency sound energy?” Please elaborate as to why this is true. Seems like this pattern would not come out if it was a minority. Also the use of energy is not the appropriate term. Acoustic energy, intensity, and power are specific measurements with their own units.
 - **Revised to: “This implied that some of the data from Wairarapa and Kaikōura contained low frequency sound that was approximately 20 dB re 1 μ Pa² / Hz louder than data recorded at other times during the deployments.”**

Page 7

- Line 4: The use of PSD to describe the units of 1/3-octave-bands might not be correct unless the authors used methods that are not 100% clear. Typically the units of bands are dB re 1 uPa and are not referred to as PSD levels either. This comment is related to other comments above.
 - **Addressed above**
- Lines 5-7: “...1/3-octave-bands at Wairarapa were less influenced” by what? Right now it is a fairly empty sentence. Please clarify.
 - **Sentence is no longer included**
- Line 16: Include (LTSA) after “time/frequency domain” if this is what you are referring to.
 - **Sentence is no longer included**

Page 8

- Line 9: The use of “correlated” here and throughout, especially statements about “strongly correlated” should only be used when accompanied by a statistical test and result in my opinion.
 - **We have removed the word ‘correlated’ throughout the manuscript.**

Page 10

- Section on Fish: You state in the discussion that no signals were detected during periods of suspected fish chorusing in the discussion section. Also mention this here, as it is important to show that you at least tried to verify that the increases in those bands were from fish chorusing. Your presentation of diel patterns helps make the case that it could be from fish.
 - **We no longer state that this is a fish chorus, and instead describe it as a biological chorus. We propose that it could be fish in the Discussion.**

Page 11

- Line 32: Use of correlated.
 - **Now omitted.**

Page 12

- Line 10: Use of “These correlations...” without statistical tests. I would recommend another choice of wording.
 - **Now omitted**

Page 14

- Lines 11-14: Given that you could not identify the source of the putative fish chorus, can you provide any explanations into why you could not identify the sources, i.e. range, propagation, etc., as well as provide other explanations as to what it could be from, i.e. invertebrates. If there are no other logical explanations than to attribute them to fish, then maybe just address the former comment briefly.
 - **The biological chorus is now discussed in the Discussion, and proposed as fish due to similarities with a fish chorus reported elsewhere.**

Page 15

- Line 34: Somewhere towards the end of this preceding paragraph, it would be good to include a statement about the possible effects of the environment and acoustic propagation on the results and differences you saw. The differences seen might not only be a factor of distances from the source but also of the environment. Of course this is a complete separate study, but it would good to include a statement about this possible confounding factor.
 - **We now discuss the effect of the local environment on recorded soundscapes in the Discussion section ‘Marine soundscape variation and overlaps between contributors’.**

Page 16

- Lines 16-18: Excellent. It is always a challenge to assess contributions to different sources that occupy similar frequencies. I am glad you mentioned this as a challenge. Do you think that these factors may have influenced your results and conclusions?
 - **Hopefully the reviewer agrees that we now discuss this in more detail!**

Page 17

- Lines 5-6: Following this sentence it would be good to include a statement that one really needs to better understand the environmental effects on propagation and the received sound field.
 - **We now discuss the effect of the local environment on recorded soundscapes in the Discussion section ‘Marine soundscape variation and overlaps between contributors’.**
- Line 9: Correlated can be replaced with something like tracked...
 - **No longer included**
- Lines 14-15: Improper use of “intensities.” Consider replacing with “amplitude” or “levels.” Also similar to other comments, received levels are not only affected by the distance to sound sources but also the environment. Given the large differences in habitats and depths, your results are undoubtedly affected by the environment and cannot be directly compared necessarily. You touched on this with your discussion about TL as a function of site depth in the first paragraph of the Discussion, which was great and not only applies to surface sources but also distant sources that propagate through the water column.
 - **Hopefully the revised Discussion addresses this more appropriately.**

Figures

- Figure 2. Include in the caption the limitations on data quality given the dips at higher frequencies at the deep sites due to the mooring design.
 - **Now included**
- Figure 4. Unless I do not fully understand how 1/3 median octaves were calculated, I do not think that PSD are the appropriate units. Please consider this and other comments above.
 - **Now corrected**
- Figure 5. Consider adding the subplots with earthquakes and seismic surveys in a similar manner as done in Figure 4. If you do not agree that is fine as it might be easy to refer to both Figure 4 and 5 in the manuscripts final published form.
 - **Now included in revised Figures 6 and 7**
- Figure 10. Consider adding supplementary .wav files for each spectrogram so that readers can hear what they are looking at.
 - **We do not believe this is necessary - similar sounds can easily be found online.**
- Figure 12. Again please consider the appropriate units of band levels. Add “putative” or “suspected” to fish chorusing.
 - **Now corrected, and described as ‘suspected biological chorus’.**
- Figure 13. Consider including .wav files of this spectrogram as supplementary material. Also there is a lot going on in this spectrogram. Is the reader supposed to know which signals are the probably fish sounds? All of them or just a few of them? It is strange that all the signals overlap temporally across

frequencies ranging from 50-2700 Hz. That is an extremely broad bandwidth for fish sounds originating from one species.

- **This figure is no longer included.**

Again, we would like to extend our sincere thanks to the two reviewers for these very helpful and constructive suggestions which have greatly improved the manuscript.

10th February 2021

Dear Prof Cicuta,

Re: Manuscript RSOS-201503

My co-authors and I would like to thank you for considering our above-referenced manuscript. We are very grateful for the positive and constructive feedback from the reviewers. We have carefully considered the reviewers' comments and have revised the manuscript accordingly. A detailed summary appears below, with our responses provided in bold.

During the time this manuscript has been under review, two additional papers pertaining to baleen whale distributions in this region have been published:

- Warren *et al.*, (2020) Migratory insights from singing humpback whales recorded around central New Zealand. *Royal Society Open Science*, 7: 201084.
- Warren *et al.*, (2021) Passive acoustic monitoring reveals spatio-temporal distributions of Antarctic and pygmy blue whales around central New Zealand. *Frontiers in Marine Science*, 7:575257.

Accordingly, we have updated the text of this manuscript to reflect these other publications. In the final paragraph of the Introduction (P2 L42) we added the citations of both papers (references 42 and 43) in relation to the statement that Cook Strait hosts migrating baleen whales. We have not updated the Methods or Results sections of this manuscript with any additional information as the results remain supported by the other papers and any supplementary details provided in the other papers were obtained using different methods. In the Discussion, we have added relevant information arising from the two additional papers:

- P8 L20-27: “At the soundscape level presented here, Antarctic blue whales were detected along the shelf edge to the east of central New Zealand... Conversely, humpback whales were detected in the shallow, sheltered bay of the South Taranaki Bight (STB), suggesting that their northbound migration route followed the opposite coast to the Antarctic blue whales. **However, when the calls of these two species were examined in the data using a targeted approach rather than an overall soundscape-level approach, both**

Antarctic blue whales and humpback whales were found to utilise migration routes along both the east and west coasts of central New Zealand (42, 43)."

- P8 L28-32: "In the present study, neither Antarctic blue whale calls or humpback whale song were noted in the soundscape during austral spring when southbound migration would be expected (38). It is possible that these species migrated southward through the study area without producing an abundance of calls, and were therefore not evident in the overall soundscape; **indeed, Antarctic blue whale calls were detected in the region during austral spring when the data were analysed using a targeted approach (43).**"
- P8 L32-34: "Migratory routes are assumed to be stable, at least over short evolutionary timescales (72), however, neither humpback nor Antarctic blue whale calls were notable in this soundscape-level analysis during winter 2017." – This is a slight re-wording to emphasise the soundscape-level nature of this analysis. Antarctic blue whale calls were evident in the data during winter 2017 when it was analysed at a finer level (Warren et al 2021).

Thank you for your consideration of this revised manuscript, please do not hesitate to contact me if you would like to see any further changes.

Yours sincerely,

Victoria Warren

Reviewer 2

I appreciated the new version of the manuscript and the approach that the authors took. The authors now frame the study to describe the acoustic presence of baleen whales and changes in the soundscape that they may perceive, which includes an in-depth look at sources of variation. I believe that the authors have addressed all my previous comments from the first review. I only have a few minor comments, which I detail below.

Page 3

Line 1: "varied" I would recommend using the present tense here (vary) as you have not introduced the results yet.

- **Changed to 'vary'**

Line 16: “with the hydrophones mounted to the AMARs.” This statement is not necessary, correct? You already stated that the AMARs were moored on vertical line moorings. I am assuming that the AMARs already incorporate hydrophones?

- **That is correct. We were trying to distinguish the difference between the bottom-mounted AMARs at Cook Strait and STB from the vertical AMARs at Wairarapa and Kaikōura. We’ve removed this clause from the sentence.**

Line 30: I do not think that you need a comma before “and per month” nor before “at each station.”

- **Commas removed.**

Line 39: “energy” is not the appropriate term here. I recommend replacing energy with levels or inputs.

- **Changed to ‘levels’.**

Page 4

Line 2: This sentence would benefit from the addition of a citation or two.

- **We have now added three example references to papers that discuss seasonal and diel variation in biophonic sound sources: Leroy et al 2016 (Antarctic blue whales), McCauley et al 2016 (Myctophidae fish), Jamie et al 2017 (coral reefs).**

Lines 5-7: Decidecades are exactly the same as 1/3 octave bands assuming that they are base ten bands, if I am not mistaken. These two sentences open the door for confusing the reader. I would recommend not even mentioning 1/3 octave bands here and just refer to the measurements as decidecade bands as they are understood by the community and the preferred term by Reviewer 1 in the previous review.

- **We have removed the references to 1/3 octave bands as we also prefer the term decidecade.**

Page 5

Lines 33-35: Given the bandwidth of the biological sound, it is likely that the rise in PSD levels might originate from more than one source. Consider adding in this possibility. Additionally in regards to Figure 8, the duration of daily increases in the 1000 Hz decidecade is much longer than one might expect from a single species if these daily increases are definitely biologic in origin, especially if it was produced by a fish species. However, it could be related to feeding sounds of mixed-fish-species origin. Another possibility would be from invertebrates I suppose. In any event, it would be good to include that it might originate from multiple sources.

- **We agree and have altered the sentence to say “but the seasonal and diel patterns of the sound implied that it was produced by one or more biophonic sources”**

Line 10: Any idea why the high tide at Wairarapa would yield the loudest median sound levels below 100 Hz? Could another source of sound or pressure contribute to this result?

- **We honestly have no idea! We have discussed with local oceanographers (Dr Craig Stevens and Dr Matt Pinkerton, both at NIWA), but we've not come to any conclusions. We thought there may possibly have been some strum in the mooring, but would have expected this to occur during flood or ebb tide rather than high tide.**

Figure 4: In response to my suggestion for including .wav files as supplementary material for this figure, the authors stated that "We do not believe this is necessary - similar sounds can easily be found online." I would suggest that they reconsider as I cannot find an example of a pygmy blue whale online, or at least easily.

- **Thank you for your recommendation. We agree that pygmy blue whale calls are difficult to find online, and we have therefore included the four baleen whale calls visualised in Figure 4 as supplementary .WAV files.**

Figure 6: It is difficult to see the components of the LTSAs as presented. Would it be possible to have two figures; one for the first deployment and one for the second deployment? Alternatively, could this figure be presented in "landscape" format. I would like to be able to see the details a bit better. For example, the biological chorus is virtually undetectable in the LTSAs. Maybe this can be addressed in the proofing stage.

- **We would prefer not to split Figure 6 into two as it currently displays the full temporal extent of the seismic survey at Wairarapa. When the figure is viewed with higher resolution the biological chorus is more evident, and we are hopeful that the journal will include this figure in landscape format.**

Reviewer 3

This is a nice paper giving a rarely done soundscape analysis of a particular study region - the details are obviously pertinent to that region, which is quite specific, but the notion of how one does a soundscape analysis, what it is good for, and what it can potentially show, is general and therefore worth publishing in a general journal like this. The analyses are not statistical, although they are clearly highly quantitative. I am mindful that I am reviewing a revised manuscript, and have tried to do so in the context of how the authors have responded to the original comments, rather than as a new submission per se. It's clear to me that the authors have responded carefully and fully to the original reviewer reports, and I don't think it's fair to ask for further major revisions in the light of

that. If it were coming to me as an original submission I might have asked the authors to reflect on how they might provide statistical support for some of the observations they make from the quantitative patterns, but I don't think it's fair in this case - there's nothing wrong with what they did, and so perhaps the authors can take my comment here as something to reflect on when planning future work of this kind. Other than that I have only minor comments.

- **Thank you for this feedback, we appreciate your honesty on these points, and will consider your thoughts in future manuscripts. Given that the soundscape of this region had never been studied before, we proceeded with a fairly qualitative overview here, and imagine that future studies on this topic will be more quantitative as further measurements can now be compared to a baseline.**

I am kind of puzzled by the insistence of one reviewer to use the obscure term 'decidecade' instead of the widely understood and used 1/3 octave - the bands referred to are clearly using the base 2 octave as they are the standard ISO / Acoustical Society bands, so it's not clear that this is correct in any case. I realise this is perhaps frustratingly contradictory for the authors so I would leave it up to them but to my mind the previous reviewer's comment is unnecessary and impedes understanding/communication. It would particularly eliminate the confusion in P20 L5-8.

- **We have decided to proceed with the term 'decidecade' and have removed the mention of 1/3 octave bands per the other reviewers comments, so there should be no longer be any confusion.**

P1

L20 'contributors' is unclear

- **'Contributors' changed to 'acoustic elements'**

L23 strike 'and'

- **Removed**

P18

L25 additional to what? suggest removing

- **Assuming the reviewer meant P2 L25 "...and additional spectral overlap between contributors can be indicative of..." – we have removed the word additional. We had been trying to point out the difference between sounds that occur in the same time and place that might be outside of audible range and therefore of little consequence, compared to sounds in the same time and place with additional frequency overlap that could cause direct interference.**

P19

L38 'periods of raised PSD' is a bit unclear - can the authors specify was this in a frequency band or broadband, and what was used to identify 'raised'? Some quantitative idea would help (e.g. over a certain level, or percentile of the data)

- **P3 L38/39 edited to read: “When PSD levels were noticeably higher than average in a particular frequency band, the data files corresponding to these periods were individually examined to determine the origins of the increased acoustic levels.”**

P19

L40 would be good practise to just mention that spectrogram parameters were chosen to best display the species vocalisations here?

- **We have updated the text to read “Specific features within the data were visualised as spectrograms in PAMlab-Lite (45) using spectrogram parameters that best displayed the features of interest.”**

Is Fig 7 really needed given Fig 6? In general the authors could consider cutting down the number of figures

- **Thank you for this feedback. We agree that the figures are numerous, but we feel that they all illustrate different aspects of the data. Unlike Figure 6, Figure 7 shows the concentration of ship sound in Cook Strait (raised 63 and 125 Hz bands) and provides a more striking visualisation of the bands raised by the seismic survey and earthquakes. If we needed to reduce the total number of figures, this could be revisited.**

Again, we would like to extend our sincere thanks to the reviewer for these constructive suggestions which have improved the manuscript.